# Isomeranzin activates Gnas-AMPK signaling to drive white adipose browning and curb obesity in mice

Menghao Shi[1,2,7], Yinsong Ye[2,3,7], Lizhi Hu[2,4,5,7], Yibo Yan [2,3], Shushu Jiang [1,2], Pengchao Wang[1,2], Fengcen Li[1,2], Mingfa Ai[1,2], Jinhui Huang[1,2], Ling Yang [4,5,6 ✉], Kai Huang [2,4,5 ✉] & Minglu Liang [2,4,5 ✉]

## Abstract

**Obesity is a major global health challenge, and promoting the browning of white adipose tissue (WAT) represents a promising therapeutic strategy. However, pharmacological approaches to induce adipose thermogenesis remain limited. Through a Connectivity Map-based screen, we identified isomeranzin (ISM) as a potent small-molecule activator of WAT browning. ISM enhances thermogenesis in adipocytes by activating the AMP-activated protein kinase (AMPK) pathway. Integrated limited proteolysis-mass spectrometry, cellular thermal shift assays, and molecular docking identified guanine nucleotide-binding protein G(s) alpha subunit (Gnas) as the direct binding target of ISM. Mechanistic studies further revealed that ISM induces WAT browning through the Gnas-dependent activation of cAMP-AMPK signaling cascade. These findings elucidate the molecular mechanism underlying ISM activity and highlight its potential as a lead compound for enhancing energy expenditure and combating obesity.**

**Keywords** Obesity; Adipose Browning; Isomeranzin; Gnas; AMPK
**Subject Category** Metabolism

## Introduction

Obesity is defined as a body mass index (BMI) greater than or equal to 30 kg/m², characterized by excessive fat accumulation within the body, and is a multifaceted disease influenced by various factors (Kopelman, 2000). Its prevalence is rising at an alarming rate, posing significant challenges to global public health (Ng et al, 2014). This condition not only impacts an individual's physical appearance but also markedly increases the risk of chronic diseases such as type 2 diabetes, cardiovascular diseases, hypertension, and certain cancers (Haslam and James, 2005). The development of obesity is primarily attributed to a prolonged imbalance between energy intake and expenditure, closely linked to the energy homeostasis of adipose tissue (Singh et al, 2021). White adipose tissue (WAT) serves as the principal energy storage organ and plays a pivotal role in obesity progression. In contrast, brown adipose tissue (BAT) is endowed with abundant mitochondria and uncoupling protein 1 (Ucp1), conferring non-shivering thermogenic capabilities and thereby offering potential metabolic advantages in energy expenditure and body weight management (Cohen and Kajimura, 2021). Recent studies have revealed that, in addition to conventional brown adipocytes, there exists a distinct type of beige adipocytes. The development of beige adipocytes within the WAT depot can be induced through cold exposure, β-adrenergic receptor (β-AR) agonist stimulation, or physical exercise, a phenomenon collectively referred to as WAT browing (Bartelt and Heeren, 2014). Recent studies increasingly suggest that promoting the browning of WAT could represent a novel therapeutic strategy for obesity and its associated metabolic disorders (Kuryłowicz and Puzianowska-Kuźnicka, 2020).

The pathways and regulatory mechanisms underlying adipocyte browning are intricate, encompassing multiple molecular and cellular levels. The β3-AR pathway is critical in inducing adipocyte browning; Activation of β3-AR increases the levels of cyclic adenosine monophosphate (cAMP) in adipocytes, thereby activating the protein kinase A (PKA) signaling pathway, promoting the upregulation of thermogenic gene expression and mitochondrial biogenesis (Liu et al, 2019). The adenosine monophosphate-activated protein kinase (AMPK) pathway also plays a crucial role in regulating adipocyte browning; AMPK activation promotes Ucp1 expression and mitochondrial biogenesis by upregulating the expression and activity of peroxisome proliferator-activated receptor gamma coactivator-1 alpha (Pgc1α), thereby enhancing browning capacity in adipocytes (Wang et al, 2021). Although β3-AR agonists have demonstrated significant anti-obesity effects in animal models (Abdul Sater et al, 2022), their safety and efficacy in humans remain to be thoroughly investigated. Most β3-AR agonists identified thus far exhibit off-target effects on other β-ARs, leading to side effects such as cardiovascular risks (Hainer, 2016), potentially limiting their clinical application in obesity control. Other endogenous factors, such as Irisin (Boström et al, 2012),

[1]Department of Cardiology, Union Hospital, Tongji Medical College, Huazhong University of Science and Technology, Wuhan, China. [2]Clinic Center of Human Genomic Research, Union Hospital, Tongji Medical College, Huazhong University of Science and Technology, Wuhan, China. [3]Liyuan Cardiovascular Center, Tongji Medical College, Huazhong University of Science and Technology, Wuhan, China. [4]Hubei Key Laboratory of Metabolic Abnormalities and Vascular Aging Huazhong University of Science and Technology, 1277 Jiefang Ave., Wuhan, China. [5]Hubei Clinical Research Center for Metabolic and Cardiovascular Disease, Huazhong University of Science and Technology, 1277 Jiefang Ave., Wuhan, China. [6]Division of Gastroenterology, Union Hospital, Tongji Medical College, Huazhong University of Science and Technology, Wuhan, China. [7]These authors contributed equally: Menghao Shi, Yinsong Ye, Lizhi Hu. ✉E-mail: yanglinguh@hust.edu.cn; huangkai1@hust.edu.cn; liangml@hust.edu.cn

fibroblast growth factor 21 (FGF21) (Lynch et al, 2016), thyroid hormone (Weiner et al, 2017), and bone morphogenetic proteins (BMP) (Whittle et al, 2012), which promote adipocyte browning and thermogenesis, may have complex and undesirable biological effects. Consequently, the path towards developing anti-obesity drugs based on adipocyte browning and thermogenic activity remains uncertain.

The Connectivity Map (CMap) is a powerful tool developed by the Broad Institute, which utilizes gene expression profiles to identify active compounds in drugs and their potential therapeutic applications (Lamb et al, 2006). By comparing gene expression profiles of cells treated with different compounds to those of cells under specific gene perturbations, CMap can identify drug candidates that mimic or reverse these perturbation effects (Subramanian et al, 2017). This study aims to utilize the Connectivity Map to explore potential small molecules that promote adipocyte browning and thermogenesis, thereby effectively addressing obesity and its associated metabolic comorbidities.

In this study, we created four gene expression profiles related to adipocyte browning using RNA sequencing (RNA-seq) data from two cold-exposed mouse inguinal WAT (iWAT) samples and two β-AR agonist CL316, 243-stimulated mouse iWAT samples obtained from the GEO database. These profiles were used to query CMap, ultimately identifying isomeranzin (ISM), a phytochemical, as a candidate mimic that promotes thermogenic gene expression. Additionally, we investigated the nature of ISM-induced intracellular signaling and confirmed that ISM activates the thermogenic effect of cultured adipocytes in vitro and adipose tissue in vivo, thereby inducing potent anti-obesity effects.

# Results

## Identification of small molecule compounds promoting adipose browning via CMap analysis

To generate a fat-specific RNA expression profile, we obtained four sets of RNA sequencing (RNA-seq) data from the GEO database: two involved inguinal white adipose tissue (iWAT) of mice under cold exposure, i.e., datasets GSE164219 and GSE133619; the other two were from mice iWAT treated with the β-AR agonist CL316,243, i.e., datasets GSE129083 and GSE98132. We performed differential gene analysis on these datasets (Fig. EV1A–D) and selected the top 200 genes with the highest upregulated expression from each set (Fig. 1B and Table EV1) as thermogenic gene markers for CMap analysis (Fig. 1A).

Through CMap analysis, we identified four datasets of small molecule compounds. We chose drugs with an enrichment score greater than or equal to 95 for Venn diagram analysis, ultimately identifying 15 small molecule compounds potentially related to iWAT thermogenesis (Fig. 1C). Among them, 11 were commercially available compounds (Table EV2). Among these, Mitoxantrone, a known anticancer drug, has been shown to promote adipose browning and combat obesity, validating the reliability of our pre-screened drug results. Next, we evaluated the potential of the remaining 10 candidate compounds to induce thermogenesis in mature primary adipocytes.

For this, we isolated the stromal vascular fraction (SVF) from iWAT and differentiated it into mature adipocytes, then treated these cells with the candidate compounds (Fig. EV1E). For adipocytes treated with compounds, we measured the mRNA and protein abundance of Ucp1 (Fig. 1D,E), as an indicator of thermogenic capacity, and evaluated fatty acid-binding protein 4 (Fabp4) as a marker of adipogenesis (Fig. 1F). Among these assessments, those cells treated with isomeranzin (ISM) showed a significant upregulation of Ucp1 (Fig. 1D,E), with minimal impact on adipogenesis (Fig. 1F). Therefore, we selected ISM for further study to explore its ability to induce thermogenesis in adipocytes.

## Validation of ISM as a Novel Thermogenic Inducer

ISM is a C-isoprenyl coumarin compound (Cheng et al, 2021) (Fig. 2A), widely present in various medicinal plants, mainly in the peels and juices of citrus species (Feger et al, 2006; Han et al, 2022; He et al, 2018; Kuo et al, 2017; Li et al, 2019; Rastogi et al, 1998), and has also been extracted from medicinal plants such as Leonurus japonicus (Xiong et al, 2015) and Murraya exotica (Xu et al, 2016). Existing literature reports that ISM mainly exerts anti-inflammatory effects (Cheng et al, 2021; Xu et al, 2016; Kuo et al, 2017) and has been reported as a potential D2 dopamine receptor antagonist (He et al, 2018). However, we have not seen any reports on the regulation of thermogenic function in adipose tissue by ISM. Therefore, we treated mature mouse primary inguinal adipocytes obtained by inducing the stromal vascular fraction (SVF) of mouse iWAT with ISM to assess its effectiveness in promoting brown-like transformation of mature white adipocytes. In the CCK8 assay for cell viability, we found that ISM at high concentrations of 50 μM did not show toxicity (Fig. 2B). We then found that ISM treatment dose-dependently increased the mRNA abundance of Ucp1 in adipocytes (Fig. 2C) and protein abundance (Fig. 2D,E). Next, mouse adipocytes were treated with the maximum concentration of ISM (50 μM). This treatment led to increased protein (Figs. 2F and EV1F) and mRNA (Fig. 2G) expression of thermogenic genes. Concomitant upregulation was also observed in genes associated with mitochondrial biogenesis and certain genes involved in β-oxidation (Fig. 2G). Consistent with these findings, MitoTracker staining indicated that ISM treatment enhanced mitochondrial abundance in adipocytes (Fig. 2H,I). Consistently, Oil Red O staining revealed that lipid droplets in ISM-treated adipocytes were smaller than those in control cells (Fig. 2J). Quantitative analysis of Oil Red O staining further indicated a reduction in triglyceride (TG) content following ISM treatment (Fig. 2K). In line with these findings, ISM significantly increased the level of non-esterified fatty acids (NEFA) in the culture medium (Fig. 2L) and enhanced the phosphorylation of hormone-sensitive lipase (HSL), although it did not affect the protein level of adipose triglyceride lipase (ATGL) (Fig. 2M,N). Moreover, ISM potentiated CL316243-stimulated lipolysis (Fig. 2L–N), indicating that ISM promotes lipolysis in adipocytes. We next examined the oxygen consumption rate (OCR) and found that, compared with control group, ISM treatment significantly upregulated basal respiration, uncoupled respiration, maximum respiration, and spare respiratory capacity (Figs. 2O and EV1G). Next, we investigated the effect of ISM on the formation of mouse primary inguinal adipocytes. We added ISM during the process of inducing the SVF of mouse iWAT to obtain mature mouse primary inguinal adipocytes (Fig. EV1H). qRT-PCR results showed that ISM had no effect on the expression of common adipocyte markers Fabp4, Peroxisome proliferator-activated receptor γ (Pparγ), and CCAAT/enhancer binding protein α (C/Ebpα) at

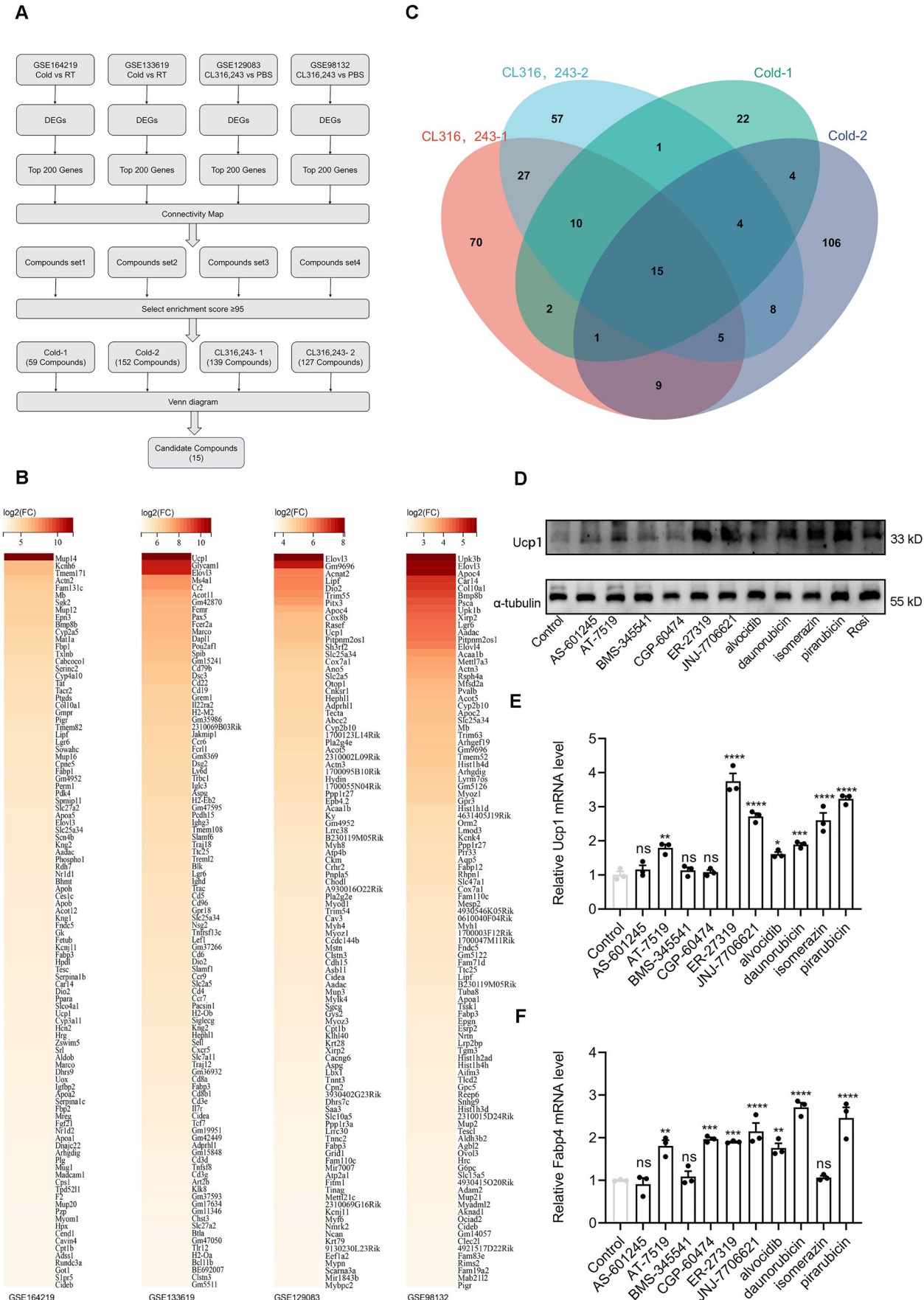

**Figure 1.   Identification of small molecule compounds promoting adipose browning using CMap.**

(A) Workflow diagram illustrating the process of obtaining thermogenic gene expression profiles associated with browning promotion from RNA-seq data in the GEO database, followed by the use of the CMap database to screen for small molecule compounds that promote browning. (B) Heatmap of the top 100 upregulated genes from four RNA-seq datasets. The remaining 100 genes are listed in Table EV1. (C) Venn diagram of the four small molecule compound sets identified. (D) Representative immunoblot of Ucp1 protein in primary inguinal adipocytes treated with ten candidate small molecule compounds. Rosiglitazone is used as a positive control for adipose browning promotion. (E) Relative mRNA expression levels of Ucp1 in primary inguinal adipocytes treated with ten candidate small molecule compounds ($n = 3$). Control vs. AS-601245, $p = 0.9610$; Control vs. AT-7519, $p = 0.0015$; Control vs. BMS-345541, $p = 0.9888$; Control vs. CGP-60474, $p = 0.9994$; Control vs. ER-27319, $p < 0.0001$; Control vs. JNJ-7706621, $p < 0.0001$; Control vs. Alvocidib, $p = 0.0170$; Control vs. Daunorubicin, $p = 0.0004$; Control vs. Isomerazin, $p < 0.0001$; Control vs. Pirarubicin, $p < 0.0001$. (F) Relative mRNA expression levels of Fabp4 in primary inguinal adipocytes treated with ten candidate small molecule compounds ($n = 3$). Control vs. AS-601245, $p = 0.9974$; Control vs. AT-7519, $p = 0.0017$; Control vs. BMS-345541, $p = 0.9975$; Control vs. CGP-60474, $p = 0.0002$; Control vs. ER-27319, $p = 0.0005$; Control vs. JNJ-7706621, $p < 0.0001$; Control vs. Alvocidib, $p = 0.0032$; Control vs. Daunorubicin, $p < 0.0001$; Control vs. Isomerazin, $p = 0.9995$; Control vs. Pirarubicin, $p < 0.0001$. $n$ presents biological replicates. Data are presented as mean ± SEM. Statistical test: one-way ANOVA followed by Dunnett's multiple comparisons test. Statistical significance is indicated as follows: *$p < 0.05$, **$p < 0.01$, ***$p < 0.001$, ****$p < 0.0001$, and ns indicates no significant difference. Source data are available online for this figure.

different concentration gradients (Fig. EV1I). Immunoblot results showed that the protein expression levels of these genes were consistent with the mRNA levels (Figs. 2P and EV1J).

Finally, we validated these findings in human mesenchymal stem cell (HMSC)-derived adipocytes. Consistent with the results in mouse adipocytes, the CCK-8 assay for cell viability indicated that ISM at a high concentration of 50 μM did not exhibit cytotoxicity (Fig. EV2A). We treated HMSC-derived adipocytes with different concentrations of ISM (Fig. EV2B). The results aligned with those from mouse adipocytes: ISM treatment dose-dependently increased both mRNA and protein levels of Ucp1 in human adipocytes (Fig. EV2C–E), without affecting the expression of common adipogenic markers such as Fabp4, Pparγ, and C/Ebpα (Fig. EV2C–E). Similarly, ISM upregulated the expression of thermogenic genes (Fig. EV2F–H) and increased mitochondrial abundance (Fig. EV2I,J). Consistent with enhanced thermogenesis, ISM resulted in smaller lipid droplets and reduced TG content (Fig. EV2K,L), and promoted lipolysis in HMSC-derived adipocytes (Fig. EV2M–O). Moreover, compared with the control group, ISM treatment significantly enhanced basal respiration, uncoupled respiration, maximal respiration, and spare respiratory capacity (Fig. EV2P,Q).

## ISM prevents obesity and improves obesity-related metabolic dysfunction

To assess the pharmacological potential of ISM in vivo, we administered a high-fat diet (HFD) to 8-week-old male C57/BJ mice for 13 weeks, and gave ISM treatment during feeding, according to the results of our previous in vitro experiments, we found that ISM can play a role in promoting adipose browning under the concentration gradient of 2 μM, 10 μM, and 50 μM. Considering the need for long-term dosing treatment of mice in the future, for the sake of biological safety, we chose an intermediate concentration of 10 μM as our reference for estimating the dose administered in vivo. Based on the liquid intake of the mouse was about 5 mL/kg, we estimated the concentration of the mouse administered in vivo to 13 mg/kg. Considering the drug loss and drug metabolism distribution, we finally adopted the administration method of 15 mg/kg, once a day, intraperitoneal injection, and the mice injected with DMSO as the control. Compared to the control group, ISM-treated mice showed slower weight gain (Fig. 3A,B), reduced iWAT and epididymal white adipose tissue

(eWAT), but no change in BAT (Fig. 3C,D). To investigate the therapeutic effects of ISM on metabolic diseases, we measured plasma lipid levels in each group. ISM-treated mice had significantly lower concentrations of TG, TC, LDL-C, and HDL-C compared to controls (Fig. 3E). Furthermore, ISM-treated HFD mice had smaller liver weights (Fig. 3C), less steatosis (Fig. EV3A), and fewer lipid droplets in liver sections as indicated by HE staining (Fig. EV3B), suggesting that ISM treatment improves lipid metabolism and obesity-related non-alcoholic fatty liver disease in HFD mice. Insulin resistance is highly correlated with obesity progression. To assess whether ISM improves insulin sensitivity in HFD mice, we conducted GTT and ITT tests. The results showed significant improvements in glucose tolerance and insulin sensitivity in ISM-treated mice (Fig. 3F–I).

Since there were no significant differences in food intake (Fig. 3J) and activity levels (Fig. 3K) among the groups, we hypothesized that weight loss might be due to increased energy expenditure in ISM-treated mice. To test this, we used metabolic cages to monitor gas exchange. ISM-treated HFD mice showed higher oxygen consumption ($VO_2$) (Fig. 3L,M) and carbon dioxide production ($VCO_2$) (Fig. EV3C,D) compared to control HFD mice, without an increase in respiratory exchange ratio (RER) (Fig. 3N,O). These findings support the role of ISM in preventing obesity by increasing energy expenditure. Similarly, when exposed to low temperatures (4 °C), ISM-treated mice maintained body temperature better than control mice, directly indicating enhanced thermogenesis (Fig. EV3E). To eliminate potential confounding effects of body weight differences on metabolic assessments, we analyzed metabolic parameters in mice after 4 weeks of HFD feeding—a time point at which no significant difference in body weight had yet emerged between groups. Compared with the control group, although body weight remained comparable at this stage, ISM-treated mice already exhibited significantly increased $VO_2$ and $VCO_2$, despite no significant change in the respiratory exchange ratio (RER). These results strongly suggest that the increase in energy expenditure is the cause rather than a consequence of reduced body weight (Fig. EV3F–K). Changes in basal energy expenditure can be influenced by multiple factors. To directly evaluate the enhancement of non-shivering thermogenesis, we treated mice with either ISM or vehicle for 4 weeks and then measured changes in metabolic parameters ($VO_2$ and $VCO_2$) following injection of the β3-adrenergic receptor agonist CL316243. We observed that ISM-treated mice showed significantly elevated $VO_2$ and $VCO_2$ in response to CL316243 stimulation compared to vehicle-treated controls,

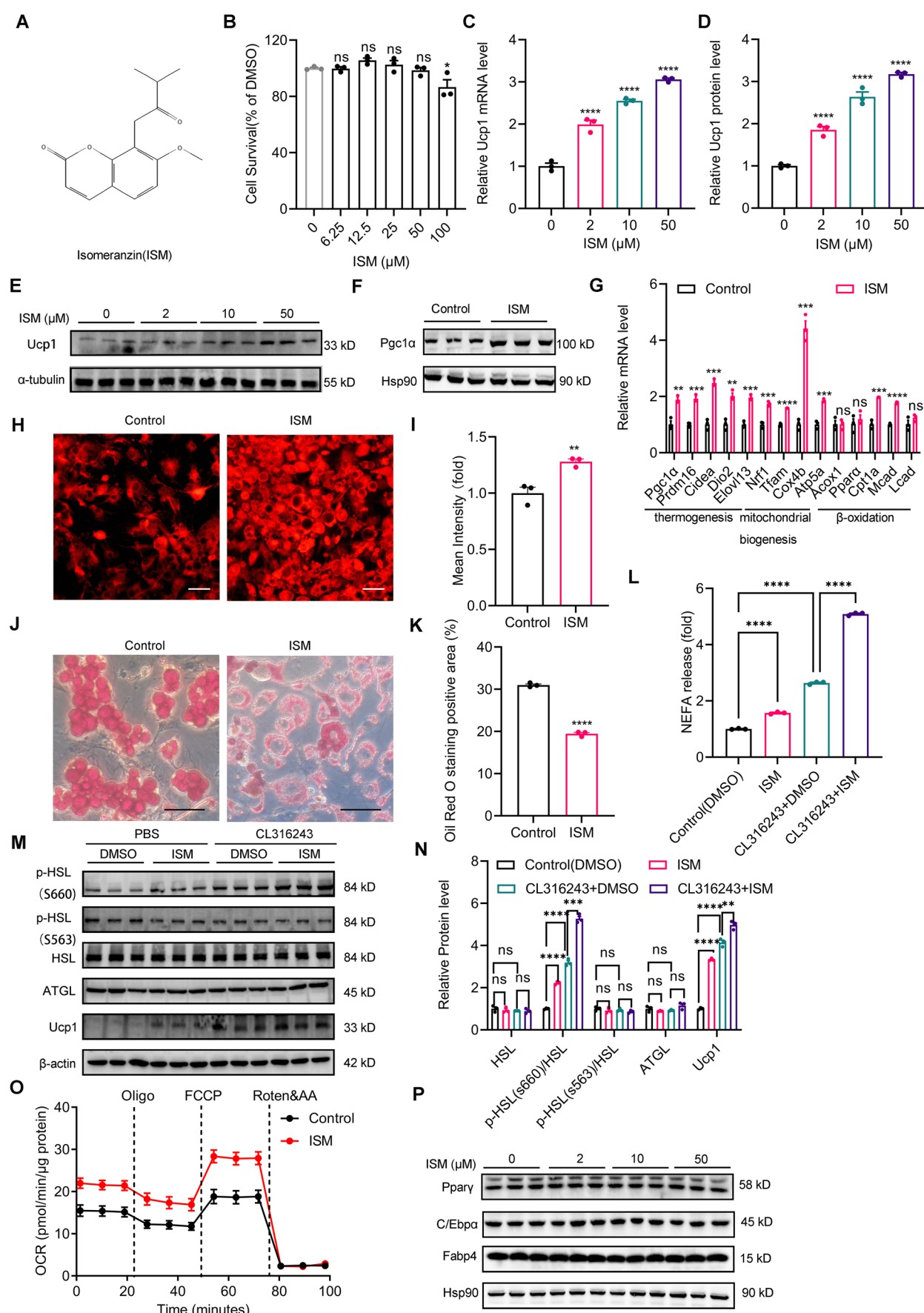

◄ **Figure 2.   Validation of ISM as a Novel Thermogenic Inducer.**

(A) Chemical structure of Isomeranzin (ISM). (B) Cell viability assessed by CCK8 assay in primary inguinal adipocytes treated with varying concentrations of ISM ($n = 3$). Statistical test: one-way ANOVA followed by Dunnett's multiple comparisons test. 0 vs. 6.25, $p > 0.9999$; 0 vs. 12.5, $p = 0.4710$; 0 vs. 25, $p = 0.9338$; 0 vs. 50, $p = 0.9938$; 0 vs. 100, $p = 0.0166$. (C) Relative mRNA expression levels of Ucp1 in mature primary inguinal adipocytes treated with varying concentrations of ISM ($n = 3$). Statistical test: one-way ANOVA followed by Dunnett's multiple comparisons test. 0 vs. 2, $p < 0.0001$; 0 vs. 10, $p < 0.0001$; 0 vs. 50, $p < 0.0001$. (D) Quantitative analysis of relative protein expression levels of Ucp1 in mature primary inguinal adipocytes treated with varying concentrations of ISM ($n = 3$). Statistical test: one-way ANOVA followed by Dunnett's multiple comparisons test. 0 vs. 2, $p < 0.0001$; 0 vs. 10, $p < 0.0001$; 0 vs. 50, $p < 0.0001$. (E) Representative immunoblot images of Ucp1 in mature primary inguinal adipocytes treated with varying concentrations of ISM. (F) Representative immunoblot images of Pgc1α in mature primary inguinal adipocytes from control and ISM (50 μM) treated groups. (G) Relative mRNA expression levels of thermogenesis, mitochondrial biogenesis and β-oxidation related genes in mature primary inguinal adipocytes from control and ISM (50 μM) treated groups ($n = 3$). Statistical test: Unpaired Student's two-tailed t-tests. Pgc1α, $p = 0.0033$; Prdm16, $p = 0.0004$; Cidea, $p = 0.0003$; Dio2, $p = 0.0030$; Elovl13, $p = 0.0003$; Nrf1, $p = 0.0008$; Tfam, $p < 0.0001$; Cox4b, $p = 0.0003$; Atp5a, $p = 0.0006$; Acox1, $p = 0.9982$; Pparα, $p = 0.4932$; Cpt1a, $p = 0.0004$; Mcad, $p < 0.0001$; Lcad, $p = 0.2094$. (H) Representative MitoTracker staining images of mature primary inguinal adipocytes from control and ISM (50 μM) treated groups; scale bar = 50 μm. (I) Quantitative analysis of MitoTracker staining images in mature primary inguinal adipocytes from control and ISM (50 μM) treated groups ($n = 3$). Statistical test: Unpaired Student's two-tailed t-tests. $p = 0.0089$. (J) Representative Oil Red O staining images of mature primary inguinal adipocytes from control and ISM (50 μM) treated groups; scale bar = 50 μm. (K) Quantitative analysis of Oil Red O staining images in mature primary inguinal adipocytes from control and ISM (50 μM) treated groups ($n = 3$). Statistical test: Unpaired Student's two-tailed t-tests. $p < 0.0001$. After 24 h of ISM treatment, the cells were stimulated with 1 μM CL316243 for an additional 24 h. (L) NEFA levels in the mature primary inguinal adipocytes culture medium from different groups ($n = 3$). Statistical test: one-way ANOVA followed by Tukey's multiple comparisons. Control (DMSO) vs. ISM, $p < 0.0001$; Control (DMSO) vs. CL316243 + DMSO, $p < 0.0001$; CL316243 + DMSO vs. CL316243 + ISM, $p < 0.0001$. (M) Representative immunoblot images of p-HSL (Ser660), p-HSL(Ser563), HSL, ATGL and Ucp1 in mature primary inguinal adipocytes from different groups. (N) Quantitative analysis of relative protein expression levels of p-HSL (Ser660), p-HSL (Ser563), HSL, ATGL and Ucp1 in mature primary inguinal adipocytes from different groups ($n = 3$). Statistical test: one-way ANOVA followed by Tukey's multiple comparisons. HSL: Control(DMSO) vs. ISM, $p = 0.5792$; Control(DMSO) vs. CL316243 + DMSO, $p = 0.3191$; CL316243 + DMSO vs. CL316243 + ISM, $p = 0.8422$; p-HSL (S660)/HSL: Control (DMSO) vs. ISM, $p < 0.0001$; Control (DMSO) vs. CL316243 + DMSO, $p < 0.0001$; CL316243 + DMSO vs. CL316243 + ISM, $p = 0.0001$; p-HSL (S563)/HSL: Control (DMSO) vs. ISM, $p = 0.4363$; Control (DMSO) vs. CL316243 + DMSO, $p = 0.4082$; CL316243 + DMSO vs. CL316243 + ISM, $p = 0.2552$; ATGL: Control (DMSO) vs. ISM, $p = 0.2511$; Control (DMSO) vs. CL316243 + DMSO, $p = 0.4175$; CL316243 + DMSO vs. CL316243 + ISM, $p = 0.0872$; Ucp1: Control (DMSO) vs. ISM, $p < 0.0001$; Control (DMSO) vs. CL316243 + DMSO, $p < 0.0001$; CL316243 + DMSO vs. CL316243 + ISM, $p = 0.0065$. (O) Oxygen consumption rates (OCR) in mature primary inguinal adipocytes from control and ISM (50 μM) treated groups ($n = 3$). (P) Representative immunoblot images of Pparγ, C/Ebpα, and Fabp4 in mature primary inguinal adipocytes treated with varying concentrations of ISM. $n$ presents biological replicates. Data are presented as mean ± SEM. Statistical significance is indicated as follows: *$p < 0.05$, **$p < 0.01$, ***$p < 0.001$, ****$p < 0.0001$, and ns indicates no significant difference. Source data are available online for this figure.

providing direct evidence that ISM promotes non-shivering thermogenesis in vivo (Figs. 3P–S and EV3L,M).

Considering biosafety, we measured plasma transaminase and creatinine (CREA) levels. Compared with SCD-fed mice, HFD-fed mice exhibited significantly elevated ALT and AST levels, while ISM treatment alleviated this HFD-induced increase in transaminases. However, ISM did not significantly affect ALT or AST levels in SCD-fed mice (Fig. EV3N,O), suggesting no apparent hepatotoxicity. Furthermore, ISM treatment had minimal impact on CREA levels in both HFD- and SCD-fed mice (Fig. EV3P), indicating no signs of nephrotoxicity.

Similar to the results in HFD-fed mice, ISM-treated SCD-fed mice also displayed reduced body weight and white adipose tissue (WAT) mass, but maintained normal brown adipose tissue (BAT) mass compared to vehicle-treated controls (Fig. EV4A–D). ISM administration also improved plasma lipid profiles, glucose tolerance, and insulin sensitivity in SCD-fed mice (Fig. EV4E–I). Likewise, ISM treatment induced similar—though less pronounced—hepatic improvements in SCD-fed mice as those observed in HFD-fed mice (Fig. EV4J, K). Additionally, ISM enhanced energy expenditure without affecting food intake or physical activity (Fig. EV4L–T). Consistent with this, after four weeks of ISM treatment—before any significant difference in body weight emerged—ISM-treated mice already exhibited elevated metabolic rates (Fig. EV4U–Z).

## ISM promotes browning of subcutaneous white adipose tissue

To further elucidate the mechanism by which ISM promotes thermogenesis, we harvested iWAT, eWAT, and BAT from HFD-fed mice following ISM treatment for histological and gene expression analyses. Consistent with gross observations, ISM treatment reduced adipocyte size in iWAT and eWAT, but not in BAT (Fig. 4A,B). Notably, H&E staining revealed a multi-locular phenotype in iWAT, indicative of white adipose tissue browning. At the molecular level, ISM treatment did not significantly alter the expression of thermogenic genes in BAT (Figs. 4C,E and EV5A) or eWAT (Figs. 4D,F and EV5B), despite the morphological changes in the latter. We hypothesize that the reduction in eWAT adipocyte size may result from enhanced systemic energy expenditure, thereby reducing energy storage. In stark contrast, ISM treatment significantly upregulated the expression of thermogenic genes, but not adipogenic genes, at both the mRNA and protein levels specifically in iWAT (Figs. 4G,H and EV5C).

Similarly, in SCD mice, ISM treatment enhanced the expression of thermogenic genes in iWAT without affecting those in BAT and eWAT (Fig. EV5D–K). To further validate our hypothesis, we performed Ucp1 immunohistochemical staining on three types of adipose tissue. The results demonstrated that ISM treatment significantly upregulated Ucp1 expression in iWAT of both HFD-fed and SCD-fed mice (Fig. 4I,J), while it had no notable effect on Ucp1 protein levels in BAT or eWAT (Fig. EV5O,P). These findings suggest that the thermogenic effect of ISM is primarily mediated by promoting the browning of subcutaneous white adipose tissue rather than by activating BAT. To further corroborate this conclusion, we conducted experiments in primary brown adipocytes derived from mice. Similar to the treatment in primary subcutaneous adipocytes, we incubated brown adipocytes with ISM at various concentrations. Western blot analysis revealed that none of the tested concentrations of ISM significantly altered Ucp1 expression levels in brown adipocytes (Fig. EV5Q–S), consistent with in vivo observations. In summary, ISM promotes

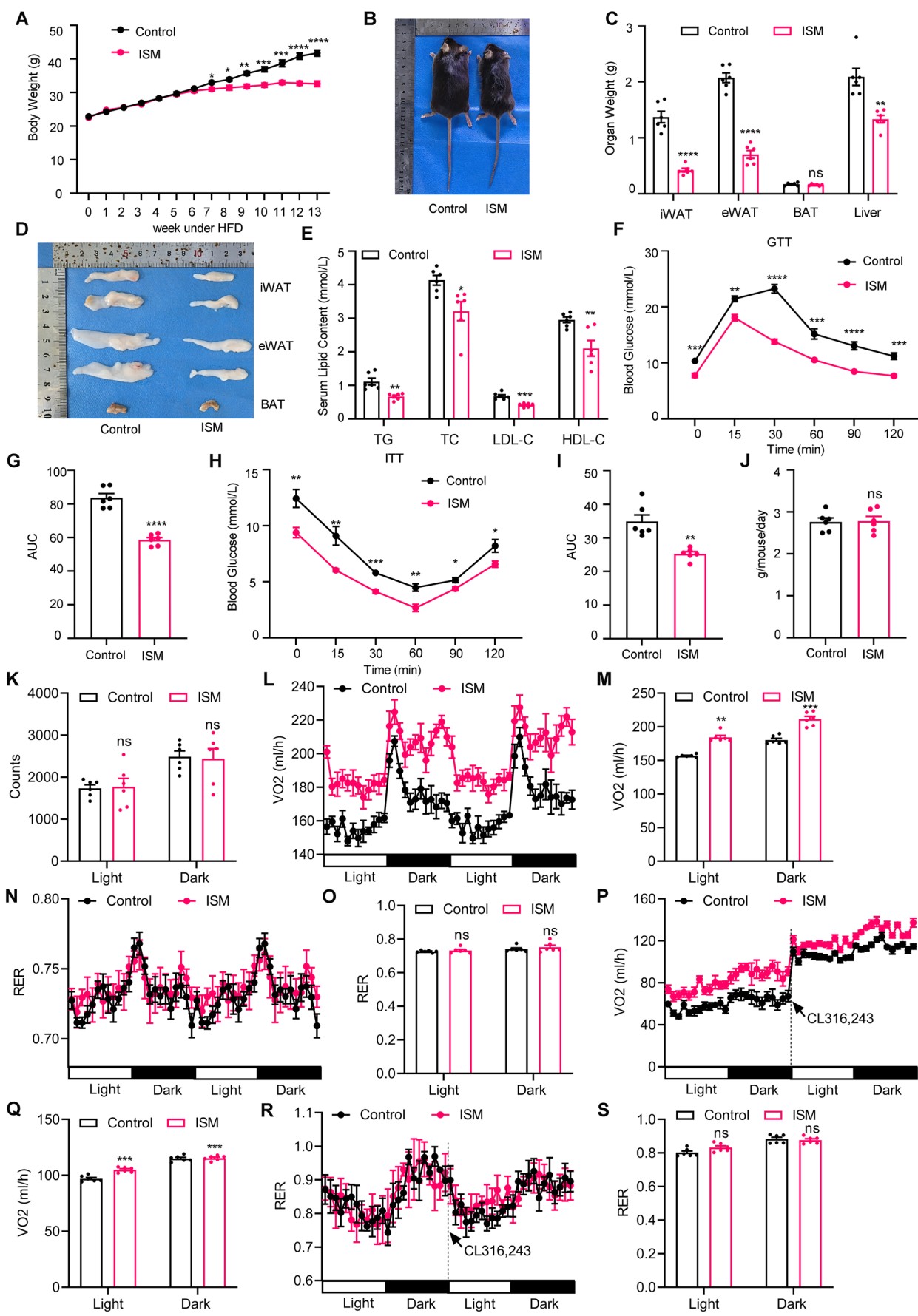

**Figure 3. ISM protects against obesity induced by high-fat diet and improves obesity-related metabolic dysfunction through promoting thermogenesis in mice.**

Mice were intraperitoneally injected with either vehicle or ISM and fed a high-fat diet for 13 weeks ($n = 6$). $n$ presents biological replicates. (A) Body weight of mice in different groups after high-fat diet feeding. 7w, $p = 0.0132$; 8w, $p = 0.0164$; 9w, $p = 0.0012$; 10w, $p = 0.0009$; 11w, $p = 0.0006$; 12w, $p < 0.0001$; 13w, $p < 0.0001$. (B) Representative gross images of mice before sacrifice. (C) Weights of visceral organs (inguinal subcutaneous adipose tissue, epididymal visceral adipose tissue, brown adipose tissue, and liver) in mice. iWAT, $p < 0.0001$; eWAT, $p < 0.0001$; BAT, $p = 0.2879$; Liver, $p = 0.0010$. (D) Representative gross images of adipose tissues (iWAT, eWAT, and BAT). (E) Serum lipid levels (triglycerides, total cholesterol, low-density lipoprotein cholesterol, high-density lipoprotein cholesterol) after 13 weeks of high-fat diet feeding. TG, $p = 0.0021$; TC, $p = 0.016$; LDL-C, $p = 0.0002$; HDL-C, $p = 0.0063$. (F) Glucose tolerance test (GTT) performed after 11 weeks of high-fat diet. 0 min, $p = 0.0008$; 15 min, $p = 0.0012$; 30 min, $p < 0.0001$; 60 min, $p < 0.0001$; 90 min, $p < 0.0001$; 120 min, $p = 0.0002$. (G) Quantitative analysis of GTT. $p < 0.0001$. (H) Insulin tolerance test (ITT) performed after 12 weeks of high-fat diet. 0 min, $p = 0.0078$; 15 min, $p = 0.0047$; 30 min, 0.0002; 60 min, $p = 0.0028$; 90 min, $p = 0.0250$; 120 min, $p = 0.0275$. (I) Quantitative analysis of ITT. $p = 0.0010$. (J) Food intake of HFD mice. $p = 0.9115$. (K) Average physical activity of HFD mice. Light, $p = 0.8786$; Dark, $p = 0.8613$. (L) $VO_2$ of HFD mice. (M) Quantitative analysis of $VO_2$ in (L). $VO_2$-Light, $p = 0.003$; $VO_2$-Dark, $p < 0.001$. (N) Respiratory exchange ratio (RER) of HFD mice. (O) Quantitative analysis of RER in (N). RER-Light, $p = 0.4686$; RER-Dark, $p = 0.4348$. Eight-week-old male mice maintained on a standard chow diet were intraperitoneally injected with vehicle or ISM (15 mg/kg) daily for 4 weeks, followed by administration of CL316243 (1 mg/kg) ($n = 6$). $n$ presents biological replicates. (P) $VO_2$ of mice treated with CL316243. (Q) Quantitative analysis of $VO_2$ in mice after CL316243 treatment. $VO_2$-Light, $p < 0.001$; $VO_2$-Dark, $p < 0.001$. (R). RER of mice treated with CL316243. (S) Quantitative analysis of RER in mice after CL316243 treatment. RER-Light, $p = 0.0567$; RER-Dark, $p = 0.6206$. Data are presented as mean ± SEM. Statistical test: The energy metabolism data ($VO_2$) were analyzed by ANCOVA, with body weight as a covariate. The others: Unpaired Student's two-tailed t-tests. Statistical significance is indicated as follows: $*p < 0.05$, $**p < 0.01$, $***p < 0.001$, $****p < 0.0001$, and ns indicates no significant difference. Source data are available online for this figure.

thermogenesis by inducing browning of subcutaneous white adipose tissue.

## ISM-mediated browning depends on the activation of AMPK signaling

Next, we aimed to explore the signaling pathways activated during ISM treatment. We examined the effects of ISM on several common thermogenic-related signaling pathways. Compared to the control, ISM treatment increased the phosphorylation of AMPK (p-AMPK-Thr172), p38 (p-P38-Thr180/Tyr182), and PKA substrates in primary adipocytes (Fig. 5A,B), but did not activate the PI3K-Akt, JNK, or Erk pathways (Figs. 5C and EV6A). Consistently, ISM also elevated the levels of p-AMPK-Thr172, p-P38-Thr180/Tyr182, and the phosphorylation of PKA substrates in adipocytes derived from human mesenchymal stem cells (HMSC) (Fig. EV6B,C). Together, these results indicate that ISM treatment activates the PKA-p38 MAPK and AMPK signaling pathways.

To investigate the role of AMPK in these findings, we treated primary adipocytes with ISM and the AMPK inhibitor Compound C (CC). The AMPK inhibitor effectively suppressed AMPK phosphorylation and significantly impaired ISM-mediated induction of Ucp1 and Pgc1α (Fig. 5D,G). In contrast, while the PKA inhibitor H89 effectively inhibited the phosphorylation of PKA substrates and p38 MAPK, it did not affect ISM-mediated upregulation of Ucp1 and Pgc1α (Fig. 5E,H). We replicated these results in HMSC-derived adipocytes, which yielded similar outcomes (Fig. EV6D–G). Consistent with its effects on cultured adipocytes, long-term ISM treatment resulted in higher levels of phosphorylated AMPK-Thr172 in iWAT of HFD mice compared to controls (Fig. 5F,I). The same phenomenon was observed in ISM-treated SCD mice (Fig. EV6H,I). Based on these findings, we tentatively conclude that the thermogenic activity of ISM is dependent on AMPK signaling and may be independent of PKA signaling.

## Discovery of direct binding targets of ISM

To identify potential direct targets of ISM, we employed limited proteolysis mass spectrometry (LiP-MS) to detect cellular proteins

directly binding to ISM (Leuenberger et al, 2017). In this workflow, primary adipocyte lysates (three biological replicates per treatment) were incubated with ISM or vehicle (DMSO). Broad-spectrum protease, Proteinase K, was used for limited proteolysis to generate structure-specific protein fragments. These fragments were further digested with trypsin and analyzed using label-free quantitative mass spectrometry (Martinez Molina et al, 2013) (Fig. 6A). To minimize false positives arising from nonspecific interactions between proteins and small molecules, we used both low (10 μM) and high (50 μM) concentrations of ISM. Using LiP-MS, we identified 68 potential target proteins in the low concentration group and 180 in the high concentration group, leading to 26 common potential target proteins when intersecting both results (Fig. 6B and Table EV3). We then performed KEGG pathway enrichment analysis, GO enrichment analysis and protein–protein interaction (PPI) network on these 26 potential targets (Fig. EV7A–D). The KEGG analysis highlighted enriched pathways (Fig. 6C), and we focused on thermogenic pathways relevant to this study. Three genes in this pathway were enriched, encoding for Gnas and Actin proteins. Since previous experiments did not show any change in Actin protein abundance following ISM treatment, we hypothesized that Gnas could be the target through which ISM exerts its browning effect in adipocytes.

Notably, we observed that Gnas mRNA (Fig. EV7E) and protein levels (Figs. 6D and EV7F) were elevated in iWAT of cold-exposed mice compared to thermoneutral controls. Additionally, ISM treatment increased Gnas protein abundance in iWAT of HFD mice (Figs. 6E and EV7G), while Gnas mRNA levels remained unchanged (Fig. EV7H). Similar results were observed in iWAT of SCD mice (Fig. EV7I–K) and in ISM-treated primary mature adipocytes (Fig. EV7L,M). Collectively, these findings suggest Gnas as a potential ISM target.

To corroborate this hypothesis, we performed molecular docking of ISM onto Gnas using Autodock. We identified a potential binding pocket in Gnas, surrounded by Lys300, Tyr163, Gln176, and Asp173 residues, where ISM forms hydrogen bonds with these residues (Fig. 6F). The predicted binding energy was $-6.85$ kcal/mol. Additionally, CETSA (cellular thermal shift assay) experiments were conducted to investigate the potential interaction between ISM and Gnas in adipocytes. ISM increased the thermal

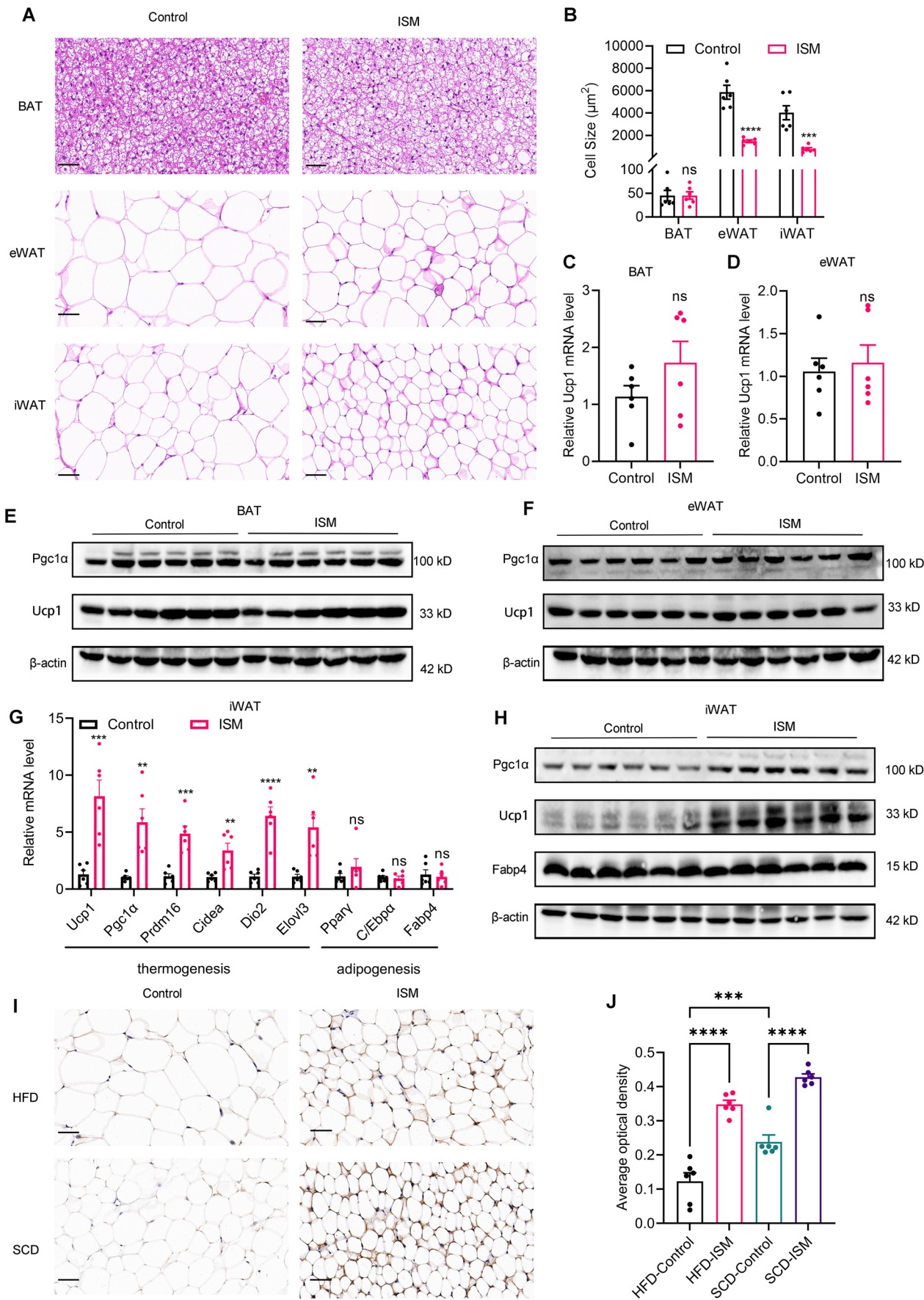

**Figure 4. ISM enhances thermogenesis by promoting browning of subcutaneous white adipose tissue in mice.**

(A) Representative H&E staining images of BAT, eWAT, and iWAT sections; scale bar = 50 μm. (B) Quantitative analysis of adipocyte size in BAT, eWAT, and iWAT (*n* = 6). Unpaired Student's two-tailed t-tests. BAT, $p = 0.9780$; eWAT, $p < 0.0001$; iWAT, $p = 0.0004$. (C) Relative mRNA expression levels of Ucp1 in BAT (*n* = 6). Unpaired Student's two-tailed t-tests. $p = 0.1863$. (D) Relative mRNA expression levels of Ucp1 in eWAT (*n* = 6). Unpaired Student's two-tailed t-tests. $p = 0.7055$. (E) Representative immunoblot images of Pgc1α and Ucp1 in BAT. (F) Representative immunoblot images of Pgc1α and Ucp1 in eWAT. (G) Relative mRNA expression levels of thermogenesis and adipogenesis related genes in iWAT (*n* = 6). Unpaired Student's two-tailed t-tests. Ucp1, $p = 0.0008$; Pgc1α, $p = 0.0021$; Prdm16, $p = 0.0003$; Cidea, $p = 0.0051$; Dio2, $p < 0.0001$; Elovl3, $p = 0.0032$; Ppary, $p = 0.3007$; C/Ebpα, $p = 0.6601$; Fabp4, $p = 0.7091$. (H) Representative immunoblot images of Pgc1α, Ucp1, and Fabp4 in iWAT. (I) Representative immunohistochemical staining images of Ucp1 in iWAT sections; scale bar = 50 μm. (J) Quantitative analysis of Ucp1 immunohistochemical staining in iWAT sections (*n* = 6). One-way ANOVA followed by Tukey's multiple comparisons. HFD-Control vs. HFD-ISM, $p < 0.0001$; HFD-Control vs. SCD-Control, $p = 0.0010$; SCD-Control vs. SCD-ISM, $p < 0.0001$. *n* presents biological replicates. Data are presented as mean ± SEM. Statistical significance is indicated as follows: *$p < 0.05$, **$p < 0.01$, ***$p < 0.001$, ****$p < 0.0001$, and ns indicates no significant difference. Source data are available online for this figure.

stability of Gnas in a temperature- and concentration-dependent manner, whereas the β-actin control did not show such effects (Figs. 6G–J and EV7N,O). These results further support a direct interaction between ISM and Gnas, reinforcing our earlier proposition that Actin is unlikely to be the ISM target.

## ISM regulates AMPK via the Gnas-cAMP axis

We next investigated whether ISM promotes adipose browning through its direct target, Gnas. When Gnas was knocked down using siRNA, the ability of ISM to enhance Ucp1 and Pgc1α expression in adipocytes was reduced at both the mRNA (Fig. 7A) and protein levels (Fig. 7B,C). Similarly, the Oil Red O staining indicated that ISM's effect on promoting lipolysis was attenuated (Fig. 7D,E). In the presence of ISM, AMPK phosphorylation levels increased, but this effect was abolished in Gnas-knockdown cells (Fig. 7B,C).

To understand how Gnas regulates AMPK phosphorylation, we reviewed the literature and found that the Gnas gene encodes the stimulatory G protein alpha subunit (Gsα). Gsα is a ubiquitous signaling protein involved in many G protein-coupled receptor (GPCR)-controlled pathways that can activate adenylyl cyclase, catalyzing the conversion of ATP to the secondary messenger cAMP (Jeong and Chung, 2023). Studies have shown that increased cAMP concentration in adipocytes can regulate AMPK activation (Yin et al, 2003). Thus, we hypothesized that ISM might regulate AMPK activation via the Gnas-cAMP axis.

To test this, we assessed intracellular cAMP levels after Gnas knockdown. We found that ISM treatment elevated cAMP levels in adipocytes, but this effect was reduced following Gnas knockdown (Fig. 7F). This result suggests that ISM activation upregulates intracellular cAMP levels. To further validate this finding, we pre-incubated cells with the adenylyl cyclase inhibitor SQ22536 (100 μM) before ISM treatment to inhibit cAMP production. SQ22536 significantly suppressed ISM-induced increases in intracellular cAMP levels (Fig. 7L) and reversed ISM-induced upregulation of Ucp1 and Pgc1α expression (Fig. 7G–I) and AMPK phosphorylation (Fig. 7H,I). Similarly, the changes in lipolysis were no longer observed (Fig. 7J,K). Collectively, these data indicate that ISM promotes adipocyte browning by activating AMPK through the Gnas-cAMP axis.

In conclusion, ISM enhances thermogenic metabolism in mice by upregulating the expression of thermogenic genes such as Pgc1α and Ucp1 through the Gnas-cAMP-AMPK axis. This improvement in thermogenic activity ameliorates obesity and obesity-related metabolic disorders. These findings not only provide mechanistic

insights into the role of ISM in promoting adipose browning but also highlight the Gnas-cAMP-AMPK axis as a potential pathway for developing new compounds to treat metabolic disorders, such as obesity.

## Discussion

In this study, we uncovered a novel role of ISM in the browning of iWAT. Initially, we utilized sequencing data from four GEO databases to establish a gene profile promoting browning, which, in conjunction with the CMap database, led to the identification of ISM as a new potential small molecule enhancer of adipose browning. Further investigations demonstrated that ISM promoted iWAT browning, increased energy expenditure, and consequently alleviated HFD-induced obesity and metabolic disorders. Mechanistically, we found that ISM facilitates adipose browning by activating the AMPK pathway. Using LiP-MS technology, we identified Gnas as a potential target of ISM in this biological process. Our subsequent research revealed that ISM functions through the Gnas-cAMP-AMPK axis to enhance thermogenic gene expression, thereby inducing the browning of white adipose tissue.

The CMap database is well-suited for the development of novel therapeutic drugs by linking input transcriptional gene profiles with gene expression patterns induced by thousands of small molecules (Qu and Rajpal, 2012). Currently, the CMap database is extensively applied in developing small molecule therapeutics for various cancer types, including breast cancer (Li et al, 2022), lung cancer (Kwon et al, 2020) and gastric cancer (Guo et al, 2023). Beyond oncology, it has been used to discover lead compounds for treating systemic diseases like obesity and diabetes (Liu et al, 2015). There has also been research utilizing the CMap database to identify small molecules promoting thermogenesis in adipose tissue to combat obesity (Chen et al, 2021). In contrast, our study deviates in the selection of gene expression profiles. Previous research used an RNA-seq profile of iWAT and BAT post-cold exposure to screen for thermogenic drugs resembling cold exposure gene profiles. In contrast, we employed four distinct RNA-seq datasets, including two from cold exposure and two from β-adrenergic agonist CL316,243-stimulated mouse iWAT, covering the two most common mouse models used to promote browning. Additionally, our study focused on a more specific phenotype. While previous studies selected drugs that promote browning in both iWAT and activate BAT, our research specifically targeted the browning of white adipose tissue. Our findings confirm the thermogenic potential of ISM predicted by CMap, demonstrating the value of

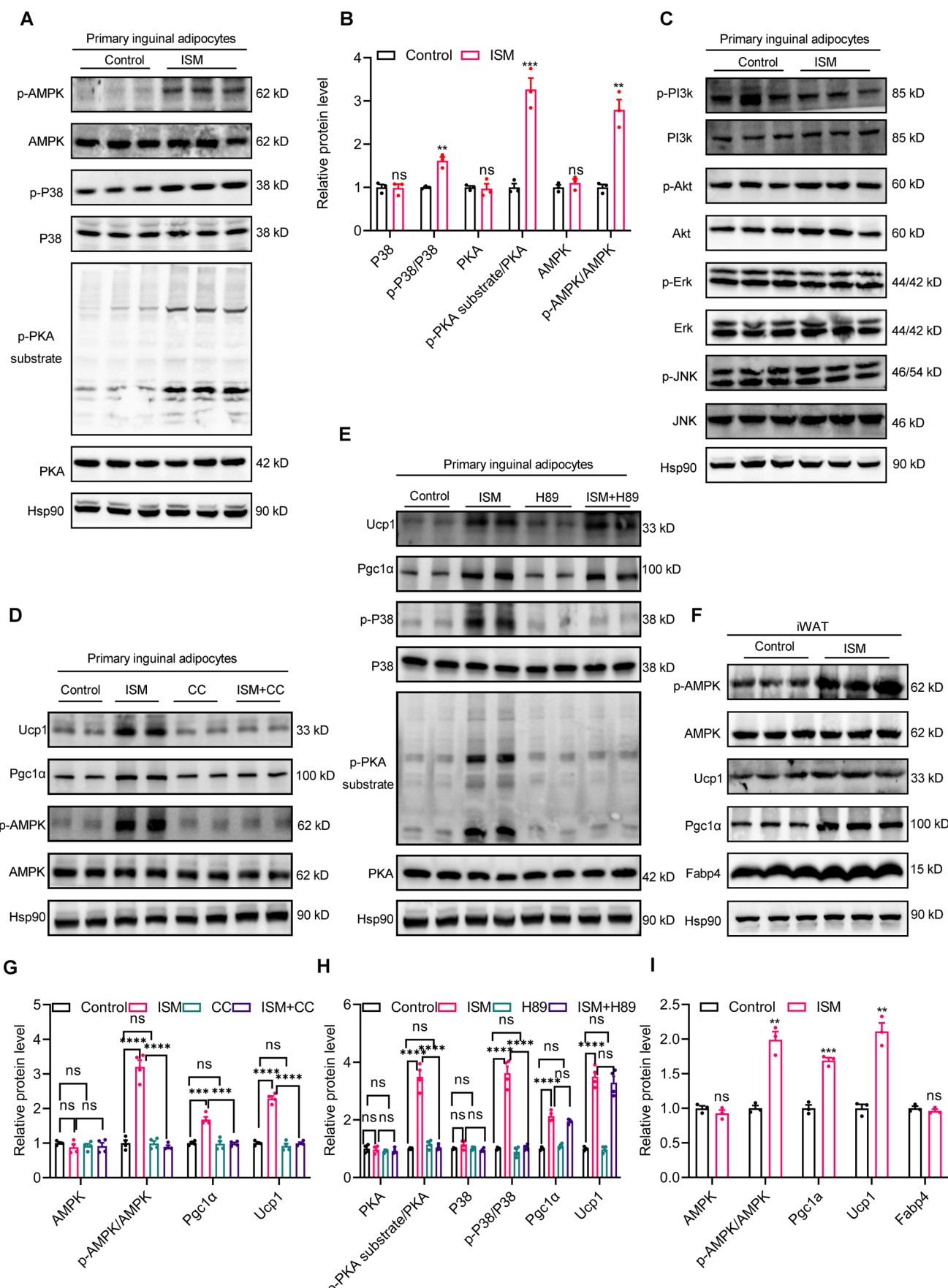

**Figure 5. ISM promotes browning of white adipose tissue via activation of the AMPK pathway.**

(A) Representative immunoblot images of key pathway proteins in mature primary inguinal adipocytes from control and ISM (50 μM) treated groups. (B) Quantitative analysis of relative protein expression levels of the pathway proteins in (A) ($n = 3$). Unpaired Student's two-tailed t-tests. P38, $p = 0.9147$; p-P38/P38, $p = 0.0023$; PKA, $p = 0.8207$; p-PKA substrate/PKA, $p = 0.0012$; AMPK, $p = 0.3805$; p-AMPK/AMPK, $p = 0.0019$. (C) Representative immunoblot images of key pathway proteins in mature primary inguinal adipocytes from control and ISM (50 μM) treated groups. (D) Representative immunoblot images of key pathway proteins in mature primary inguinal adipocytes incubated with or without the AMPK inhibitor CC (10 μM) for 2 h, followed by treatment with ISM or DMSO for 48 h. (E) Representative immunoblot images of key pathway proteins in mature primary inguinal adipocytes incubated with or without the PKA inhibitor H89 (30 μM) for 2 h, followed by treatment with ISM or DMSO for 48 h. (F) Representative immunoblot images of AMPK, p-AMPK, Ucp1, Pgc1α, and Fabp4 in iWAT of high-fat diet-fed mice injected with ISM or vehicle. (G) Quantitative analysis of relative protein expression levels of the pathway proteins in (D) ($n = 3$). One-way ANOVA followed by Tukey's multiple comparisons. AMPK: Control vs. ISM, $p = 0.2458$; Control vs. CC, $p = 0.2676$; ISM vs. ISM + CC, $p = 0.8229$; p-AMPK/AMPK: Control vs. ISM, $p < 0.0001$; Control vs. CC, $p = 0.9168$; ISM vs. ISM + CC, $p < 0.0001$; Pgc1α: Control vs. ISM, $p = 0.0002$; Control vs. CC, $p = 0.8168$; ISM vs. ISM + CC, $p = 0.0002$; Ucp1: Control vs. ISM, $p < 0.0001$; Control vs. CC, $p = 0.3091$; ISM vs. ISM + CC, $p < 0.0001$. (H) Quantitative analysis of relative protein expression levels of the pathway proteins in (E) ($n = 3$). One-way ANOVA followed by Tukey's multiple comparisons. PKA: Control vs. ISM, $p = 0.7547$; Control vs. H89, $p = 0.2997$; ISM vs. ISM + H89, $p = 0.5289$; p-PKA substrate/PKA: Control vs. ISM, $p < 0.0001$; Control vs. H89, $p = 0.0966$; ISM vs. ISM + H89, $p < 0.0001$; P38:Control vs. ISM, $p = 0.1015$; Control vs. H89, $p = 0.8706$; ISM vs. ISM + H89, $p = 0.0541$; p-P38/P38: Control vs. ISM, $p < 0.0001$; Control vs. H89, $p = 0.2699$; ISM vs. ISM + H89, $p < 0.0001$; Pgc1α: Control vs. ISM, $p < 0.0001$; Control vs. H89, $p = 0.0768$; ISM vs. ISM + H89, $p = 0.1121$; Ucp1: Control vs. ISM, $p < 0.0001$; Control vs. H89, $p = 0.6628$; ISM vs. ISM + H89, $p = 0.4793$. (I) Quantitative analysis of relative protein expression levels of the proteins shown in (F) ($n = 3$). Unpaired Student's two-tailed t-tests. AMPK, $p = 0.29114516$; p-AMPK/AMPK, $p = 0.0013$; Pgc1α, $p = 0.0004$; Ucp1, $p = 0.0011$; Fabp4, $p = 0.3707$. $n$ presents biological replicates. Data are presented as mean ± SEM. Statistical significance is indicated as follows: $*p < 0.05$, $**p < 0.01$, $***p < 0.001$, $****p < 0.0001$, and ns indicates no significant difference. Source data are available online for this figure.

this approach in identifying lead compounds for treating multi-system diseases.

ISM is a C-prenylated coumarin compound widely distributed in various medicinal plants, notably abundant in the peels and juices of citrus fruits (Feger et al, 2006; Han et al, 2022; He et al, 2018; Kuo et al, 2017; Li et al, 2019; Rastogi et al, 1998). Additionally, it is found in other traditional medicinal plants such as Leonurus japonicus and Murraya exotica (Xiong et al, 2015; Xu et al, 2016). Research literature indicates that ISM exhibits significant anti-inflammatory properties by downregulating the NF-κB and ERK signaling pathways, which reduces the production of pro-inflammatory cytokines (such as IL-1β, IL-6, and TNF-α) associated with M1 macrophages and regulates their polarization. This underscores its potential therapeutic application in chronic inflammatory diseases (Xu et al, 2016). Notably, ISM is not limited to anti-inflammatory effects; studies also suggest that it may act as an antagonist of D2 dopamine receptors, indicating potential applications in the nervous system (He et al, 2018). This finding opens new avenues for exploring its applications in neuroscience. However, despite the increasing research on ISM across various fields, to date, we have found no reports on its role in regulating thermogenic functions of adipose tissue. This suggests that while ISM has been widely studied, there remain many unresolved questions regarding its diverse bioactivities and potential mechanisms, which necessitate further investigation and validation. Our study is the first to reveal the biological role of ISM in promoting adipose browning, filling a gap in this area of research and providing a scientific basis for its further medicinal development.

Traditional small molecule target identification methods, such as radiolabeling and photoreactive small molecule tagging, primarily rely on using probes or linkers to label small molecules to identify their interactions with candidate target proteins. However, these processes require purification of target proteins and modification of small molecules, which may alter the bioactivity of the molecules or affect their binding specificity. This can limit the accuracy of the studies (Lomenick et al, 2009; Di Michele et al, 2015). In contrast, the LiP-MS technique provides an unbiased approach to identify small molecule target proteins. Its key advantage is that it does not require elaborate recombinant

protein purification or modification of the small molecules (Lomenick et al, 2009; Piazza et al, 2018).The core principle of the LiP-MS method is based on the binding affinity and stoichiometry between the small molecule and its target protein. By observing the reduced protease sensitivity of target proteins upon small molecule binding, altered proteins and peptide fragments can be identified. Initially, the LiP-MS technique (also known as DART) relied on silver-stained SDS-PAGE for target protein identification (Lomenick et al, 2009). However, this method has limitations in high-throughput screening or identifying low-abundance target proteins due to the low resolution and sensitivity of silver staining. To address these limitations, the LiP-MS method was subsequently optimized by integrating high-resolution liquid chromatography-tandem mass spectrometry (LC-MS/MS) and high-throughput quantitative proteomics. This modified LiP-MS method not only improves the accuracy and sensitivity of target protein identification but also enables the identification of specific binding motifs by providing standardized abundance data at the peptide level (Chen et al, 2021). This enhancement has made LiP-MS a powerful tool for studying true drug targets and mechanisms of action. Currently, the LiP-MS technique is being applied in metabolic diseases such as obesity (Chen et al, 2021) and non-alcoholic fatty liver disease (Zhang et al, 2021), as well as in oncological diseases such as non-small cell lung cancer (Wang et al, 2024) and colorectal tumors (Zuo et al, 2021). Using the optimized LiP-MS technique, we identified Gnas as a potential target of the small molecule ISM in the biological process of adipocyte browning. This study also represents the first disclosure of a direct action target for ISM.

Gnas encodes the Gαs protein, which mediates GPCR signaling and is crucial for various physiological processes, including energy metabolism, cell growth, and differentiation (Weinstein et al, 2004). Studies have shown that mutations in Gnas are associated with endocrine tumors, fibrous dysplasia of bone, and McCune-Albright syndrome (Weinstein et al, 2004). Recently, the role of the Gnas gene in tumorigenesis has received widespread attention. Research indicates that mutations in the Gnas gene may be related to the development of endocrine tumors such as colorectal cancer (Nummela et al, 2024), breast cancer (Jin et al, 2019), and

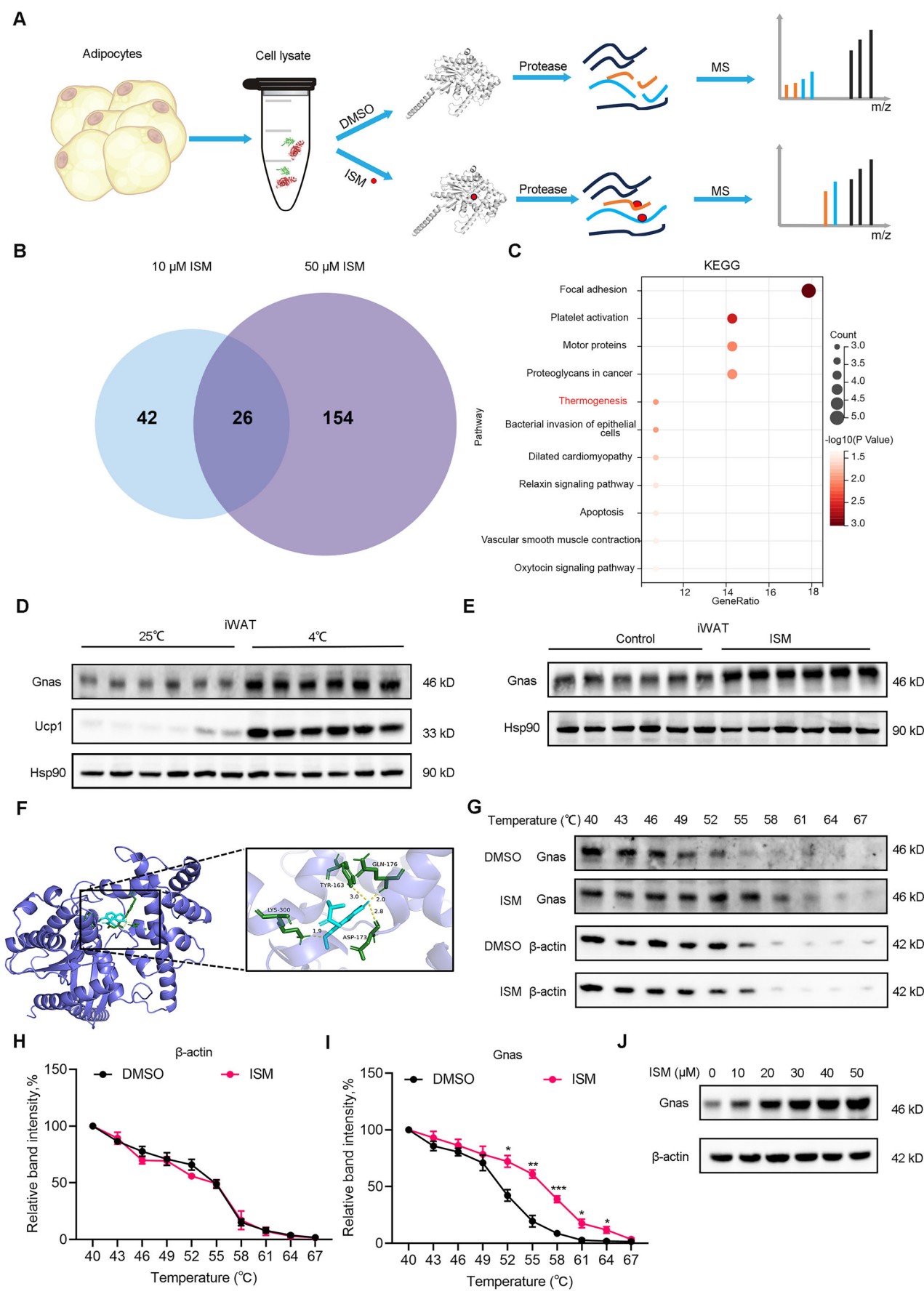

**Figure 6. Discovery of direct binding targets of ISM.**

(A) Flowchart of the LiP-MS assay. Freshly prepared whole-cell lysates were treated with or without ISM, followed by proteinase K (PK) digestion and mass spectrometry analysis. Binding of ISM inhibits PK digestion, leading to differences in mass spectrometry profiles. (B) Venn diagram of potential target proteins identified at different concentrations. Twenty-six common potential target proteins were obtained by intersecting 68 potential target proteins identified in the low concentration (10 μM) ISM treatment group and 180 potential target proteins identified in the high concentration (50 μM) ISM treatment group. (C) KEGG enrichment analysis bubble chart for the 26 potential target proteins. Statistical test: Hypergeometric test and Benjamini-Hochberg procedure. (D) Representative immunoblot images of Gnas and Ucp1 in iWAT from 8-week-old male C57BL/6J mice exposed to 25 °C or 4 °C environments for 3 days. (E) Representative immunoblot images of Gnas in iWAT from mice fed a high-fat diet and injected with ISM. (F) Representative molecular docking image of ISM and Gnas. (G) Representative immunoblot images of thermoprecipitation experiments with temperature dependency. (H) Quantitative analysis of relative grayscale values of β-actin in (G) ($n = 3$). Unpaired Student's two-tailed t-tests. 43, $p = 0.6356$; 46, $p = 0.2132$; 49, $p = 0.8271$; 52, $p = 0.1056$; 55, $p = 0.9272$; 58, $p = 0.8466$; 61, $p = 0.8705$; 64, $p = 0.7553$; 67, $p = 0.7894$. (I) Quantitative analysis of relative grayscale values of Gnas in (G) ($n = 3$). Unpaired Student's two-tailed t-tests. 43, $p = 0.3644$; 46, $p = 0.4101$; 49, $p = 0.4898$; 52, $p = 0.0142$; 55, $p = 0.0026$; 58, $p = 0.0007$; 61, $p = 0.0202$; 64, $p = 0.0326$; 67, $p = 0.1899$. (J) Representative immunoblot images of dose-dependent thermoprecipitation experiments. $n$ presents biological replicates. Data are presented as mean ± SEM. Statistical significance is indicated as follows: *$p < 0.05$, **$p < 0.01$, ***$p < 0.001$, ****$p < 0.0001$, and ns indicates no significant difference. Source data are available online for this figure.

pancreatic cancer (Kawabata et al, 2022). These findings underscore the significance of this gene in cancer research and provide new avenues for targeted therapy. Bone fibrous dysplasia associated with the Gnas gene includes conditions like Albright's hereditary osteodystrophy (AHO), pseudohypoparathyroidism, and progressive osseous heteroplasia (POH) (Yang et al, 2023). Some studies have also suggested that Gnas mutations are related to obesity development, possibly linked to pseudohypoparathyroidism caused by Gnas mutations (Mendes de Oliveira et al, 2021). Research on Gnas gene knockout mice demonstrates that this gene is crucial in regulating lipid and glucose metabolism (Chen et al, 2005). Our study reveals that cold exposure leads to upregulation of Gnas gene expression, which could be closely related to its role in the thermogenesis process mediated by ISM. Given this finding, Gnas may serve as a potential target for obesity treatment, providing significant support for further exploring the roles of ISM and Gnas in metabolic regulation. To capitalize this discovery, we suggest further structural optimization of the lead compound ISM to enhance its efficacy as a therapeutic agent. Additionally, it is worth exploring related compounds with higher binding affinity, optimal pharmacokinetic characteristics, and favorable safety profiles in animals and humans.

In this study, we employed two advanced technological platforms: CMap and LiP-MS. These technologies not only helped identify ISM as a potential therapeutic compound for obesity but also clarified Gnas as the direct protein target of ISM. Our research establishes a new paradigm for drug development in metabolic diseases by combining multiple levels and methods, laying the foundation for the development of more effective therapeutic strategies. These technological approaches demonstrate their strong capability in identifying and validating potential therapeutic compounds and their targets, providing significant guidance for developing treatment strategies for metabolic diseases.

However, this study has several limitations: (1) Although we identified Gnas as the direct target of ISM in adipose browning for the first time and validated its role in vitro, genetic rescue experiments have not yet been conducted at the whole-animal level. Future studies will involve generating adipocyte-specific conditional Gnas knockout mice to verify the necessity and sufficiency of the ISM–Gnas axis in vivo. (2) Although molecular docking simulations suggested hydrogen bond interactions between ISM and Gnas, and functional interactions along with therapeutic outcomes were experimentally confirmed, the precise molecular

details of Gnas activation require further investigation. As a G protein, Gnas cycles between inactive (GDP-bound) and active (GTP-bound) states. ISM binding may stabilize the active GTP-bound conformation, promote nucleotide exchange, or prevent GTP hydrolysis. We plan to employ GTPase activity assays, G-protein conformational sensors, and crystallographic studies to elucidate the exact mechanism of ISM-mediated Gnas activation, and to validate the functional importance of key amino acid residues through site-directed mutagenesis. (3) While the current study focuses on the direct metabolic effects of ISM, we acknowledge that its anti-inflammatory properties may also contribute indirectly. M1 macrophage polarization in adipose tissue secretes pro-inflammatory cytokines that suppress browning. By inhibiting M1 polarization, ISM may create a more favorable microenvironment for browning. However, this potential indirect mechanism was not directly investigated in our study and will be explored in future research. (4) Although this study reveals the important role of ISM in promoting white adipose tissue browning, we must acknowledge its limitations. First, our primary in vivo evidence is derived from mouse models. Differences in fat metabolism, distribution, and responses to cold exposure and β-adrenergic stimulation between mice and humans limit the direct extrapolation of our conclusions to human obesity. However, several points are worth emphasizing, as they strongly support the potential translational value of our findings. Firstly, the core mechanism we investigated—the induction of mitochondrial biogenesis and activation of the adipocyte thermogenic program via the AMPK/Pgc1α signaling pathway—is a relatively evolutionarily conserved process. More importantly, we successfully replicated the pro-browning effect of ISM in primary human adipocytes (HMSCs), providing the most direct experimental evidence for the bioactivity of this compound in human systems. Numerous clinical studies have confirmed that functional beige adipose tissue is present in adults, and its activity is positively correlated with lower body mass index, improved insulin sensitivity, and overall metabolic health. Therefore, the pharmacological activation of human beige fat is regarded as a highly attractive strategy for treating obesity and related metabolic syndromes. In summary, our work not only elucidates the mechanism of action of a novel natural compound, ISM, but also, through its demonstrated effectiveness in human-derived cells, establishes it as a lead compound worthy of further in-depth investigation. Future studies will focus on optimizing the administration regimen of ISM and evaluating its safety and

efficacy in more advanced preclinical models, such as humanized mouse or non-human primate models, ultimately paving the way for developing innovative therapies for human obesity.

# Methods

### Reagents and tools table

| Reagent/Resource | Reference or Source | Identifier or Catalog Number |
|---|---|---|
| **Experimental models** | | |
| Mouse primary adipocytes | Derived from the C57BL/6J mice | |
| Human mesenchymal stem cell | Procell | CP-H202 |
| C57BL/6J mice (*M.musculus*) | Sibeifu Biotechnology | |
| **Antibodies** | | |
| α-tubulin | Proteintech | 11224-1-AP |
| β-actin | Proteintech | 20536-1-AP |
| Hsp90 | Proteintech | 13171-1-AP |
| Ucp1 | Abcam | ab10983 |
| Pgc1α | Abclonal | A12348 |
| p-HSL(S660) | Cell Signaling Technology | 45804 |
| p-HSL(S563) | Cell Signaling Technology | 4139 |
| HSL | Proteintech | 17333-1-AP |
| ATGL | Proteintech | 55190-1-AP |
| PPARγ | Proteintech | 16643-1-AP |
| C/EBPα | Proteintech | 18311-1-AP |
| Fabp4 | Proteintech | 12802-1-AP |
| Pi3k | Cell Signaling Technology | 4292 |
| p-Pi3k | Cell Signaling Technology | 4228 |
| Akt | Cell Signaling Technology | 9272 |
| p-Akt | Cell Signaling Technology | 9271 |
| PKA | Cell Signaling Technology | 4782 |
| p-PKA Substrate | Cell Signaling Technology | 9624 |
| P38 | Proteintech | 14064-1-AP |
| p-P38 | Cell Signaling Technology | 9216 |
| ERK1/2 | Cell Signaling Technology | 4668 |
| p-ERK1/2 | Cell Signaling Technology | 5726 |
| JNK | Cell Signaling Technology | 9252 |
| p-JNK | Cell Signaling Technology | 9255 |
| AMPK | Cell Signaling Technology | 2532 |
| p-AMPK | Cell Signaling Technology | 2535 |
| Gnas | Proteintech | 10150-2-AP |
| **Oligonucleotides and other sequence-based reagents** | | |
| qRT-PCR primers | This study | Table EV4 |
| siRNA | This study | Table EV5 |
| **Chemicals, Enzymes and other reagents** | | |
| Insulin | Sigma-Aldrich | 1342106 |
| IBMX | Sigma-Aldrich | I5879 |
| Dexamethasone | Sigma-Aldrich | D1756 |
| Rosiglitazone | Sigma-Aldrich | R2408 |
| Alvocidib | MedChemExpress | HY-10005 |
| AS-601245 | MedChemExpress | HY-11010 |
| AT-7519 | MedChemExpress | HY-50940 |
| BMS-345541 | MedChemExpress | HY-10519 |
| CGP-60474 | MedChemExpress | HY-11009 |
| Daunorubicin | MedChemExpress | HY-13062 |
| ER-27319 | MedChemExpress | HY-138961 |
| JNJ-7706621 | MedChemExpress | HY-10329 |
| Pirarubicin | MedChemExpress | HY-13725 |
| Isomerazin | ChemFaces | CFN99090 |
| DMSO | Sigma-Aldrich | D2650 |
| CL316243 | Sigma-Aldrich | C5976 |
| Compound C | Selleck | S7840 |
| H 89 2HCl | Selleck | S1582 |
| SQ22536 | Selleck | S8283 |
| DMEM | Gibco | 11995500 |
| FBS | Gibco | A3160901 |
| Human adipose-derived mesenchymal stem cell medium | Procell | CM-H202 |
| **Software** | | |
| GraphPad Prism 8 | https://www.graphpad.com/features | |
| SPSS 26 | https://www.ibm.com/cn-zh/products/spss-statistics | |
| ImageJ | https://imagej.nih.gov/ij/ | |
| Connectivity Map | http://clue.io/cmap | |
| AutoDock 4.2 | http://autodock.scripps.edu/ | |
| PyMOL (Version 2.4.0) | https://www.pymol.org/ | |
| **Other** | | |
| CCK8 kit | Beyotime | C0038 |
| NEFA kit | Beyotime | S0215S |
| ALT kit | Rayto | S03030 |
| AST kit | Rayto | S03040 |
| CREA kit | Rayto | S03076 |
| TG kits | Rayto | S03027 |
| TC kit | Rayto | S03042 |
| HDL-C kit | Rayto | S03025 |
| LDL-C kit | Rayto | S03029 |
| cAMP ELISA kit | Bioswamp | MU31007 |

| Reagent/Resource | Reference or Source | Identifier or Catalog Number |
|---|---|---|
| High-fat diet | Research diet | D12492 |
| Standard chow diet | SLACOM | P1101F-25 |

## Animal models

All animal experiments were approved by the Animal Care and Use Committee of Tongji Medical College, Huazhong University of Science and Technology (4108) and complied with the Guide for the Care and Use of Laboratory Animals published by the National Institutes of Health. C57BL/6J male mice (SPF grade) were obtained from Beijing Sibeifu Biotechnology Co., Ltd. (Beijing, China) and housed in animal facilities at 22–23 °C with 40–60% humidity and a 12-h light/dark cycle. Mice had free access to food and water throughout the study. All mice used in the experiments exhibited normal health status. After acclimatization for 1 or 2 weeks upon entering the facility, all experiments were conducted using 8-week-old male mice. Mice were randomly assigned to four groups: SCD-Control Group: fed standard chow diet (ND, SLACOM, P1101F-25) and intraperitoneally injected with vehicle for 13 weeks; SCD-ISM Treatment Group: fed standard chow diet and intraperitoneally injected with ISM (15 mg/kg) for 13 weeks; HFD-Control Group: fed high-fat diet (60% fat, Research Diet, D12492) and intraperitoneally injected with vehicle for 13 weeks; HFD-ISM Treatment Group: fed high-fat diet and intraperitoneally injected with ISM (15 mg/kg) for 13 weeks. Food intake and body weight were measured once weekly. Metabolic parameter measurements were performed using the Comprehensive Laboratory Animal Monitoring System (CLAMS, Columbus Instruments) at both the 4th week and the endpoint of the feeding period. Glucose tolerance tests (GTT) and insulin tolerance tests (ITT) were performed after continuous feeding for 11 and 12 weeks. At the end of 13 weeks, mice were euthanized under anesthesia, and blood serum and tissues were collected for further analysis. For CL316243 stimulation, 8-week-old male mice maintained on a standard chow diet were intraperitoneally injected with vehicle or ISM (15 mg/kg) daily for 4 weeks, followed by administration of CL316243 (1 mg/kg).

## Cell culture and differentiation

Primary mouse adipocytes were differentiated from stromal vascular fraction (SVF) isolated from the iWAT or BAT of 6-week-old male C57BL/6J mice. Fully confluent cells were induced to differentiate in DMEM containing 10% fetal bovine serum (FBS), 2.5 μM rosiglitazone (Sigma-Aldrich, R2408), and MDI (0.5 mM 3-isobutyl-1-methylxanthine (Sigma-Aldrich, I5879), 2 μM dexamethasone (Sigma-Aldrich, D1756), and 10 μg/mL insulin (Sigma-Aldrich, 1342106)) for 2 days. Cells were then further incubated in DMEM containing 10% FBS and 10 μg/mL insulin, with medium changes every two days for 3–4 times. After adipocyte maturation, cells were incubated with DMSO or small molecule compounds for 2 days. Human adipose-derived mesenchymal stem cells (Procell, Cat: CP-H202) were cultured in complete human adipose-derived mesenchymal stem cell medium (Procell, Cat: CM-H202) until confluence and then induced to differentiate into mature adipocytes with rosiglitazone and MDI for 2 days. Cells were further incubated in DMEM containing 10% FBS, 2.5 μM rosiglitazone, and 10 μg/mL insulin for 2 days. This induction process was repeated 3–4 times until lipid droplets appeared. All cells were maintained at 37 °C in a humidified atmosphere with 5% $CO_2$. Mycoplasma contamination was checked at least once a month for all cell lines.

## Connectivity map (CMap)

Differential expression analysis was performed using the limma package with empirical Bayes moderation of the standard errors. RNA-seq data were from two independent datasets of inguinal white adipose tissue (iWAT) from cold-exposed mice (Li et al, 2021; Choi et al, 2021) and two additional datasets from β-adrenergic agonist CL316, 243-stimulated mouse iWAT (Wang et al, 2019; Hepler et al, 2017), all obtained from the GEO database. For RNA-seq data, the voom transformation was first applied to account for the mean-variance relationship. Moderated t-tests were used to calculate p-values, which were subsequently adjusted using the Benjamini-Hochberg procedure. Genes with an adjusted p-value (Padj) < 0.05 and an absolute log2 fold change |Log2(FC)| > 1 were considered differentially expressed. Specifically, genes with Log2(FC) > 1 were classified as upregulated, and those with Log2(FC) < −1 as down-regulated. This process yielded four distinct sets of differentially expressed genes. We selected the top 200 upregulated genes from each dataset as browning gene signatures for CMap analysis (http://clue.io/cmap). The CMap database employs a pattern-matching algorithm to compare the input gene signatures with gene signatures from over 450,000 chemical perturbations under various backgrounds. Enrichment scores for these perturbations range from +100 to −100, ranking them based on their similarity or difference to the input browning gene signature. Finally, four datasets of small molecule compounds were obtained, and drugs with an enrichment score ≥95 were selected for Venn diagram analysis.

## Cell counting kit-8 (CCK8) cytotoxicity assay

To evaluate drug toxicity, mouse primary adipocytes and human adipose-derived mesenchymal stem cells (HMSC) were seeded at a density of 5000 cells per well in 96-well plates. Cells were then incubated with 0.1% DMSO and varying concentrations of isomeranzin (6.25 μM, 12.5 μM, 25 μM, 50 μM, 100 μM). After 48 h, 10 μL of CCK8 reagent (Beyotime, C0038) was added, and the mixture was incubated for 2 h. The absorbance at 450 nm was measured using a microplate reader (Multiskan FC, Thermo Fisher Scientific, Rockford, IL, United States).

## Quantitative real-time polymerase chain reaction (qRT-PCR)

RNA extraction was performed using TRIzol reagent (Takara, Japan). The isolated RNA was reverse transcribed to cDNA using the PrimeScript™ RT reagent kit (RR036A, Takara Bio). SYBR Green (Bio-Rad, USA) was used for quantitative amplification of the cDNA products on the PRISM 7900 Sequence Detection System (Applied Biosystems, Foster City). All gene expressions were

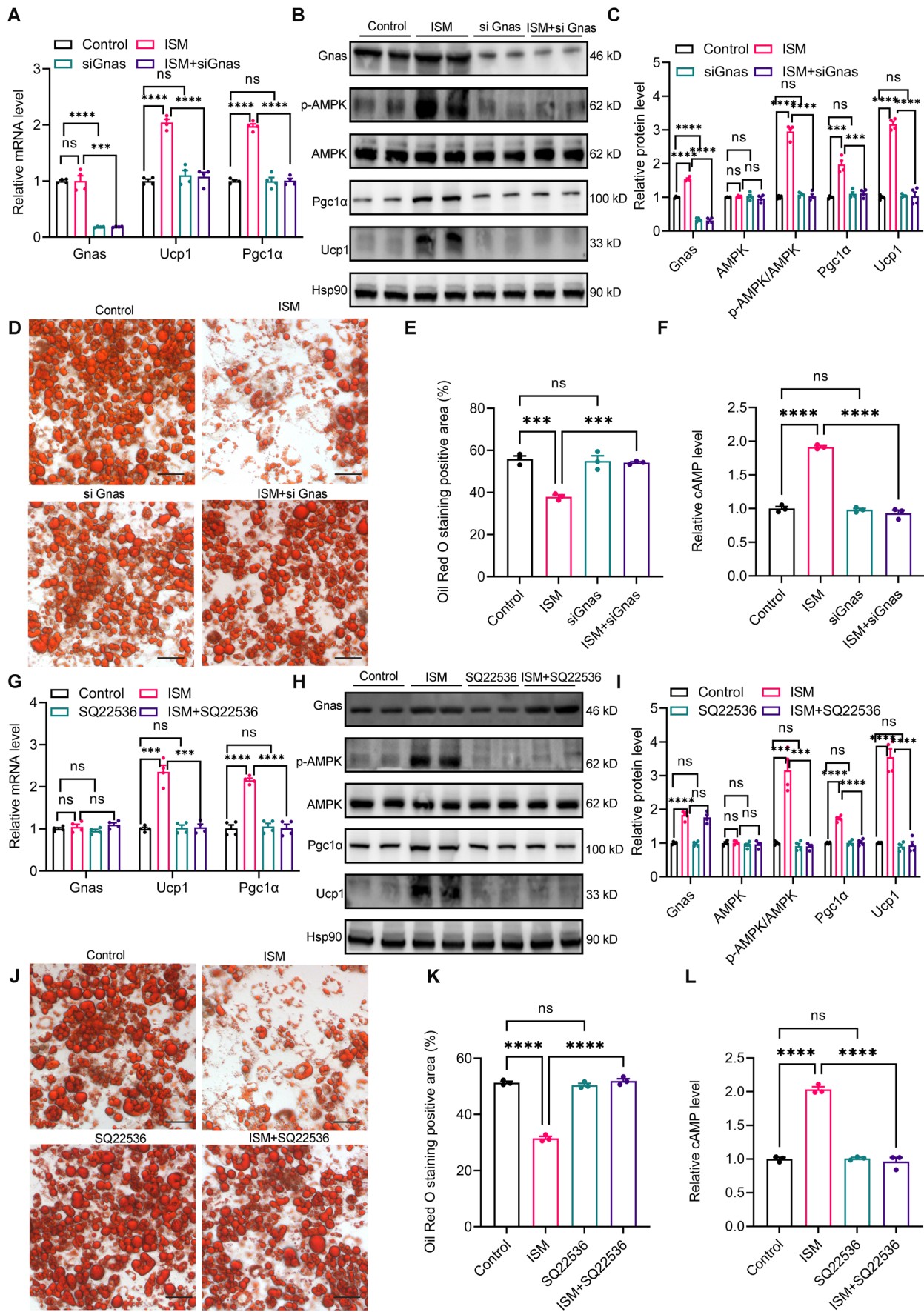

**Figure 7. ISM promotes browning through the Gnas-cAMP-AMPK axis.**

In mature primary inguinal adipocytes treated with siRNA to knock down Gnas, followed by ISM treatment for 48 h: (A) Relative mRNA expression levels of Gnas, Ucp1, and Pgc1α ($n = 3$). Gnas: Control vs. ISM, $p = 0.9667$; Control vs. siGnas, $p < 0.0001$; ISM vs. ISM+siGnas, $p = 0.0002$; Ucp1: Control vs. ISM, $p < 0.0001$; Control vs. siGnas, $p = 0.2905$; ISM vs. ISM+siGnas, $p < 0.0001$; Pgc1α: Control vs. ISM, $p < 0.0001$; Control vs. siGnas, $p = 0.9892$; ISM vs. ISM+siGnas, $p < 0.0001$. (B) Representative immunoblot images of Gnas, AMPK, p-AMPK, Ucp1, and Pgc1α. (C) Quantitative analysis of relative protein expression levels of the proteins shown in (B) ($n = 3$). Gnas: Control vs. ISM, $p < 0.0001$; Control vs. siGnas, $p < 0.0001$; ISM vs. ISM+siGnas, $p < 0.0001$; AMPK: Control vs. ISM, $p = 0.3623$; Control vs. siGnas, $p = 0.7042$; ISM vs. ISM+siGnas, $p = 0.3713$; p-AMPK/AMPK: Control vs. ISM, $p < 0.0001$; Control vs. siGnas, $p < 0.0001$; ISM vs. ISM+siGnas, $p = 0.078$; Pgc1α: Control vs. ISM, $p = 0.0002$; Control vs. siGnas, $p = 0.1323$; ISM vs. ISM+siGnas, $p = 0.0005$; Ucp1: Control vs. ISM, $p < 0.0001$; Control vs. siGnas, $p = 0.3034$; ISM vs. ISM+siGnas, $p < 0.0001$. (D) Representative Oil Red O staining images; scale bar = 50 μm. (E) Quantitative analysis of Oil Red O staining images ($n = 3$). Control vs. ISM, $p = 0.0002$; Control vs. siGnas, $p = 0.9749$; ISM vs. ISM+siGnas, $p = 0.0003$. (F) Relative levels of intracellular cAMP ($n = 3$). Control vs. ISM, $p < 0.0001$; Control vs. siGnas, $p = 0.9715$; ISM vs. ISM+siGnas, $p < 0.0001$. In mature primary inguinal adipocytes pretreated with or without the adenylyl cyclase inhibitor SQ22536 (100 μM) for 1 h, followed by ISM treatment for 48 h: (G) Relative mRNA expression levels of Gnas, Ucp1, and Pgc1α ($n = 3$). Gnas: Control vs. ISM, $p = 0.5182$; Control vs. SQ22536, $p = 0.3181$; ISM vs. ISM + SQ22536, $p = 0.5158$; Ucp1: Control vs. ISM, $p = 0.0002$; Control vs. SQ22536, $p = 0.7577$; ISM vs. ISM + SQ22536, $p = 0.0002$; Pgc1α: Control vs. ISM, $p < 0.0001$; Control vs. SQ22536, $p = 0.6914$; ISM vs. ISM + SQ22536, $p < 0.0001$. (H) Representative immunoblot images of Gnas, AMPK, p-AMPK, Ucp1, and Pgc1α. (I) Quantitative analysis of relative protein expression levels of the proteins shown in (G) ($n = 3$). Gnas: Control vs. ISM, $p < 0.0001$; Control vs. SQ22536, $p = 0.5614$; ISM vs. ISM + SQ22536, $p = 0.5316$; AMPK: Control vs. ISM, $p = 0.7370$; Control vs. SQ22536, $p = 0.3402$; ISM vs. ISM + SQ22536, $p = 0.4129$; p-AMPK/AMPK: Control vs. ISM, $p = 0.0004$; Control vs. SQ22536, $p = 0.3011$; ISM vs. ISM + SQ22536, $p = 0.0003$; Pgc1α: Control vs. ISM, $p < 0.0001$; Control vs. SQ22536, $p = 0.8275$; ISM vs. ISM + SQ22536, $p < 0.0001$; Ucp1: Control vs. ISM, $p < 0.0001$; Control vs. SQ22536, $p = 0.1630$; ISM vs. ISM + SQ22536, $p < 0.0001$. (J) Representative Oil Red O staining images; scale bar = 50 μm. (K) Quantitative analysis of Oil Red O staining images ($n = 3$). Control vs. ISM, $p < 0.0001$; Control vs. SQ22536, $p = 0.7647$; ISM vs. ISM + SQ22536, $p < 0.0001$. (L) Relative levels of intracellular cAMP ($n = 3$). Control vs. ISM, $p < 0.0001$; Control vs. SQ22536, $p = 0.9973$; ISM vs. ISM + SQ22536, $p < 0.0001$. $n$ presents biological replicates. Data are presented as mean ± SEM. Statistical test: one-way ANOVA followed by Tukey's multiple comparisons. Statistical significance is indicated as follows: *$p < 0.05$, **$p < 0.01$, ***$p < 0.001$, ****$p < 0.0001$, and ns indicates no significant difference. Source data are available online for this figure.

quantified relative to 18S rRNA as an internal control. Primer sequences are shown in the Table EV4.

## Western blot

Cells were lysed in RIPA buffer containing 1 mM PMSF and 4 mM phosphatase inhibitors. Ultrasonication was used to promote the release of intracellular proteins. Protein concentration was adjusted using a BCA protein assay kit (Thermo Scientific™, 23227), and equal amounts of protein were separated by SDS-PAGE according to molecular weight and transferred to PVDF membranes. The following primary antibodies were used for immunoblotting: α-tubulin (Proteintech, 11224-1-AP, 1:1000), β-actin (Proteintech, 20536-1-AP, 1:1000), Hsp90 (Proteintech, 13171-1-AP, 1:1000), Ucp1 (Abcam, ab10983, 1:1000), Pgc1α (Abclonal, A12348, 1:1000), p-HSL(S660) (Cell Signaling Technology, 45804, 1:1000), p-HSL (S563) (Cell Signaling Technology,4139, 1:1000), HSL(Proteintech,17333-1-AP, 1:1000), ATGL (Proteintech, 55190-1-AP, 1:1000), PPARγ (Proteintech,16643-1-AP, 1:1000), C/EBPα (Proteintech, 18311-1-AP, 1:1000), Fabp4 (Proteintech, 12802-1-AP, 1:1000), PI3K (Cell Signaling Technology, 4292, 1:1000), p-PI3K (Cell Signaling Technology, 4228, 1:1000), Akt (Cell Signaling Technology, 9272, 1:1000), p-Akt (Cell Signaling Technology, 9271, 1:1000), PKA (Cell Signaling Technology, 4782, 1:1000), p-PKA Substrate (Cell Signaling Technology, 9624, 1:1000), P38 (Proteintech, 14,064-1-AP, 1:1000), p-P38 (Cell Signaling Technology, 9216, 1:1000), ERK1/2 (Cell Signaling Technology, 4668, 1:1000), p-ERK1/2 (Cell Signaling Technology, 5726, 1:1000), JNK (Cell Signaling Technology, 9252, 1:1000), p-JNK (Cell Signaling Technology, 9255, 1:1000), AMPK (Cell Signaling Technology, 2532, 1:1000), p-AMPK (Cell Signaling Technology, 2535, 1:1000), Gnas (Proteintech, 10150-2-AP, 1:1000). The membranes were then probed with appropriate horseradish peroxidase-conjugated secondary antibodies. The final immunoblot signals were visualized using a Shanghai Qinxiang (6000TQ) imaging system and analyzed using the IMAGE J software.

## MitoTracker staining

The culture medium was removed from differentiated cells, followed by the addition of a pre-warmed (37 °C) staining solution containing 200 nM MitoTracker Red CMXRos (Beyotime, C1035). Cells were incubated with the probe for 30 min under standard culture conditions. After staining, images were acquired using a fluorescence microscope (Nikon). Quantitative analysis was performed using the Image-J software.

## Oil Red O staining

Adipocytes were fixed in 4% paraformaldehyde for 10 min, washed twice with PBS, and stained with Oil Red O solution (Servicebio, G1015) for 30 min. The cells were then differentiated with 60% isopropanol for 5 s and washed three times with pure water. Oil Red O staining images were recorded under a microscope and analyzed using the IMAGE J software.

## Biochemical analysis

The non-esterified fatty acids (NEFA) levels in the culture medium and serum levels of triglycerides (TG), total cholesterol (TC), low-density lipoprotein cholesterol (LDL-C), high-density lipoprotein cholesterol (HDL-C), alanine aminotransferase (ALT), aspartate aminotransferase (AST), and creatinine (CREA) were measured using specific kits (Rayto, Guangdong, China or Beyotime, Wuhan, China) according to the manufacturer's instructions.

## Seahorse

SVF cells were seeded into XF24 cell culture microplates (Agilent, 100777-004) at a density of approximately 10,000 cells per well and differentiated for 8 days. Following differentiation, the mature adipocytes were washed twice with XF assay medium (Agilent, 103575-100). The cells were then incubated in XF medium for 1 h

at 37 °C in a non-CO₂ environment to allow temperature and pH equilibration. Oxygen consumption rate (OCR) was subsequently measured using an XF Extracellular Flux Analyzer (Agilent, 102340-100). Mitochondrial stress test compounds—including oligomycin (1 μM), FCCP (1 μM), rotenone (0.5 μM), and antimycin A (2 μM)—were preloaded into injection ports of the sensor cartridge and sequentially injected into the wells during the assay protocol at designated time points. The OCR were adjusted by protein amount.

## GTT and ITT

For the glucose tolerance test (GTT), mice were fasted for 16–18 h before being intraperitoneally injected with glucose (2 g/kg body weight). Blood glucose levels were measured at 0, 15, 30, 60, 90, and 120 min. One week after the GTT, an insulin tolerance test (ITT) was performed. Mice were fasted for 4 h before being intraperitoneally injected with insulin (0.75 U/kg body weight; Sigma-Aldrich, 1342106). Blood glucose levels were measured at 0, 15, 30, 60, 90, and 120 min post-injection.

## Metabolic cages

Metabolic analysis was conducted using the Comprehensive Laboratory Animal Monitoring System (CLAMS, Columbus, USA) at 26 °C under a 12-h light/dark cycle for 24 h.

## Cold exposure test

Rectal basal temperature was measured at room temperature. Mice were then individually housed in cages and exposed to a 4 °C cold room for 4 h. Rectal temperature was measured hourly.

## Histological analysis

BAT, iWAT, eWAT, and liver tissues were fixed in 4% paraformaldehyde at room temperature overnight, dehydrated with ethanol gradients, and infiltrated with xylene before paraffin embedding. Serial 5-μm-thick sections were cut and stained with hematoxylin and eosin (H&E). Images were observed and photographed under a light microscope, and adipocyte size in the three fat tissues was quantified using the Image-J software.

## Immunohistochemistry

Tissue sections were incubated with 3% H₂O₂ to inactivate endogenous peroxidase and blocked with 5% goat serum for 2 h. Sections were then incubated with anti-UCP1 antibody (1:100, ab23841, Abcam) at 4 °C overnight. Afterward, tissues were incubated with horseradish peroxidase-labeled anti-rabbit secondary antibody at room temperature for 1 h. Sections were counterstained with hematoxylin, and images were observed and photographed under a light microscope. Quantitative analysis was performed using the Image-J software (IHC Box).

## LiP-MS

LiP-MS was developed based on previous reports (Zhang et al, 2021). Briefly, the proteome of adipocytes was extracted under

conditions preserving native protein structures (lysis buffer containing 5% sodium deoxycholate, 1 mM KH₂PO₄, 3 mM Na₂HPO₄, 155 mM NaCl, pH 7.5, at 4 °C). The cell lysate was evenly divided into three groups, each with three replicates, and incubated with equal volumes of DMSO, 10 μM, or 50 μM of drug at 25 °C for 10 min. Subsequently, 1% broad-specificity protease K was added to each group and incubated at 25 °C for another 10 min, followed by 95 °C heating for 3 min to inactivate the enzyme, resulting in limited proteolysis to produce structurally specific protein fragments. Reductive alkylation reactions were then carried out by adding tri (2-carboxyethyl) phosphine (TCEP) and chloroacetamide (CAA) and incubating at 37 °C for 1 h. Samples were then diluted with 100 mM Tris-HCl solution to reduce SDC concentration below 1%, and trypsin, at a mass ratio of enzyme to protein of 1:50, was added for overnight incubation at 37 °C with shaking. The reaction was stopped with TFA, yielding peptide mixtures suitable for bottom-up proteomic analysis. The supernatant was subjected to desalting using SDB-RPS columns, dried under vacuum, and stored at −20 °C. Mass spectrometry was performed on a QE-HF instrument in data-dependent acquisition (DDA) mode. Mass spectrometry data were analyzed using Fragpipe v20.0 for LFQ-MBR quantification and identification, with the database being the UniProt mouse reviewed proteins downloaded on November 17, 2023. Parameters were set for semi-tryptic digestion with one missed cleavage site, fixed modification of cysteine urea methylation, and variable modifications of methionine oxidation and protein N-terminal acetylation.

## Molecular docking

The structure of the small molecule compound isomeranzin (CID 473252) was obtained from the PubChem Compound Database. The protein structure of Gnas (AF-P63094-F1-v4) was downloaded from the AlphaFold Protein Structure Database. Autodock 4.2 software (Morris et al, 2009) was utilized for computer-simulated protein-ligand docking. First, all water molecules were removed from the protein and ligand files, and polar hydrogens were added. The grid box was centered to encompass each protein domain, allowing unrestricted molecular migration. The entire docking process was visualized in the model using PyMOL (TM) Molecular Graphics System, Version 2.4.0.

## Cellular thermal shift assay (CETSA)

The cellular thermal shift assay was conducted as previously described (Jafari et al, 2014). In temperature-dependent thermal shift assays, 150 μl of adipocyte lysate (3 mg/ml) was incubated with 50 μM ISM at each temperature point from 40 °C to 67 °C for 3 min. Samples were repeatedly frozen and thawed with liquid nitrogen. Subsequently, centrifugation was performed at 20,000 × g for 10 min at 4 °C to separate the supernatant and pellet. The supernatant was mixed with 5× loading buffer and separated on a 10% SDS-PAGE gel for Gnas immunoblot analysis. In dose-dependent thermal shift assays, 150 μl of lysate (3 mg/ml) was incubated with varying concentrations of ISM (0–50 μM) at 52 °C for 3 min. Samples were similarly frozen and thawed and centrifuged to separate the supernatant, which was then analyzed for Gnas by immunoblotting.

### siRNA production

siRNAs were designed and synthesized by GENERAL BIOL LTD (Anhui, China). Cell transfections were carried out using a commercial transfection reagent (Sangon Biotech, E607402-1000). The siRNA target sequences used in this study are shown in the Table EV5.

### cAMP analysis

The cAMP levels in the culture medium were measured using specific ELISA kit (Bioswamp, Wuhan, China) according to the manufacturer's instructions.

### Statistical analysis

Data were statistically analyzed and plotted using Graphpad Prism 8.0 or SPSS 26.0 software. Experimental data are presented as mean ± standard error of the mean (Mean ± SEM). Unpaired Student's two-tailed t-tests were used to compare differences between two groups assuming a normal distribution; otherwise, Mann–Whitney non-parametric tests were employed. For multiple group comparisons, one-way or two-way ANOVA followed by Tukey's or Dunnett's multiple comparisons test (assuming equal variances) was performed; otherwise, non-parametric Kruskal–Wallis tests were used for further analysis. The energy metabolism data, including $VO_2$ and $VCO_2$ were analyzed by ANCOVA, with body weight as a covariate. Differences were considered statistically significant at $p < 0.05$.

---

#### The paper explained

#### Problem
Obesity has become a global health problem, and stimulating the browning of white adipose tissue (WAT) represents a promising therapeutic approach. However, pharmacological activation of this process remains challenging.

#### Results
Through a combination of Connectivity Map analysis and in vitro functional screening, we identified isomeranzin (ISM) as a novel small-molecule thermogenic activator. ISM upregulates thermogenic genes, including *Ucp1* and *Pgc1a*, in both mouse and human white adipocytes, promoting brown-like morphological and metabolic features without impairing adipogenesis. In mice, ISM induces the browning of sub-cutaneous WAT, enhances thermogenesis, and alleviates high-fat diet (HFD)-induced obesity and metabolic dysfunction. It also improves glucose tolerance and insulin sensitivity in mice on a standard diet. Mechanistically, ISM activates the AMP-activated protein kinase (AMPK) pathway by directly targeting the guanine nucleotide-binding protein G(s) alpha subunit (Gnas), as demonstrated by limited proteolysis–mass spectrometry, cellular thermal shift assays, and molecular docking. Functional rescue experiments confirmed that ISM elevates intracellular cAMP via Gnas to drive AMPK phosphorylation and thermogenic gene expression.

#### Impact
ISM activates the Gnas–cAMP–AMPK signaling to induce thermogenic gene expression, enhance energy expenditure, and improve metabolic health in obese mice. These findings provide a mechanistic insight underlying ISM-driven adipose tissue browning and identify the Gnas–cAMP–AMPK pathway as a promising therapeutic target for obesity and related metabolic disorders.

---

## Data availability

This study includes no data deposited in external repositories.

The source data of this paper are collected in the following database record: biostudies:S-SCDT-10_1038-S44321-025-00335-y.

## Peer review information

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

## Acknowledgements

This work was supported by the National Natural Science Foundation of China (No: 82495174, 81830014, 91949201, 82270614, 81974078, 81570530, 2023YFC2413804).

## Author contributions

**Menghao Shi**: Conceptualization; Data curation; Formal analysis; Validation; Visualization; Methodology; Writing—original draft. **Yinsong Ye**: Data curation; Validation; Methodology. **Lizhi Hu**: Conceptualization; Resources; Formal analysis; Visualization. **Yibo Yan**: Validation. **Shushu Jiang**: Validation. **Pengchao Wang**: Data curation. **Fengcen Li**: Data curation. **Mingfa Ai**: Data curation. **Jinhui Huang**: Data curation. **Ling Yang**: Supervision; Funding acquisition; Writing—review and editing. **Kai Huang**: Supervision; Funding acquisition; Writing—review and editing. **Minglu Liang**: Conceptualization; Supervision; Writing—original draft; Project administration; Writing—review and editing.

Source data underlying figure panels in this paper may have individual authorship assigned. Where available, figure panel/source data authorship is listed in the following database record: biostudies:S-SCDT-10_1038-S44321-025-00335-y.

## Disclosure and competing interests statement

The authors declare no competing interests.

# Expanded View Figures

**Figure EV1. Identification of small-molecule compounds promoting adipocyte browning using CMap.** ▶

(A) Volcano plot of differentially expressed genes in the RNA-seq dataset GSE164219 from cold exposure-induced browning of iWAT in mice ($n = 3$). Statistical test: Moderated t-tests and Benjamini-Hochberg procedure. (B) Volcano plot of differentially expressed genes in the RNA-seq dataset GSE133619 from cold exposure-induced browning of iWAT in mice ($n = 3$). Statistical test: Moderated t-tests and Benjamini-Hochberg procedure. (C) Volcano plot of differentially expressed genes in the RNA-seq dataset GSE129083 from β-adrenergic agonist CL316,243-induced browning of iWAT in mice ($n = 4$). Statistical test: Moderated t-tests and Benjamini-Hochberg procedure. (D) Volcano plot of differentially expressed genes in the RNA-seq dataset GSE98132 from β-adrenergic agonist CL316,243-induced browning of iWAT in mice ($n = 3$). Statistical test: Moderated t-tests and Benjamini-Hochberg procedure. (E) Schematic illustration of the treatment method for maturation and differentiation induction in primary inguinal adipocytes, along with the administration of candidate compounds (10 μM) screened from CMap. (F) Quantitative analysis of relative protein expression levels of Pgc1α in mature primary inguinal adipocytes from control and ISM (50 μM) treated groups ($n = 3$). Statistical test: Unpaired Student's two-tailed t-tests. $p < 0.0001$. (G) Quantitative analysis of OCR in mature primary inguinal adipocytes from control and ISM (50 μM) treated groups in Fig. 2O ($n = 3$). Statistical test: Unpaired Student's two-tailed t-tests. Basal respiration, $p = 0.0142$; Proton leak, $p = 0.0299$; Maximal respiration, $p = 0.0100$; Spare respiratory capacity, $p = 0.0043$; Non-mitochondrial respiration, $p = 0.8941$; ATP production, $p = 0.3578$. (H) Schematic illustration of the treatment method for adipogenesis in adipocytes with ISM. (I) Relative mRNA expression levels of Pparγ, C/Ebpα, and Fabp4 in mature primary inguinal adipocytes treated with different concentrations of ISM ($n = 3$). Statistical test: one-way ANOVA followed by Dunnett's multiple comparisons test. Pparγ: 0 μM vs. 2 μM, $p = 0.6106$; 0 μM vs. 10 μM, $p = 0.6716$; 0 μM vs. 50 μM, $p = 0.7857$; C/Ebpα: 0 μM vs. 2 μM, $p = 0.2238$; 0 μM vs. 10 μM, $p = 0.2367$; 0 μM vs. 50 μM, $p = 0.1733$; Fabp4: 0 μM vs. 2 μM, $p = 0.8648$; 0 μM vs. 10 μM, $p = 0.3654$; 0 μM vs. 50 μM, $p = 0.9361$. (J) Quantitative analysis of relative protein expression levels of Pparγ, C/Ebpα, and Fabp4 in mature primary inguinal adipocytes treated with varying concentrations of ISM ($n = 3$). Statistical test: one-way ANOVA followed by Dunnett's multiple comparisons test. Pparγ: 0 μM vs. 2 μM, $p = 0.3690$; 0 μM vs. 10 μM, $p = 0.9934$; 0 μM vs. 50 μM, $p = 0.5491$; C/Ebpα: 0 μM vs. 2 μM, $p = 0.2496$; 0 μM vs. 10 μM, $p = 0.4520$; 0 μM vs. 50 μM, $p = 0.8255$; Fabp4: 0 μM vs. 2 μM, $p = 0.6430$; 0 μM vs. 10 μM, $p = 0.8337$; 0 μM vs. 50 μM, $p = 0.9254$. $n$ presents biological replicates. Data are presented as mean ± SEM. Statistical significance is indicated as follows: *$p < 0.05$, **$p < 0.01$, ***$p < 0.001$, ****$p < 0.0001$, and ns indicates no significant difference.

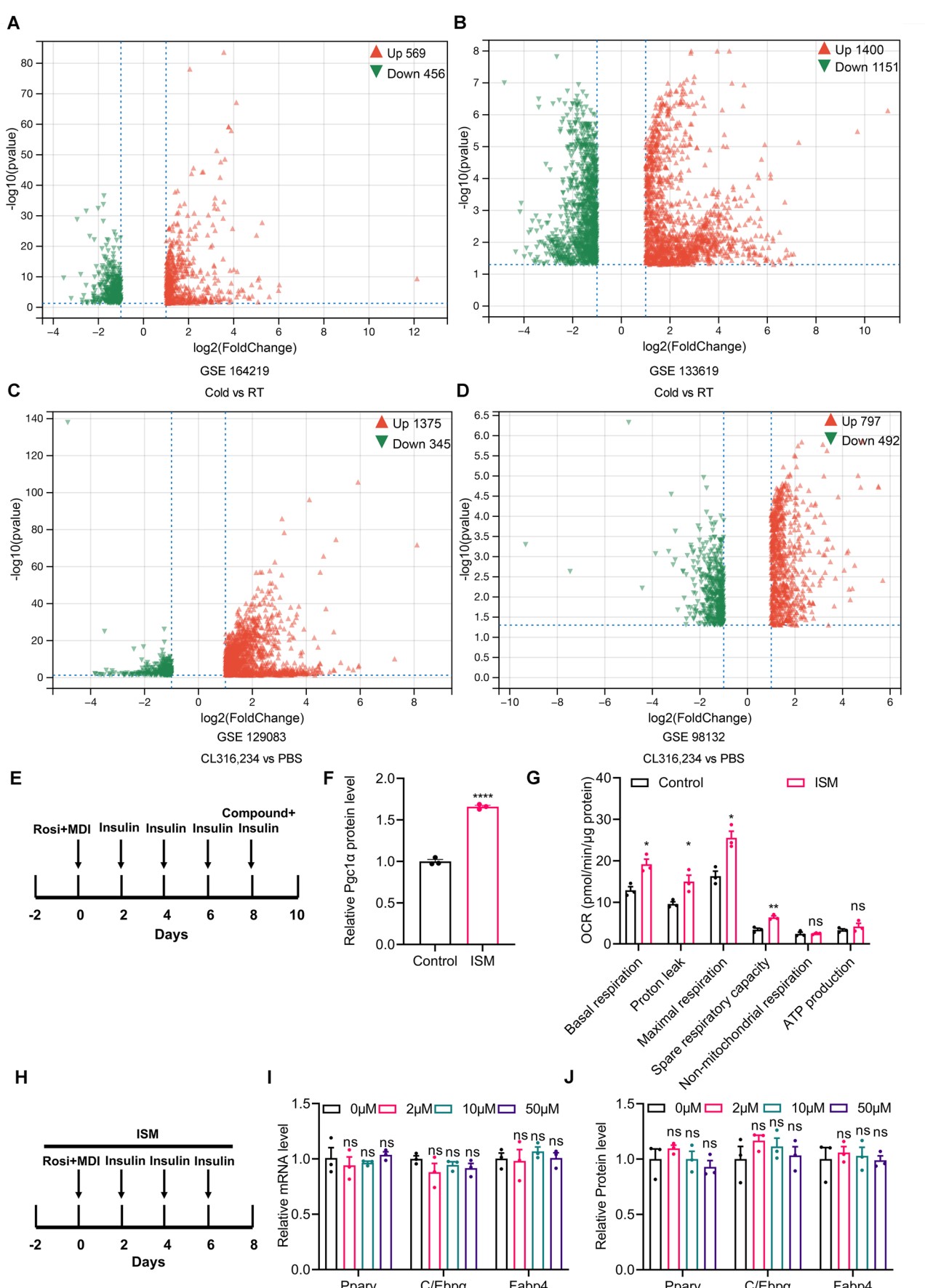

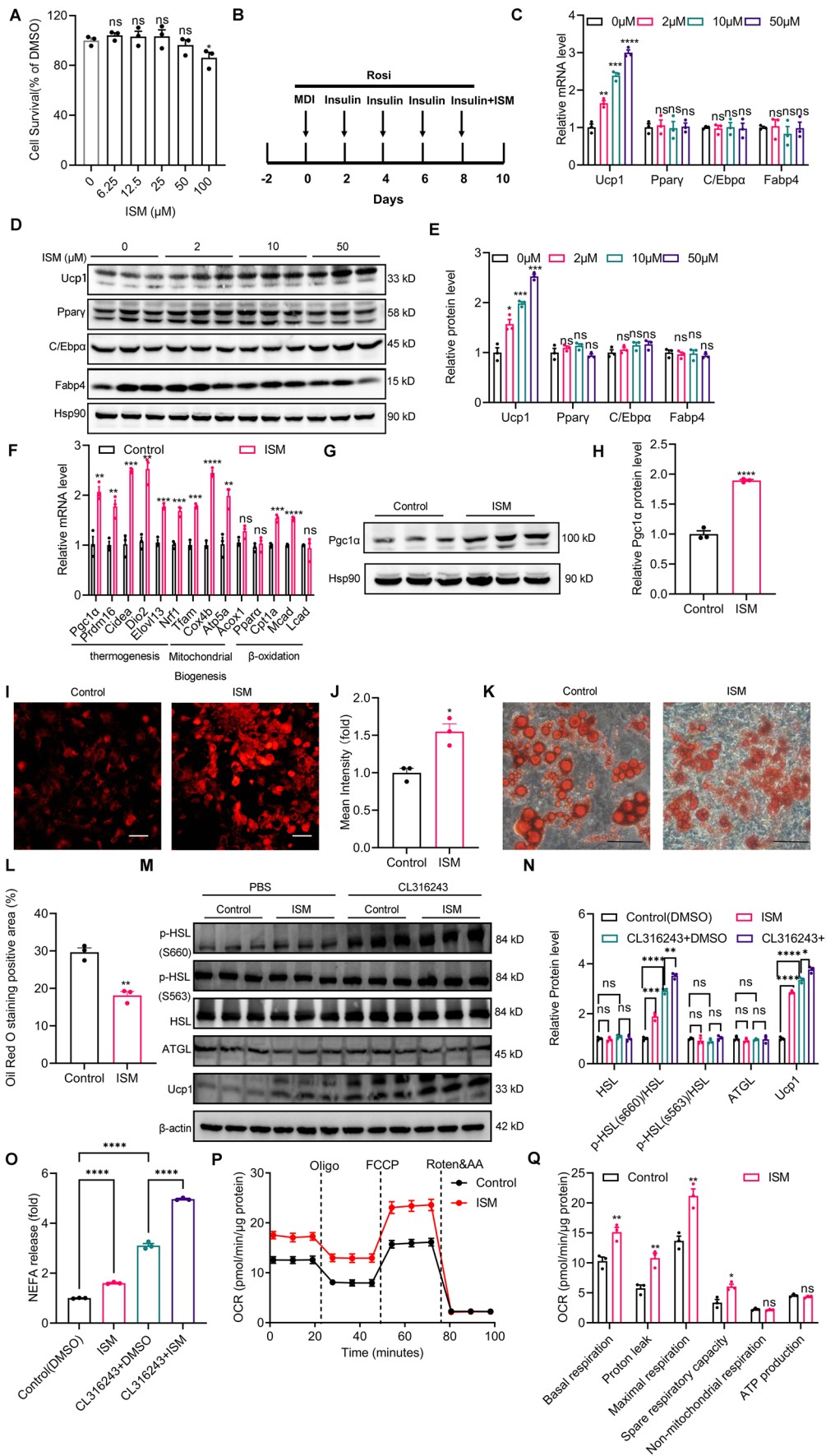

**Figure EV2.** Validation of ISM as a Novel Thermogenic Inducer.

(A) CCK8 assay measuring cell viability in human mesenchymal stem cells (HMSC) treated with different concentrations of ISM ($n = 3$). Statistical test: one-way ANOVA followed by Dunnett's multiple comparisons test.0 vs. 6.25, $p = 0.9070$; 0 vs. 12.5, $p = 0.9608$; 0 vs. 25, $p = 0.9549$; 0 vs. 50, $p = 0.9422$; 0 vs. 100, $p = 0.0420$. (B) Schematic illustration of the treatment method for maturation and differentiation induction in HMSC and the promotion of browning in mature human adipocytes by ISM. (C) Relative mRNA expression levels of Ucp1, Pparγ, C/Ebpα, and Fabp4 in mature human adipocytes derived from HMSC treated with different concentrations of ISM ($n = 3$). Statistical test: one-way ANOVA followed by Dunnett's multiple comparisons test. Ucp1: 0 μM vs. 2 μM, $p = 0.0042$; 0 μM vs. 10 μM, $p = 0.0003$; 0 μM vs. 50 μM, $p < 0.0001$; Pparγ: 0 μM vs. 2 μM, $p = 0.3909$; 0 μM vs. 10 μM, $p = 0.9130$; 0 μM vs. 50 μM, $p = 0.8935$; C/Ebpα: 0 μM vs. 2 μM, $p = 0.6652$; 0 μM vs. 10 μM, $p = 0.6192$; 0 μM vs. 50 μM, $p = 0.8497$; Fabp4: 0 μM vs. 2 μM, $p = 0.8454$; 0 μM vs. 10 μM, $p = 0.4445$; 0 μM vs. 50 μM, $p = 0.9231$. (D) Representative immunoblot images of Ucp1, Pparγ, C/Ebpα, and Fabp4 in mature human adipocytes derived from HMSCs treated with varying concentrations of ISM. (E) Quantitative analysis of relative protein expression levels of Ucp1, Pparγ, C/Ebpα, and Fabp4 in mature human adipocytes derived from HMSCs treated with varying concentrations of ISM ($n = 3$). Statistical test: one-way ANOVA followed by Dunnett's multiple comparisons test. Ucp1: 0 μM vs. 2 μM, $p = 0.0134$; 0 μM vs. 10 μM, $p = 0.0007$; 0 μM vs. 50 μM, $p = 0.0001$; Pparγ: 0 μM vs. 2 μM, $p = 0.3452$; 0 μM vs. 10 μM, $p = 0.2204$; 0 μM vs. 50 μM, $p = 0.5461$; C/Ebpα: 0 μM vs. 2 μM, $p = 0.4425$; 0 μM vs. 10 μM, $p = 0.1332$; 0 μM vs. 50 μM, $p = 0.1133$; Fabp4: 0 μM vs. 2 μM, $p = 0.6264$; 0 μM vs. 10 μM, $p = 0.8377$; 0 μM vs. 50 μM, $p = 0.3627$. (F) Relative mRNA expression levels of thermogenesis, mitochondrial biogenesis and β-oxidation related genes in mature human adipocytes derived from HMSCs from control and ISM (50 μM) treated groups ($n = 3$). Statistical test: Unpaired Student's two-tailed t-tests. Pgc1α, $p = 0.0045$; Prdm16, $p = 0.0060$; Cidea, $p = 0.0004$; Dio2, $p = 0.0013$; Elovl13, $p = 0.0004$; Nrf1, $p = 0.0008$; Tfam, $p = 0.4859$; Cox4b, $p = 0.0004$; Atp5a, $p = 0.0067$; Acox1, $p = 0.0745$; Pparα, $p = 0.4859$; Cpt1a, $p = 0.0004$; Mcad, $p < 0.0001$; Lcad, $p = 0.6934$. (G) Representative immunoblot images of Pgc1α in mature human adipocytes derived from HMSC in control and ISM (50 μM) treatment groups. (H) Quantitative analysis of relative protein expression levels of Pgc1α in mature human adipocytes derived from HMSC in control and ISM (50 μM) treatment groups ($n = 3$). Statistical test: Unpaired Student's two-tailed t-tests. $p < 0.0001$; (I) Representative MitoTracker staining images of mature human adipocytes derived from HMSC from control and ISM (50 μM) treated groups; scale bar = 50 μm. (J) Quantitative analysis of MitoTracker staining images in mature primary inguinal adipocytes from control and ISM (50 μM) treated groups ($n = 3$). Statistical test: Unpaired Student's two-tailed t-tests. $p = 0.0105$. (K) Representative Oil Red O staining images of mature human adipocytes derived from HMSC from control and ISM (50 μM) treated groups; scale bar = 50 μm. (L) Quantitative analysis of Oil Red O staining images in mature human adipocytes derived from HMSC from control and ISM (50 μM) treated groups ($n = 3$). Statistical test: Unpaired Student's two-tailed t-tests. $p = 0.0017$. After 24 h of ISM treatment, the cells were stimulated with 1 μM CL316243 for an additional 24 h. (M) Representative immunoblot images of p-HSL(Ser660), p-HSL(Ser563), HSL, ATGL and Ucp1 in mature human adipocytes derived from HMSC from different groups. (N) Quantitative analysis of relative protein expression levels of p-HSL(Ser660), p-HSL (Ser563), HSL, ATGL and Ucp1 in mature human adipocytes derived from HMSC from different groups ($n = 3$). Statistical test: one-way ANOVA followed by Tukey's multiple comparisons. HSL: Control (DMSO) vs. ISM, $p = 0.5801$; Control (DMSO) vs. CL316243 + DMSO, $p = 0.2296$; CL316243 + DMSO vs. CL316243 + ISM, $p = 0.4517$; p-HSL(S660)/HSL: Control (DMSO) vs. ISM, $p = 0.0009$; Control (DMSO) vs. CL316243 + DMSO, $p < 0.0001$; CL316243 + DMSO vs. CL316243 + ISM, $p = 0.0032$; p-HSL(S563)/HSL: Control (DMSO) vs. ISM, $p = 0.5010$; Control (DMSO) vs. CL316243 + DMSO, $p = 0.1447$; CL316243 + DMSO vs. CL316243 + ISM, $p = 0.1580$; ATGL: Control (DMSO) vs. ISM, $p = 0.3646$; Control (DMSO) vs. CL316243 + DMSO, $p = 0.5620$; CL316243 + DMSO vs. CL316243 + ISM, $p = 0.8974$; Ucp1: Control (DMSO) vs. ISM, $p < 0.0001$; Control (DMSO) vs. CL316243 + DMSO, $p < 0.0001$; CL316243 + DMSO vs. CL316243 + ISM, $p = 0.0146$. (O) NEFA levels in the mature human adipocytes derived from HMSC culture medium from different groups ($n = 3$). Statistical test: one-way ANOVA followed by Tukey's multiple comparisons. Control (DMSO) vs. ISM, $p < 0.0001$; Control (DMSO) vs. CL316243 + DMSO, $p < 0.0001$; CL316243 + DMSO vs. CL316243 + ISM, $p < 0.0001$. (P) Oxygen consumption rates (OCR) in mature human adipocytes derived from HMSC from control and ISM (50 μM) treated groups ($n = 3$). (Q) Quantitative analysis of OCR in mature human adipocytes derived HMSC from control and ISM (50 μM) treated groups ($n = 3$). Statistical test: Unpaired Student's two-tailed t-tests. Basal respiration, $p = 0.0095$; Proton leak, $p = 0.0070$; Maximal respiration, $p = 0.0058$; Spare respiratory capacity, $p = 0.0147$; Non-mitochondrial respiration, $p = 0.5466$; ATP production, $p = 0.3115$. $n$ presents biological replicates. Data are presented as mean ± SEM. Statistical significance is indicated as follows: *$p < 0.05$, **$p < 0.01$, ***$p < 0.001$, ****$p < 0.0001$, and ns indicates no significant difference.

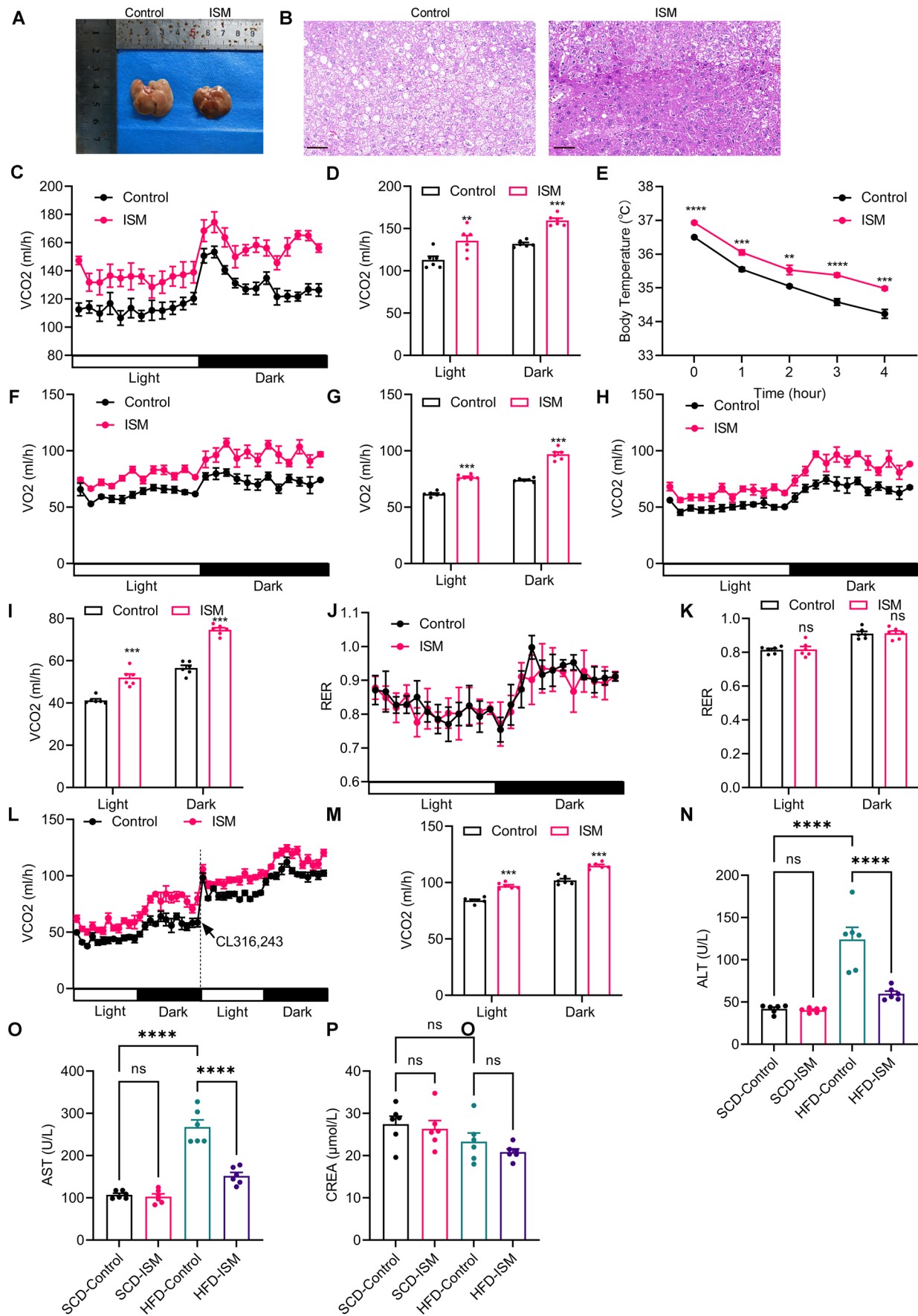

◄

**Figure EV3.  ISM protects against obesity induced by high-fat diet and improves obesity-related metabolic dysfunction through promoting thermogenesis in mice.**

(A) Representative images of the general appearance of livers from HFD mice. (B) Representative H&E staining images of liver sections; scale bar = 50 μm. (C) $VCO_2$ of HFD mice. (D) Quantitative analysis of $VCO_2$ in (C). Statistical test: ANCOVA, with body weight as a covariate. $VCO_2$-Light, $p = 0.004$; $VCO_2$-Dark, $p < 0.001$. (E) Rectal temperature of HFD mice under different treatments measured at specified times at 4 °C ($n = 6$). Statistical test: Unpaired Student's two-tailed t-tests. 0 h, $p < 0.0001$; 1 h, $p = 0.0005$; 2 h, $p = 0.0075$; 3 h, $p < 0.0001$; 4 h, $p = 0.0003$. (F, G) $VO_2$ of mice injected with either vehicle or ISM and fed a high-fat diet for 4 weeks ($n = 6$). Statistical test: ANCOVA, with body weight as a covariate. $VO_2$-Light, $p < 0.001$; $VO_2$-Dark, $p < 0.001$. (H, I) $VCO_2$ of mice injected with either vehicle or ISM and fed a high-fat diet for 4 weeks ($n = 6$). Statistical test: ANCOVA, with body weight as a covariate. $VCO_2$-Light, $p < 0.001$; $VCO_2$-Dark, $p < 0.001$. (J, K) Respiratory exchange ratio (RER) of mice injected with either vehicle or ISM and fed a high-fat diet for 4 weeks ($n = 6$). Statistical test: Unpaired Student's two-tailed t-tests. RER-Light, $p = 0.9647$; RER-Dark, $p = 0.3803$. (L) $VCO_2$ of mice treated with CL316243. (M) Quantitative analysis of $VCO_2$ in mice after CL316243 treatment ($n = 6$). Statistical test: ANCOVA, with body weight as a covariate. $VCO_2$-Light, $p < 0.001$; $VCO_2$-Dark, $p < 0.001$. (N) Quantitative analysis of serum ALT levels in mice from different groups ($n = 6$). Statistical test: one-way ANOVA followed by Tukey's multiple comparisons. SCD-Control vs. SCD-ISM, $p = 0.9989$; SCD-Control vs. HFD-Control, $p < 0.0001$; HFD-Control vs. HFD-ISM, $p < 0.0001$. (O) Quantitative analysis of serum AST levels in mice from different groups ($n = 6$). Statistical test: one-way ANOVA followed by Tukey's multiple comparisons. SCD-Control vs. SCD-ISM, $p = 0.9903$; SCD-Control vs. HFD-Control, $p < 0.0001$; HFD-Control vs. HFD-ISM, $p < 0.0001$. (P) Quantitative analysis of serum CREA levels in mice from different groups ($n = 6$). Statistical test: one-way ANOVA followed by Tukey's multiple comparisons. SCD-Control vs. SCD-ISM, p = 0.9681; SCD-Control vs. HFD-Control, $p = 0.3478$; HFD-Control vs. HFD-ISM, $p = 0.7391$. $n$ presents biological replicates. Data are presented as mean ± SEM. Statistical significance is indicated as follows: *$p < 0.05$, **$p < 0.01$, ***$p < 0.001$, ****$p < 0.0001$, and ns indicates no significant difference.

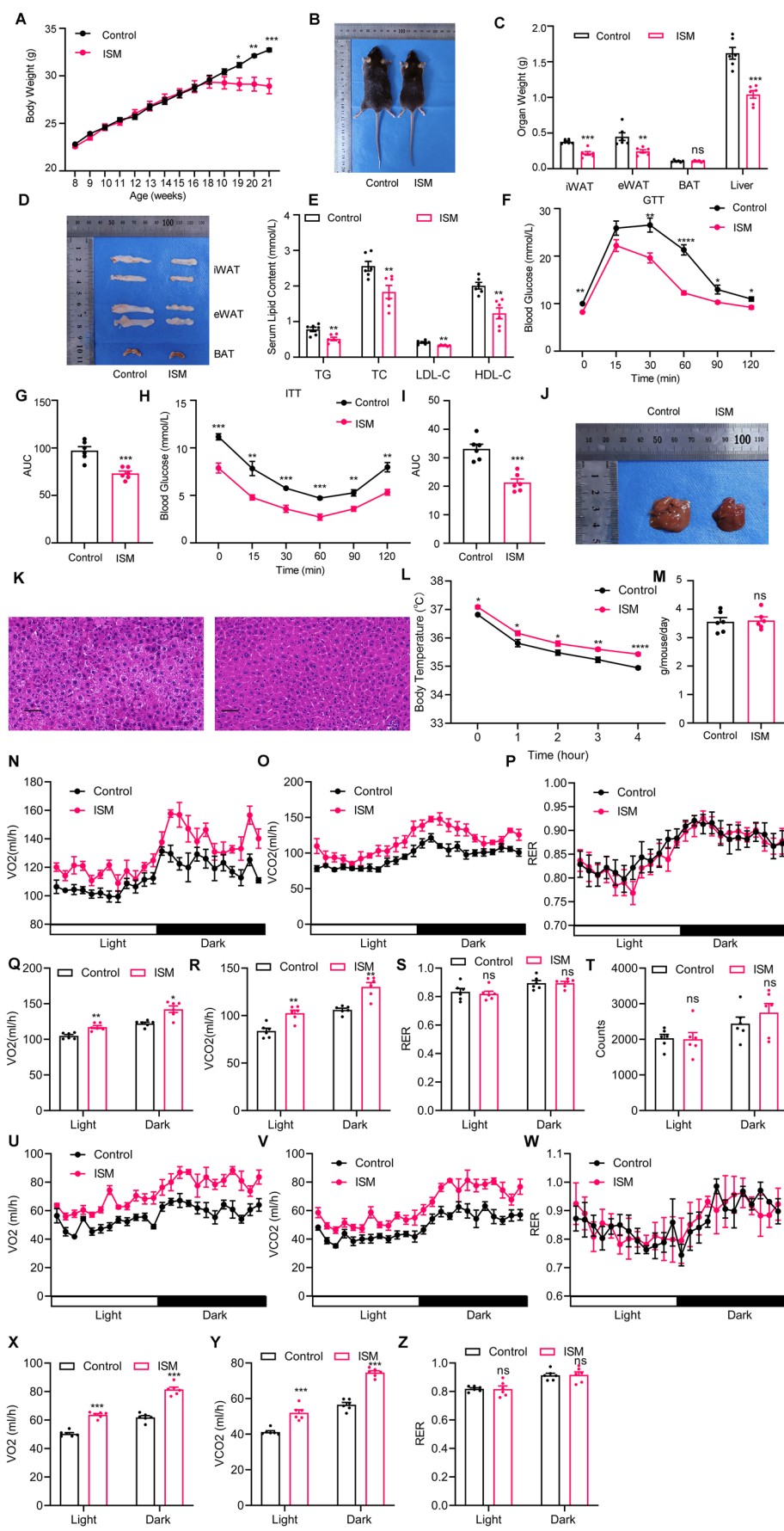

◀

**Figure EV4. Administration of ISM also improves glucose tolerance and insulin sensitivity in mice fed a standard chow diet (SCD).**

Mice were intraperitoneally injected with either vehicle or ISM and fed a standard chow diet for 13 weeks ($n = 6$). $n$ presents biological replicates. (**A**) Body weight of mice in different groups after feeding on a standard chow diet. 19w, $p = 0.0173$; 20w, $p = 0.0025$; 21w, $p = 0.0009$. (**B**) Representative images of the general appearance of mice prior to euthanasia. (**C**) Weights of tissue organs (inguinal subcutaneous adipose tissue, epididymal visceral adipose tissue, brown adipose tissue, and liver) from mice. iWAT, $p = 0.0001$; eWAT, $p = 0.0058$; BAT, $p = 0.9624$; Liver, $p = 0.0002$. (**D**) Representative images of the general appearance of adipose tissues (iWAT, eWAT, and BAT) from mice. (**E**) Serum lipid content (triglycerides, total cholesterol, low-density lipoprotein cholesterol, high-density lipoprotein cholesterol) after feeding on a standard chow diet for 13 weeks. TG, $p = 0.0059$; TC, $p = 0.0090$; LDL-C, $p = 0.0037$; HDL-C, $p = 0.0015$. (**F**) Glucose tolerance test (GTT) after feeding on SCD for 11 weeks. 0 min, $p = 0.0023$; 15 min, $p = 0.0881$; 30 min, $p = 0.0034$; 60 min, $p < 0.0001$; 90 min, $p = 0.0205$; 120 min, $p = 0.0134$. (**G**) Quantitative analysis of the glucose tolerance test (GTT). $p = 0.0005$. (**H**) Insulin tolerance test (ITT) after feeding on SCD for 12 weeks. 0 min, $p = 0.0003$; 15 min, $p = 0.0033$; 30 min, $p = 0.0008$; 60 min, $p = 0.0002$; 90 min, $p = 0.0021$; 120 min, $p = 0.0010$. (**I**) Quantitative analysis of the insulin tolerance test (ITT). $p = 0.0002$. (**J**) Representative images of the general appearance of livers from SCD mice. (**K**) Representative H&E staining images of liver sections; scale bar $= 50$ μm. (**L**) Rectal temperature of SCD mice under different treatments measured at specified times at 4 °C. 0 h, $p = 0.0267$; 1 h, $p = 0.0426$; 2 h, $p = 0.0324$; 3 h, $p = 0.0059$; 4 h, $p < 0.0001$. (**M**) Food intake in mice. $p = 0.8306$. (**N–P**) $VO_2$, $VCO_2$ and Respiratory exchange ratio (RER) of SCD mice. (**Q–S**) Quantitative analysis of $VO_2$, $VCO_2$ and RER of SCD mice in (**N–P**). $VO_2$-Light, $p = 0.007$; $VO_2$-Dark, $p = 0.018$; $VCO_2$-Light, $p = 0.001$; $VCO_2$-Dark, $p = 0.001$; RER-Light, $p = 0.6621$; RER-Dark, $p = 0.9868$. (**T**) Mean physical activity in mice. Light, $p = 0.9077$; Dark, $p = 0.3431$. (**U–W**) $VO_2$, $VCO_2$ and RER of mice injected with either vehicle or ISM and fed a standard chow diet for 4 weeks ($n = 6$). $n$ presents biological replicates. (**X–Z**) Quantitative analysis of $VO_2$, $VCO_2$ and RER of SCD mice in (**U–W**) ($n = 6$). $VO_2$-Light, $p < 0.001$; $VO_2$-Dark, $p < 0.001$; $VCO_2$-Light, $p < 0.001$; $VCO_2$-Dark, $p < 0.001$; RER-Light, $p = 0.8021$; RER-Dark, $p = 0.7714$. Data are presented as mean ± SEM. Statistical test: The energy metabolism data, including $VO_2$ and $VCO_2$ were analyzed by ANCOVA, with body weight as a covariate. The others: Unpaired Student's two-tailed t-tests. Statistical significance is indicated as follows: *$p < 0.05$, **$p < 0.01$, ***$p < 0.001$, ****$p < 0.0001$, and ns indicates no significant difference.

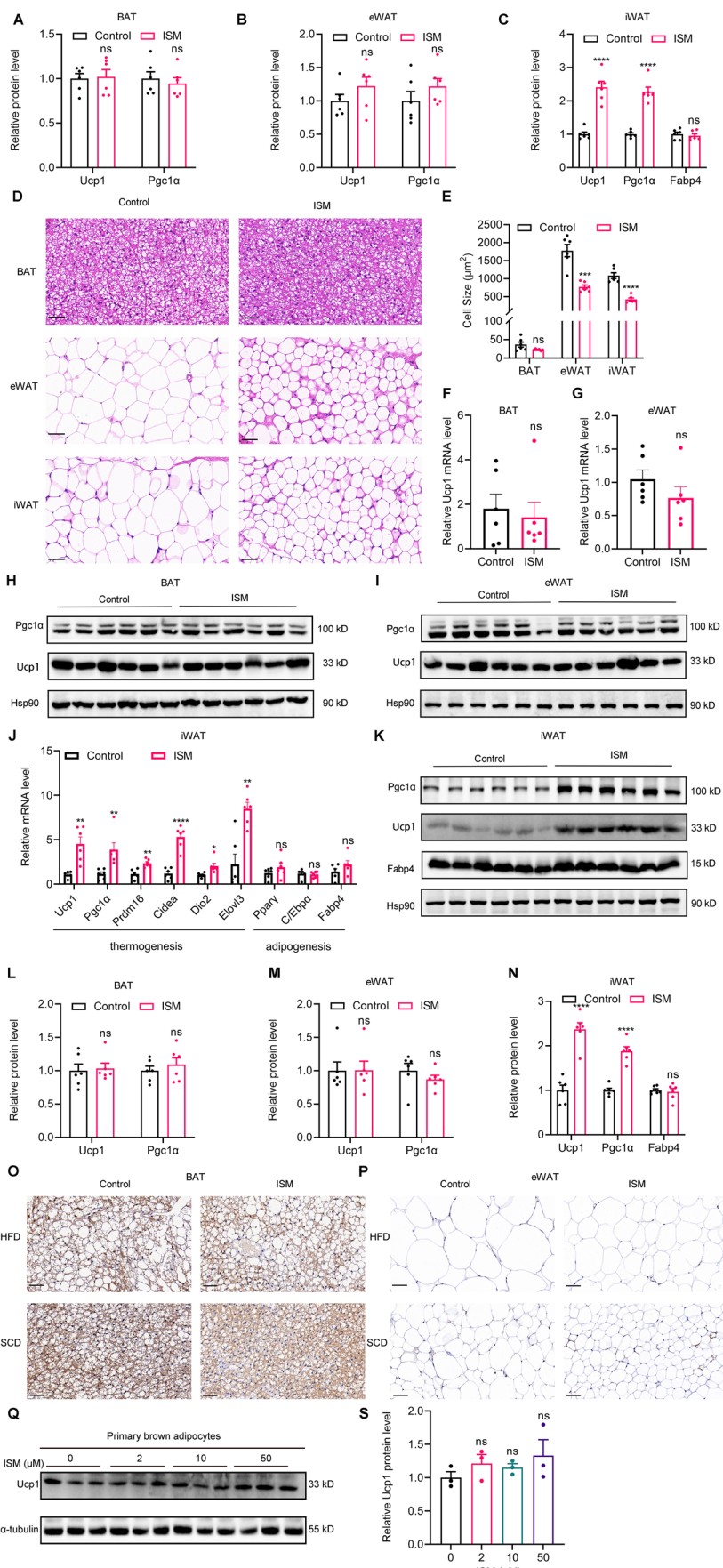

**Figure EV5.  ISM administration also promotes subcutaneous adipose browning and increases thermogenesis in mice fed a standard chow diet (SCD).**

(A) Quantitative analysis of relative protein expression levels for the proteins shown in Fig. 4E ($n = 6$). Ucp1, $p = 0.8490$; Pgc1α, $p = 0.6006$. (B) Quantitative analysis of relative protein expression levels for the proteins shown in Fig. 4F ($n = 6$). Ucp1, $p = 0.2182$; Pgc1α, $p = 0.2610$. (C) Quantitative analysis of relative protein expression levels for the proteins shown in Fig. 4H ($n = 6$). Ucp1, $p < 0.0001$; Pgc1α, $p < 0.0001$; Fabp4, $p = 0.6071$. (D) Representative H&E staining images of BAT, eWAT, and iWAT sections; scale bar $= 50$ μm. (E) Quantitative analysis of adipocyte size in BAT, eWAT, and iWAT ($n = 6$). BAT, $p = 0.0708$; eWAT, $p = 0.0002$; iWAT, $p < 0.0001$. (F) Relative mRNA expression levels of Ucp1 in BAT ($n = 6$). $p = 0.6939$. (G) Relative mRNA expression levels of Ucp1 in eWAT ($n = 6$). $p = 0.2254$. (H) Representative immunoblot images of Pgc1α and Ucp1 in BAT. (I) Representative immunoblot images of Pgc1α and Ucp1 in eWAT. (J) Relative mRNA expression levels of thermogenesis and adipogenesis related genes in iWAT ($n = 6$). Ucp1, $p = 0.0015$; Pgc1α, $p = 0.0063$; Prdm16, $p = 0.0016$; Cidea, $p < 0.0001$; Dio2, $p = 0.0150$; Elovl3, $p = 0.0010$; Ppary, $p = 0.2402$; C/Ebpα, $p = 0.7316$; Fabp4, $p = 0.1663$. (K) Representative immunoblot images of Pgc1α, Ucp1, and Fabp4 in iWAT. (L) Quantitative analysis of relative protein expression levels for the proteins shown in (H) ($n = 6$). Ucp1, $p = 0.8064$; Pgc1α, $p = 0.4848$. (M) Quantitative analysis of relative protein expression levels for the proteins shown in (I) ($n = 6$). Ucp1, $p = 0.9714$; Pgc1α, $p = 0.3209$. (N) Quantitative analysis of relative protein expression levels for the proteins shown in (K) ($n = 6$). Ucp1, $p < 0.0001$; Pgc1α, $p < 0.0001$; Fabp4, $p = 0.6702$. (O) Representative immunohistochemical staining images of Ucp1 in BAT sections; scale bar $= 50$ μm. (P) Representative immunohistochemical staining images of Ucp1 in eWAT sections; scale bar $= 50$ μm. (Q) Representative immunoblot images of Ucp1 in mature primary brown adipocytes treated with varying concentrations of ISM. (S) Quantitative analysis of relative protein expression levels of Ucp1 in mature primary brown adipocytes treated with varying concentrations of ISM ($n = 3$). 0 vs. 2, $p = 0.6256$; 0 vs. 10, $p = 0.8106$; 0 vs. 50, $p = 0.3200$. $n$ presents biological replicates. Data are presented as mean ± SEM. Statistical test: (S): one-way ANOVA followed by Dunnett's multiple comparisons test. The others: unpaired Student's two-tailed t-tests. Statistical significance is indicated as follows: *$p < 0.05$, **$p < 0.01$, ***$p < 0.001$, ****$p < 0.0001$, and ns indicates no significant difference.

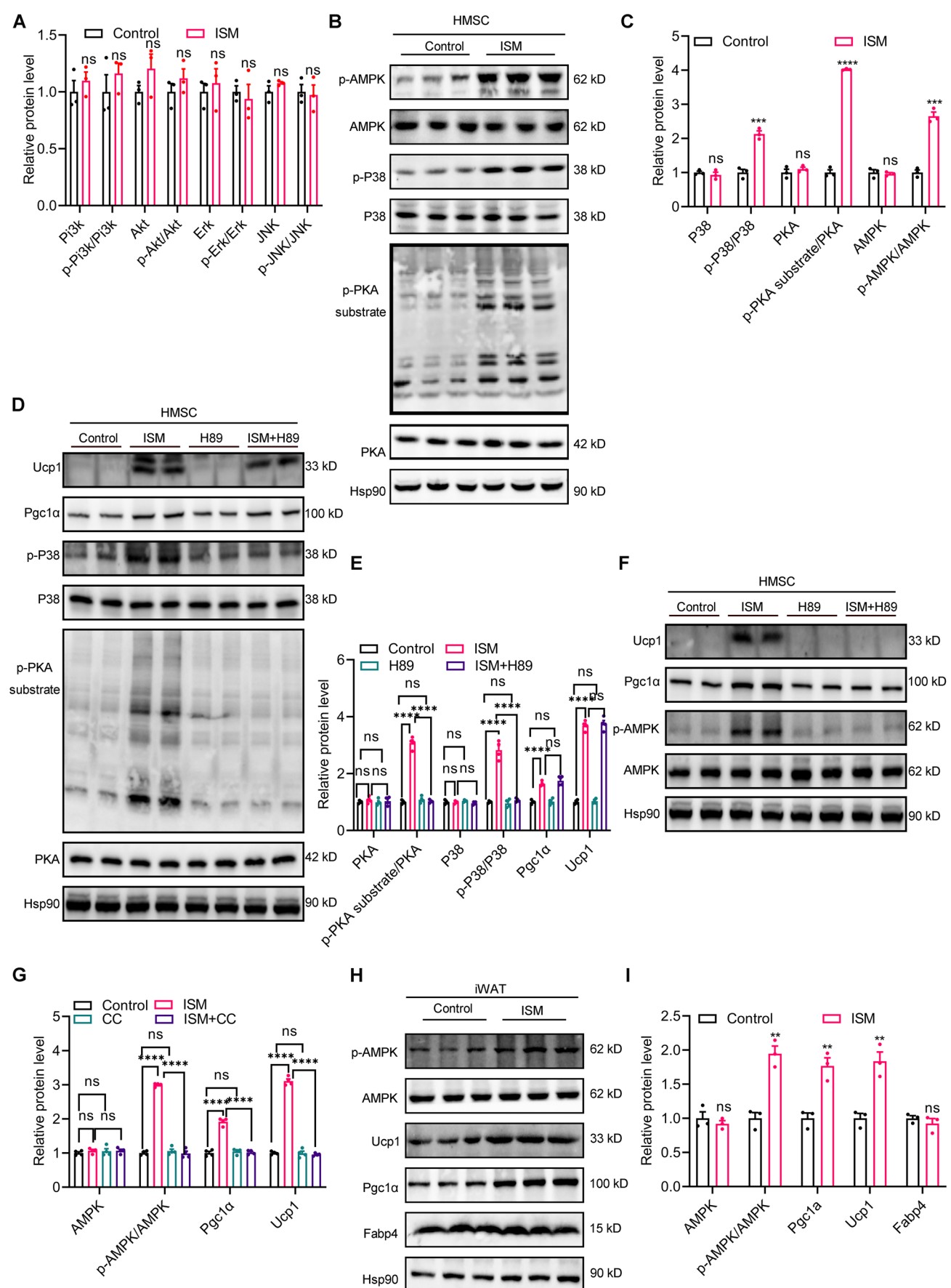

◀ **Figure EV6.  ISM promotes browning by activating the AMPK pathway.**

(A) Quantitative analysis of relative protein expression levels for the signaling pathway proteins shown in Fig. 5C ($n = 3$). Unpaired Student's two-tailed t-tests. Pi3k, $p = 0.4749$; p-Pi3k/Pi3k, $p = 0.4071$; Akt, $p = 0.2135$; p-Akt/Akt, $p = 0.3419$; Erk, $p = 0.6300$; p-Erk/Erk, $p = 0.6766$; JNK, $p = 0.2356$; p-JNK/JNK, $p = 0.8187$. (B) Representative immunoblot images of signaling pathway proteins in mature human adipocytes from control and ISM (50 μM) treatment groups. (C) Quantitative analysis of relative protein expression levels for the signaling pathway proteins shown in (B) ($n = 3$). Unpaired Student's two-tailed t-tests. P38, $p = 0.4837$; p-P38/P38, $p = 0.0010$; PKA, $p = 0.4066$; p-PKA substrate/PKA, $p < 0.0001$; AMPK, $p = 0.6970$; p-AMPK/AMPK, $p = 0.0003$. (D) Representative immunoblot images of signaling pathway proteins in mature human adipocytes treated with or without PKA inhibitor H89 (30 μM) for 2 h, followed by ISM or DMSO treatment for 48 h. (E) Quantitative analysis of relative protein expression levels for the signaling pathway proteins shown in (D) ($n = 3$). One-way ANOVA followed by Tukey's multiple comparisons. PKA: Control vs. ISM, $p = 0.3360$; Control vs. H89, $p = 0.9175$; ISM vs. ISM + H89, $p = 0.7420$; p-PKA substrate/PKA: Control vs. ISM, $p < 0.0001$; Control vs. H89, $p = 0.2637$; ISM vs. ISM + H89, $p < 0.0001$; P38: Control vs. ISM, $p = 0.7647$; Control vs. H89, $p = 0.4293$; ISM vs. ISM + H89, $p = 0.4467$; p-P38/P38: Control vs. ISM, $p < 0.0001$; Control vs. H89, $p = 0.5466$; ISM vs. ISM + H89, $p < 0.0001$; Pgc1α: Control vs. ISM, $p < 0.0001$; Control vs. H89, $p = 0.8756$; ISM vs. ISM + H89, $p = 0.2669$; Ucp1: Control vs. ISM, $p < 0.0001$; Control vs. H89, $p = 0.5887$; ISM vs. ISM + H89, $p = 0.6622$. (F) Representative immunoblot images of signaling pathway proteins in mature human adipocytes treated with or without AMPK inhibitor CC (10 μM) for 2 h, followed by ISM or DMSO treatment for 48 h. (G) Quantitative analysis of relative protein expression levels for the signaling pathway proteins shown in (F) ($n = 3$). One-way ANOVA followed by Tukey's multiple comparisons. AMPK: Control vs. ISM, $p = 0.2372$; Control vs. CC, $p = 0.5120$; ISM vs. ISM + CC, $p = 0.9842$; p-AMPK/AMPK: Control vs. ISM, $p < 0.0001$; Control vs. CC, $p = 0.3807$; ISM vs. ISM + CC, $p < 0.0001$; Pgc1α: Control vs. ISM, $p < 0.0001$; Control vs. CC, $p = 0.4606$; ISM vs. ISM + CC, $p < 0.0001$; Ucp1: Control vs. ISM, $p < 0.0001$; Control vs. CC, $p = 0.9252$; ISM vs. ISM + CC, $p < 0.0001$. (H) Representative immunoblot images of AMPK, p-AMPK, Ucp1, Pgc1α, and Fabp4 in iWAT from mice fed a normal diet and injected with ISM or vehicle. (I) Quantitative analysis of relative protein expression levels for the proteins shown in (H) ($n = 3$). Unpaired Student's two-tailed t-tests. AMPK, $p = 0.4924$; p-AMPK/AMPK, $p = 0.0026$; Pgc1α, $p = 0.0055$; Ucp1, $p = 0.0056$; Fabp4, $p = 0.4072$. $n$ presents biological replicates. Data are presented as mean ± SEM. Statistical significance is indicated as follows: *$p < 0.05$, **$p < 0.01$, ***$p < 0.001$, ****$p < 0.0001$, and ns indicates no significant difference.

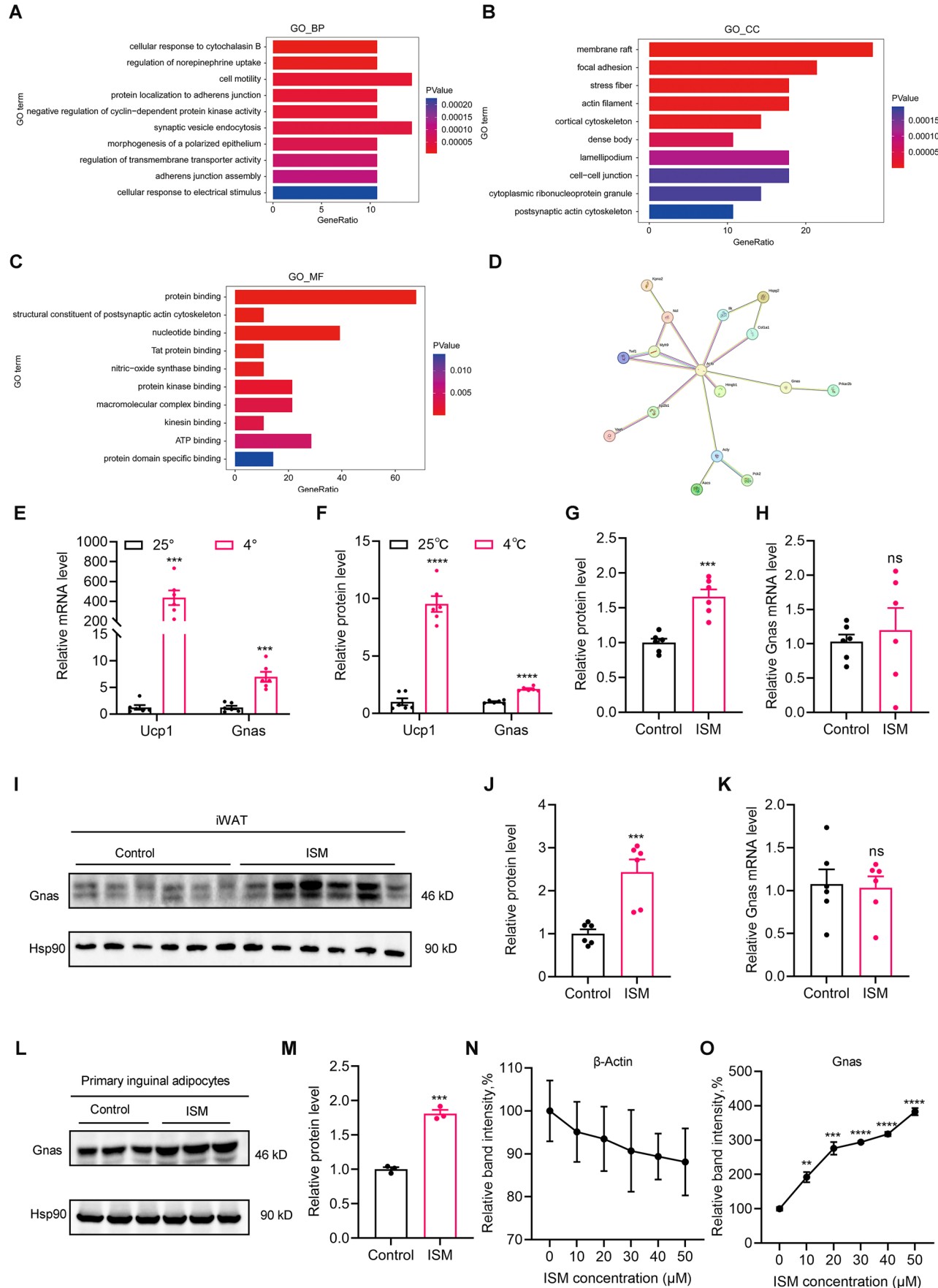

**Figure EV7.** **Discovery of direct binding targets of ISM.**

(**A–C**) Bar charts of GO enrichment analysis for 26 potential target proteins. (**D**) Protein–protein interaction (PPI) network diagram of 26 potential target proteins. (**E**) Relative mRNA expression levels of Gnas and Ucp1 in iWAT of 8-week-old male C57BL/6J mice exposed to 25 °C or 4 °C environments for 3 days ($n = 6$). Ucp1, $p = 0.0001$; Gnas, $p = 0.0002$. (**F**) Quantitative analysis of relative expression levels for the proteins shown in Fig. 6D ($n = 6$). Ucp1, $p < 0.0001$; Gnas, $p < 0.0001$. (**G**) Quantitative analysis of relative protein levels for the Gnas shown in Fig. 6E ($n = 6$). $p = 0.0003$. (**H**) Relative mRNA levels of Gnas in iWAT of mice fed a high-fat diet and injected with ISM or vehicle ($n = 6$). $p = 0.6227$. (**I**) Representative immunoblot images of Gnas in iWAT of mice fed a normal diet and injected with ISM or vehicle. (**J**) Quantitative analysis of relative protein levels for the Gnas shown in (**I**) ($n = 6$). $p = 0.0009$. (**K**) Relative mRNA levels of Gnas in iWAT of mice fed a normal diet and injected with ISM or vehicle ($n = 6$). $p = 0.8522$. (**L**) Representative immunoblot images of Gnas in mature primary inguinal adipocytes from control and ISM (50 µM) treatment groups. (**M**) Quantitative analysis of relative protein levels for the Gnas shown in (**L**) ($n = 3$). $p = 0.0001$. (**N**) Quantitative analysis of relative grayscale values for β-actin in Fig. 6J ($n = 3$). 0 vs. 10, $p = 0.9860$; 0 vs. 20, $p = 0.9547$; 0 vs. 30, $p = 0.8439$; 0 vs. 40, $p = 0.7713$; 0 vs. 50, $p = 0.6964$. (**O**) Quantitative analysis of relative grayscale values for Gnas in Fig. 6J ($n = 3$). 0 vs. 10, $p = 0.0003$; 0 vs. 20, $p < 0.0001$; 0 vs. 30, $p < 0.0001$; 0 vs. 40, $p < 0.0001$; 0 vs. 50, $p < 0.0001$. $n$ presents biological replicates. Data are presented as mean ± SEM. Statistical test: (**N** and **O**): one-way ANOVA followed by Dunnett's multiple comparisons test. The others: unpaired Student's two-tailed t-tests. Statistical significance is indicated as follows: *$p < 0.05$, **$p < 0.01$, ***$p < 0.001$, ****$p < 0.0001$, and ns indicates no significant difference.

