## [Peer Review File · EMBO Molecular Medicine]

Isomeranzin activates Gnas-AMPK signaling to drive white adipose browning and curb obesity in mice

Menghao Shi, Yinsong Ye, Lizhi Hu, Yibo Yan, Shushu Jiang, Pengchao Wang, Fengcen Li, Mingfa Ai, Jinhun Huang, Ling Yang, Kai Huang, and Minglu Liang

Corresponding authors: Minglu Liang (liangml@hust.edu.cn), Ling Yang (yanglinguh@hust.edu.cn), Kai Huang (huangkai1@hust.edu.cn)

Review Timeline:

Submission Date:	23rd Feb 25
Editorial Decision:	28th Mar 25
Revision Received:	12th Sep 25
Editorial Decision:	14th Oct 25
Revision Received:	21st Oct 25
Accepted:	29th Oct 25

Editor: Zeljko Durdevic

Transaction Report:

28th Mar 2025

Dear Dr. Liang,

Thank you for the submission of your manuscript to EMBO Molecular Medicine. We have now received feedback from the three reviewers who agreed to evaluate your manuscript. All referees recognize interest of the study but also raise serious concerns that should be addressed in a major revision. Focus of the revision should be in strengthening the in vivo data but also in providing more experimental evidence for the major claims of the study. Considering that the revision will require extensive experimentation we think six months rather than three months would be more appropriate to provide the complete revision. If you would like to discuss further the points raised by the referees, I am available to do so via email or video. Let me know if you are interested in this option.

We would welcome the submission of a revised version within six months for further consideration. Please let us know if you require longer to complete the revision.

I look forward to receiving your revised manuscript.

Yours sincerely,

Zeljko Durdevic

We require:

- 1) A .docx formatted version of the manuscript text (including legends for main figures, EV figures and tables). Please make sure that the changes are highlighted to be clearly visible.
- 2) Individual production quality figure files as .eps, .tif, .jpg (one file per figure). For guidance, download the 'Figure Guide PDF': (<https://www.embopress.org/page/journal/17574684/authorguide#figureformat>).
- 3) A .docx formatted letter INCLUDING the reviewers' reports and your detailed point-by-point responses to their comments. As part of the EMBO Press transparent editorial process, the point-by-point response is part of the Review Process File (RPF), which will be published alongside your paper.
- 4) A complete author checklist, which you can download from our author guidelines (<https://www.embopress.org/page/journal/17574684/authorguide#submissionofrevisions>). Please insert information in the checklist that is also reflected in the manuscript. The completed author checklist will also be part of the RPF.
- 5) Please note that all corresponding authors are required to supply an ORCID ID for their name upon submission of a revised manuscript.

6) It is mandatory to include a 'Data Availability' section after the Materials and Methods. Before submitting your revision, primary datasets produced in this study need to be deposited in an appropriate public database, and the accession numbers and database listed under 'Data Availability'. Please remember to provide a reviewer password if the datasets are not yet public (see <https://www.embopress.org/page/journal/17574684/authorguide#dataavailability>).

12) Author contributions: You will be asked to provide CRediT (Contributor Role Taxonomy) terms in the submission system. These replace a narrative author contribution section in the manuscript.

13) A Conflict of Interest statement should be provided in the main text.

14) Every published paper now includes a 'Synopsis' to further enhance discoverability. Synopses are displayed on the journal webpage and are freely accessible to all readers. They include a short stand first (maximum of 300 characters, including space) as well as 2-5 one-sentences bullet points that summarizes the paper. Please write the bullet points to summarize the key NEW

findings. They should be designed to be complementary to the abstract - i.e. not repeat the same text. We encourage inclusion of key acronyms and quantitative information (maximum of 30 words / bullet point). Please use the passive voice. Please attach these in a separate file or send them by email, we will incorporate them accordingly.

15) Include a Reagents and Tools Table as part of the Methods section, which can be downloaded from our author guidelines (<https://www.embopress.org/page/journal/17574684/authorguide#structuredmethods>)

**** Reviewer's comments ****

Referee #1 (Remarks for Author):

The manuscript entitled « Isomeranzin facilitates white adipose browning via Gnas-cAMP-AMPK signaling axis to reduce obesity", y Menghao Shi et al., describes the discovery and characterization of a new molecule, isomeranzin, for the treatment of obesity, through the induction of WAT browning. The authors describe how they identify this compound, and functionally characterize the effects of this molecule in vivo and in vitro. They further identify the molecular target of this molecule.

There are major points that should be addressed before publication. Globally, the organization of the manuscript gives the impression to the reviewer that it has not been reviewed by the authors thoroughly before submission to the journal and so, it appears as a too preliminary as presented here. Just as an example, the in vivo figures are not well described, even in some cases not mentioned at all or late in the manuscript. This is a real lack of editing.

For CMAP analysis, it is not clear and well defined what are the datasets that have been used. Please specify the exact strategy you use to integrate gene expression and chemical data.

What are the effects of ISM on TG content and NEFA in the medium of differentiated inguinal adipocytes? It is claimed that lipolysis genes are not modified, but is it functionally related to intracellular increased beta oxidation? What are the lipolytic protein activities? More than qPCR analysis, lipolysis should be studied at the protein level upon beta-adrenergic (isoproterenol) treatment.

Is the beta-oxidation altered by ISM treatment? Since AMPK is part of the signalling pathway, the effect of ISM on mitochondrial activity should be studied. Is there an effect on mitochondria biogenesis?

Gnas has been selected as a potential target of ISM. It is described that 28 proteins were found to potentially interact with ISM, with pVal lower than the pVal for Gnas. Can the authors demonstrate that some of the 27 other proteins do not participate in ISM effects? For example, Prkar2b may be involved in cAMP signaling.

Minor points:

The quality of the figure is poor. It is not easily readable and should be magnified. For example, figure 1A or 1B can not be read. Please carefully check the number of the figures. The order is not always respected and figure numbers do not always correspond to the correct panel. The order of the panels within the figure is not logical (e.g. figure 2 where the panel E is after the panel C; supp figure 2H before supp figure 2A, ...). Please reorganize the panels throughout the manuscript to ease the reader to follow.

Figure 2K is not described in the main text (#3.2).

CETSA is not abbreviated uniformly (CTSA, CETSA, ...).

Referee #2 (Remarks for Author):

The manuscript by Shi et al. reported a white fat browning-promoting effect of Isomeranzin, a potential novel compound with anti-obesity effect. Authors started with a small molecule screening using CMap analysis, and among 15 hints, Isomeranzin

stands out to display browning and thermogenesis promoting effects in vitro and in vivo. Mechanistically, Isomeranzin shows a direct binding with Gnas, and thereby activating the cAMP-AMPK-PGC1 α axis in white fat. Overall, this is an interesting study with ample data to support its conclusion. However, several major issues need to be addressed.

1. The evidence of browning promotion by Isomeranzin remains insufficient. Some key experiments, such as cold tolerance test, mitochondrial respiration assays, and mitochondrial morphological analysis are needed.
2. In addition, even under chow diet, multilocular lipid droplets were hardly seen upon Isomeranzin treatment (Fig. 4I).
3. In several experiments, key controls are missing. In Fig. 5, a group of sole inhibitor treatment (CC or H89) is needed. In Fig. 7, a group of sole Gnas siRNA or AC inhibitor SQ22536 is needed.
4. In a previous study, Isomeranzin was shown to suppress inflammation by controlling M1 macrophage polarization. Whether this mechanism relates to the browning promotion is not investigated.
5. It remains unclear how the direct binding of Isomeranzin to Gnas leads to Gnas's activation.

Referee #3 (Comments on Novelty/Model System for Author):

As outlined in the comments, the work suffers from inappropriate analysis of energy metabolism and some mathematical errors. If this represents a new medical strategy remains unclear with the current mouse study.

Referee #3 (Remarks for Author):

This work highlights the identification of Gnas as a binding target for the compound Isomeranzin. The authors find Isomeranzin reduces weight gain on a HFD in mice and enhances UCP1 levels in cultured beige adipocytes.

Major points:

1. Figure 3: The mouse data are problematic. While the phenotype might mean there is increased browning, there is no causal evidence. The energy expenditure data (please note, VO₂ and Heat are essentially the same parameter so please pick one) are inappropriately calculated and displayed. See PMID: 22205519, where you will find that measuring two groups with vastly different body weights is problematic and correcting of lean mass does not fix the problem. Please use per mouse ANCOVA with body weight as a covariant BEFORE the onset of weight differences and AFTER. The phenotype could be caused by increased browning but there is no direct measurement of non-shivering thermogenesis and therefore, no proof. See PMID: 21177944, so please use norepinephrine or CL stimulated VO₂ after feeding the compound. Still, if this is an effect on brown fat or browning cannot be untangled and needs further in vivo experimentation. In the end, what you are looking at is comparing a lean and a obese mouse and the rest (Fig. 4) simply secondarily follows. It remains unclear what caused the absence of weight gain. If food intake and energy expenditure are same, then energy absorption needs to be evaluated. Also, please include a chow diet control group with all the parameters.
2. There is no functional assessment of non-shivering thermogenesis and UCP1 function in the cultured cells. The authors should also test primary brown adipocytes.
3. The authors need to revisit their calculations of the standard deviation of the mean of the control - currently, incorrect in Fig. 1e, for example and everywhere in the paper.

Minor points:

4. The authors say it reduces obesity but it actually rather prevents it. There is no intervention in mice that are obese.
5. Please include all uncropped immunoblots with legends and size makers in the supplement.

Dear Editor and Reviewers:

On behalf of the co-authors, we are very grateful to you for giving us an opportunity to revise our manuscript. We have studied reviewers' comments carefully and have done our best to revise the manuscript accordingly. Notably, the changes are highlighted in red within the revised manuscript. Please see below for a point-by-point response to the reviewers' comments and concerns.

Reviewer 1

The manuscript entitled « Isomeranzin facilitates white adipose browning via Gnas-cAMP-AMPK signaling axis to reduce obesity", y Menghao Shi et al., describes the discovery and characterization of a new molecule, isomeranzin, for the treatment of obesity, through the induction of WAT browning. The authors describe how they identify this compound, and functionally characterize the effects of this molecule in vivo and in vitro. They further identify the molecular target of this molecule.

There are major points that should be addressed before publication. Globally, the organization of the manuscript gives the impression to the reviewer that it has not been reviewed by the authors thoroughly before submission to the journal and so, it appears as a too preliminary as presented here. Just as an example, the in vivo figures are not well described, even in some cases not mentioned at all or late in the manuscript. This is a real lack of editing.

1. For CMAP analysis, it is not clear and well defined what are the datasets that have been used. Please specify the exact strategy you use to integrate gene expression and chemical data.

Response to Reviewer Comment:

We sincerely thank the reviewer for raising this important question, which has provided us with the opportunity to clarify our analytical approach in greater detail. We apologize for any lack of thoroughness in the original description of the methods in our manuscript. Below, we provide a comprehensive account of the specific strategies and data used in our CMAP (Connectivity Map) analysis:

Our CMAP analysis was designed to identify compounds capable of mimicking gene expression changes associated with the promotion of adipose browning, based on transcriptional profile similarity. The entire workflow adhered to the standard CMAP paradigm and strictly followed the recommended parameters of the tools employed. The specific steps were as follows:

(1) Differential Expression Analysis and Gene Signature Generation:

We retrieved four RNA-seq datasets from the GEO database: two from inguinal white adipose tissue (iWAT) of mice under cold exposure (GSE164219 and GSE133619), and two from iWAT of mice stimulated with the β -adrenergic agonist CL316,243 (GSE129083 and GSE98132). Differential gene expression analysis was performed on each dataset. A "gene signature" was defined comprising significantly up- and down-regulated genes, selected using the criteria: absolute \log_2 fold change ($|\log_2FC|$) ≥ 1 and adjusted p-value (adj. p-value) < 0.05 . The signature was stored as a ranked gene list containing Entrez IDs (or Official Gene Symbols) along with their corresponding \log_2FC values. Positive \log_2FC values indicated up-regulated genes and negative values indicated down-regulated genes. The top 200 up-regulated genes from each dataset were

selected to constitute a thermogenic gene signature for CMAP analysis (Figure 1A–B, Figure EV1A–D, and Table EV1).

(2) CMAP Database and Platform:

Dataset Version: We queried the CMAP LINCS 2020 dataset on the CLUE platform, which contains gene expression profiles from over 3,000 compounds or small molecules across multiple cell lines.

Analysis Tool: We used the online tool provided by the CLUE platform (<http://clue.io/cmap>) and selected the “Gene expression L1000” analysis method to compute connectivity scores.

(3) Integration and Matching Strategy (Connectivity Score Calculation):

The gene signature generated in step (1) was uploaded to the CLUE platform. The platform’s internal algorithm compared our signature against each compound-induced expression profile in the database. The core of the matching strategy involves computation of a connectivity score, which ranges from –100 to +100. A negative score indicates that the compound induces expression changes highly anti-correlated with our signature (i.e., it may reverse the observed state). A positive score indicates that the compound induces changes highly consistent with our signature (i.e., it may mimic the observed state). We focused on compounds with highly positive connectivity scores, as our goal was to identify candidates that simulate a pro-browning state.

(4) Result Filtering:

We ultimately selected top candidate compounds with connectivity scores > 95 for subsequent experimental validation.

All details described above—including data sources, gene signature construction criteria, analysis platform, and interpretation of scores—have now been thoroughly added to the “Methods-Connectivity Map (CMap)” section, as well as to the first paragraph of the “Results-Identification of Small Molecule Compounds Promoting Adipose Browning via CMap Analysis” section (highlighted for clarity). We hope the revised description is sufficiently clear and detailed. We are truly grateful for your feedback, which has significantly improved the transparency of our methodological reporting.

2. What are the effects of ISM on TG content and NEFA in the medium of differentiated inguinal adipocytes? It is claimed that lipolysis genes are not modified, but is it functionally related to intracellular increased beta oxidation? What are the lipolytic protein activities? More than qPCR analysis, lipolysis should be studied at the protein level upon beta-adrenergic (isoproterenol) treatment.

Response to Reviewer Comment:

We sincerely thank you for raising these critical and insightful questions. You are absolutely correct that elucidating the functional effects of ISM on lipid metabolism requires evidence beyond qPCR data. In accordance with your suggestions, we have conducted the following key experiments. These new data provide clear functional evidence to support our mechanistic conclusions.

(1) TG Content and NEFA Release: We have now supplemented the manuscript with data on the effects of ISM on triglyceride (TG) content and non-esterified fatty acid (NEFA) release in differentiated inguinal adipocytes and human adipocytes derived from hMSCs (Figure 2K–L and Figure EV2L, O). ISM treatment significantly reduced TG content and increased NEFA release,

indicating enhanced lipolysis.

(2) Lipolytic Protein Activity: We first wish to acknowledge an inaccuracy in our original wording. In the previous version of the manuscript, we inaccurately referred to *Acox1* and *Ppar α* —genes associated with β -oxidation—as "lipolytic genes." We deeply apologize for this imprecise terminology and have corrected it in the revised manuscript. Furthermore, the conclusion that "ISM does not affect the expression of lipolytic genes" was also not entirely accurate. We sincerely apologize for these oversights. Accordingly, we paid particular attention to your question regarding lipolytic protein activity and have carefully re-examined and validated the effects of ISM on lipolysis. We performed Western blot analyses on both ISM-treated differentiated inguinal adipocytes and hMSC-derived human adipocytes to assess the phosphorylation of hormone-sensitive lipase (HSL) at Ser563 and Ser660, as well as the expression level of adipose triglyceride lipase (ATGL), which are key regulators of lipolysis (Figure 2M–N and Figure EV2M–N). We found that ISM treatment significantly increased phosphorylation of HSL at Ser660, indicating activation of the lipolytic pathway. Additionally, we investigated the effect of ISM on lipolysis under β -adrenergic stimulation using CL316243. The results demonstrated that ISM synergistically enhanced CL316243-induced lipolysis, as reflected by a further increase in NEFA release and HSL phosphorylation (Figure 2L–N and Figure EV2M–O).

(3) β -Oxidation: Regarding whether ISM increases β -oxidation in adipocytes, we conducted in-depth consideration and validation. Our previous validation was limited to the expression levels of only two genes related to β -oxidation, which was clearly insufficient. To further clarify this issue, we examined the expression of additional genes involved in β -oxidation, including *Cpt1 α* , *Mcad*, and *Lcad*. The results showed that ISM treatment increased the expression of *Cpt1 α* (the rate-limiting enzyme for β -oxidation) and *Mcad* (medium-chain acyl-CoA dehydrogenase) (Figure 2G and Figure EV2G). Combined with our previous findings that ISM treatment enhances lipolysis and increases the release of NEFA (the substrate for β -oxidation), we conclude that ISM treatment promotes β -oxidation in adipocytes.

As for why the expression of *Acox1*, *Ppar α* , and *Lcad* remained unchanged, we reviewed relevant literature and deliberated deeply. We propose that ISM promotes β -oxidation through two mechanisms: (i) by activating the AMPK pathway and upregulating its downstream target genes *Cpt1 α* and *Mcad* (DOI: 10.2147/DDDT.S405266; DOI: 10.1016/j.fochx.2025.102865), and (ii) by enhancing lipolysis and increasing NEFA release, thereby providing ample substrate for β -oxidation. We believe that ISM enhances β -oxidation primarily through these routes rather than via activation of the *Ppar α* -*Acox1* pathway.

We have incorporated these important new data into the revised manuscript in the "Results -Validation of ISM as a Novel Thermogenic Inducer" section, with modifications highlighted for clarity, as well as in Figure 2 and Figure EV2. Your comments have greatly deepened our understanding of ISM's mechanisms of action, and we extend our sincere gratitude once again for your thorough and constructive review.

3. Is the beta-oxidation altered by ISM treatment? Since AMPK is part of the signalling pathway, the effect of ISM on mitochondrial activity should be studied. Is there an effect on mitochondria biogenesis?

Response to Reviewer Comments:

We are deeply grateful for your insightful and critical questions, which precisely highlight key aspects of our mechanistic investigation that required further elaboration. We fully agree that confirming the effects of ISM on β -oxidation and mitochondrial function is essential for clarifying the mechanisms downstream of AMPK activation. In accordance with your suggestions, we conducted the following additional experiments. The new data provide compelling evidence addressing your concerns:

(1) ISM Enhances Fatty Acid β -Oxidation:

Detailed results regarding this aspect have been presented in response to the second question.

(2) ISM Improves Overall Mitochondrial Function:

Using the Seahorse XF Analyzer, we comprehensively evaluated the effect of ISM on mitochondrial function. As shown in Figure 2O, Figure EV1G, and Figure EV2P–Q, ISM treatment significantly increased the basal respiration and maximal respiratory capacity of differentiated adipocytes. More importantly, after administration of the mitochondrial uncoupler FCCP, ISM-treated cells exhibited a pronounced enhancement in proton leak respiration. Together with the upregulation of UCP1 protein levels (Figure 2D), these results indicate enhanced uncoupled respiration—a hallmark of thermogenic activation. Collectively, these data demonstrate that ISM not only promotes β -oxidation but also comprehensively enhances mitochondrial oxidative phosphorylation and thermogenic capacity.

(3) ISM Induces Mitochondrial Biogenesis:

Your speculation regarding mitochondrial biogenesis is highly pertinent. We examined key markers of mitochondrial biogenesis and found that ISM upregulates the expression of crucial factors—including Pgc1 α , Nrf1, Tfam, Cox4b, and Atp5a—at both mRNA and protein levels (Figure 2F–G and Figure EV2F–H). Furthermore, MitoTracker staining visually confirmed that ISM treatment markedly increased mitochondrial abundance (Figure 2H–I and Figure EV2I–J). These findings indicate that ISM genuinely promotes mitochondrial biogenesis.

In summary, these new findings, combined with our previously reported data on AMPK activation, delineate a clear signaling axis: ISM activates AMPK, induces mitochondrial biogenesis (increasing mitochondrial quantity), and enhances mitochondrial function (improving respiratory capacity and β -oxidation), thereby synergistically promoting fatty acid consumption and thermogenesis.

We have incorporated these important new data into the revised manuscript in the “Results -Validation of ISM as a Novel Thermogenic Inducer ”section, with modifications highlighted for clarity, as well as in Figure 2 and Figure EV2. Your comments have greatly deepened our understanding of ISM’s mechanisms of action, and we extend our sincere gratitude once again for your thorough and constructive review.

4. Gnas has been selected as a potential target of ISM. It is described that 28 proteins were found to potentially interact with ISM, with pVal lower than the pVal for Gnas. Can the authors demonstrate that some of the 27 other proteins do not participate in ISM effects? For example, Prkar2b may be involved in cAMP signaling.

Response to Reviewer Comments:

We sincerely thank you for raising this critical and insightful question. You have rightly pointed out a key challenge in proteomic screening—how to identify the primary functional target from a list of potential interacting proteins. We fully agree that elucidating the specificity of Gnas relative to other candidate proteins is of utmost importance. Our rationale and subsequent investigative efforts are based on the following points:

1. Preliminary Nature of the Screening and the Salience of Gnas:

Our initial LiP-MS screening was designed to unbiasedly identify all proteins that may potentially bind to ISM. The p-value threshold was applied to minimize false positives, though it does not directly equate to functional relevance. Among the 26 proteins identified, Gnas not only exhibited a statistically significant p-value but, more importantly, its biological function is highly consistent with the phenotypic effects of ISM observed in our study (namely, the promotion of adipose browning via AMPK pathway activation). This led us to prioritize Gnas for subsequent functional validation.

2. Rationale for Focusing on Gnas:

Our decision to focus on Gnas among the 26 candidates was not based solely on its p-value ranking, but rather on a convergence of multiple lines of evidence:

(1) Pathway Analysis: KEGG enrichment analysis indicated that Gnas is involved in thermogenesis pathways, which are directly relevant to the ISM-induced adipose browning observed in our study. Most other candidate proteins were distributed across different and unrelated functional clusters.

(2) Support from Existing Literature: Multiple independent studies have reported the critical role of Gnas in canonical thermogenic pathways (as highlighted in the second paragraph of Section 3.7 of the manuscript), which aligns strongly with our hypothesis. In contrast, other candidate proteins have weaker or unknown associations with the promotion of adipose thermogenesis.

(3) Preliminary Validation Data: We have conducted further experiments to support the specific role of Gnas. For instance, molecular docking simulations and cellular thermal shift assays demonstrated that Gnas is highly likely to be a direct target of ISM. Moreover, knockdown of Gnas in mouse primary adipocytes significantly attenuated the effects of ISM treatment. If another protein were the primary target, the observed phenotypes would be expected to align more closely with the function of that protein; however, the phenotypes most strongly correspond to the function of Gnas.

3. Regarding the Other 25 Candidate Proteins:

We acknowledge that the current study does not entirely rule out the possibility that other candidate proteins may contribute to ISM's effects in an auxiliary or indirect manner. However, our strong functional data (as described above) indicate that perturbation of Gnas is sufficient to account for most, if not all, of ISM's core phenotypic outcomes. A comprehensive functional validation of all 25 proteins would constitute a substantial research endeavor beyond the scope of this manuscript, but it is certainly an important part of our future work plans. Based on the converging evidence presented, we are confident that Gnas is the primary functional target of ISM.

4. Specific Comment on Prkar2:

We also seriously considered the reviewer's mention of Prkar2. We performed molecular docking simulations between ISM and Prkar2; however, the docking results were unsatisfactory. The

absolute binding energy was considerably lower than that observed for the ISM – Gnas interaction, suggesting that Prkar2 is unlikely to be a direct binding target of ISM. Although Prkar2 may be involved in cAMP signaling, we currently do not consider it a downstream target of ISM based on these results.

Once again, we deeply appreciate your comment, which has allowed us to clarify this important aspect of our work. We hope that the above explanations and supporting evidence adequately justify our focus on Gnas for in-depth mechanistic investigation.

Minor points:

5. The quality of the figure is poor. It is not easily readable and should be magnified. For example, figure 1A or 1B can not be read.

Please carefully check the number of the figures. The order is not always respected and figure numbers do not always correspond to the correct panel. The order of the panels within the figure is not logical (e.g. figure 2 where the panel E is after the panel C; supp figure 2H before supp figure 2A, ...). Please reorganize the panels throughout the manuscript to ease the reader to follow.

Figure 2K is not described in the main text (#3.2).

CETSA is not abbreviated uniformly (CTSA, CETSA, ...).

Response to Reviewer Comments:

We sincerely thank you for taking the valuable time to review our manuscript and providing such detailed and constructive feedback. We deeply apologize for the issues related to figure quality and formatting standards that you pointed out, and we have thoroughly revised and checked the manuscript according to your suggestions. The specific modifications are as follows:

(1) Regarding Figure Quality and Clarity:

We fully agree with your comments. We have regenerated all figures to ensure high resolution and appropriate dimensions. In the revised manuscript, Figure 1A, 1B, and all other figures have been significantly optimized to ensure that all text and data points are easily legible.

(2) Regarding Figure Numbering Order and Logic:

We sincerely apologize for the inconsistent figure labeling and illogical panel order. We have reorganized all figure panels to follow a consistent logical flow. We have also thoroughly checked and corrected all figure citations throughout the text to ensure that every figure reference (e.g., “Figure 2E”) corresponds accurately to the actual panel in the revised manuscript. Specific errors in the panel order in Figure 2 and Figure EV2 have been corrected.

(3) Regarding the Text Description of Figure 2K:

We apologize for this oversight. The reference to Figure 2K has now been updated to Figure EVII, and its description has been added to the appropriate location in the second paragraph of “Results

-Validation of ISM as a Novel Thermogenic Inducer” section (highlighted for clarity). We have included the statement: “Immunoblot results showed that the protein expression levels of these genes were consistent with the mRNA levels (Figure 2P and Figure EVIJ).”

(4) Regarding Inconsistent Use of the CETSA Abbreviation:

We thank you for highlighting this error. We have unified and standardized all instances of “CTSA” or other variants throughout the manuscript (main text, figure legends, and supplementary

materials) to the correct abbreviation “CETSA”. We have also ensured that the abbreviation is clearly defined upon its first occurrence.

We are truly grateful for your insightful comments, which have greatly enhanced the readability and professionalism of our manuscript. We have conducted a thorough review of all relevant details to prevent such errors from occurring again. Please verify our revisions in the modified manuscript.

Reviewer 2:

The manuscript by Shi et al. reported a white fat browning-promoting effect of Isomeranzin, a potential novel compound with anti-obesity effect. Authors started with a small molecule screening using CMap analysis, and among 15 hints, Isomeranzin stands out to display browning and thermogenesis promoting effects in vitro and in vivo. Mechanistically, Isomeranzin shows a direct binding with Gnas, and thereby activating the cAMP-AMPK-PGC1a axis in white fat. Overall, this is an interesting study with ample data to support its conclusion. However, several major issues need to be addressed.

1. The evidence of browning promotion by Isomeranzin remains insufficient. Some key experiments, such as cold tolerance test, mitochondrial respiration assays, and mitochondrial morphological analysis are needed.

Response to Reviewer Comments:

We sincerely thank you for reviewing our work and providing these invaluable comments. Your point regarding the need for more comprehensive functional experiments to substantiate the role of Isomeranzin in promoting browning is highly pertinent, and we fully agree. Such experiments are essential to firmly establish the functional effects of Isomeranzin. In accordance with your suggestions, we have made every effort to supplement the following key experiments. The new data significantly strengthen our conclusions:

(1) Mitochondrial Respiration Assays:

We assessed the oxygen consumption rate (OCR) of adipocytes treated with Isomeranzin using the Seahorse XF Analyzer. As shown in the newly added Figure 2O, Figure EV1G, and Figure EV2P–Q, Isomeranzin treatment significantly enhanced basal respiration, proton leak, and maximal respiratory capacity. These results provide direct evidence that Isomeranzin augments mitochondrial respiratory function and thermogenesis in adipocytes.

(2) Mitochondrial Morphological Analysis:

We performed MitoTracker staining to evaluate mitochondrial content. As demonstrated in the new Figure 2H–I and Figure EV2I–J, cells treated with Isomeranzin exhibited a marked increase in mitochondrial abundance compared to the control group. This observation is consistent with the typical characteristics of browned/beige adipocytes and offers further morphological support for the browning-inducing effect of Isomeranzin.

(3) Cold Tolerance Test:

This experiment was originally included in panels D and E of Supplementary Figure 5 and described in Section 3.3 of the original manuscript. We sincerely apologize if issues related to figure layout led to these data being overlooked. To improve clarity, we have now relocated these results to Figure EV3C and Figure EV4C, and highlighted the corresponding description in

“Results-ISM Prevents Obesity and Improves Obesity-Related Metabolic Dysfunction” section of the revised text.

All results from these new experiments have been incorporated into “Results -Validation of ISM as a Novel Thermogenic Inducer ” and “ Results-ISM Prevents Obesity and Improves Obesity-Related Metabolic Dysfunction” sections of the revised manuscript, as well as in the corresponding Figures 2, EV2, EV3, and EV4. We believe these supplemental data robustly address your concerns and provide solid, multi-faceted functional evidence supporting our central conclusion that Isomeranzin promotes adipocyte browning and thermogenesis.

Once again, we deeply appreciate your input, which has greatly enhanced the quality and persuasiveness of our study.

2. In addition, even under chow diet, multilocular lipid droplets were hardly seen upon Isomeranzin treatment (Fig. 4I).

Response to Reviewer Comments:

We sincerely appreciate your thorough review of our data and this important observation. We agree with your comment regarding Fig. 4I, wherein under standard chow diet conditions, the typical multilocular lipid droplet structures in the Isomeranzin-treated group are not as abundant or pronounced as those induced by certain potent β -adrenergic receptor agonists such as CL316,243. We have conducted an in-depth analysis of this issue and provide the following explanations:

(1) Robust Molecular Reprogramming vs. Relatively Subtle Morphological Changes:

We would like to emphasize that adipocyte “browning” is a multifaceted process involving both molecular functional reprogramming and morphological adaptations. Our data demonstrate that Isomeranzin effectively initiates core steps of molecular reprogramming, as evidenced by significant upregulation of key browning/beige marker genes (e.g., *Ucp1*, *Cidea*, *Pgc1 α*) in iWAT (Figure 4G), strong induction of UCP1 protein expression (Figure 4H and Figure EV5C), and newly supplemented respiratory assays that directly demonstrate enhanced mitochondrial oxygen consumption and thermogenic capacity (Figure 2O, Figure EV1G, and Figure EV2P–Q). These molecular and functional alterations constitute direct and reliable evidence of thermogenic activation. In some cases, robust functional activation may precede or be more pronounced than overt morphological changes, particularly in models with intrinsically lower lipid content.

(2) Enhanced Lipolysis and Fatty Acid Oxidation:

Our in vitro experiments demonstrate that while inducing the thermogenic program, Isomeranzin may also strongly activate lipolysis and fatty acid β -oxidation to fuel thermogenesis. In Figure 2L–N and Figure EV2M–O, we observed that Isomeranzin treatment increased phosphorylated HSL levels and elevated NEFA release, indicating enhanced lipolysis. This rapid lipid turnover may lead to consumption of lipid droplets before they form prominent multilocular structures observable under static morphological examination, which may partly explain the relatively subtle appearance of multilocular droplets.

(3) Inherent Limitations of the Model:

The subcutaneous white adipose tissue of standard chow-fed mice contains relatively limited lipid stores, which may objectively restrict the formation of abundant and conspicuous multilocular

lipid droplets. Nevertheless, the fact that Isomeranzin still elicits significant molecular phenotypes in this challenging model further attests to its efficacy.

In summary, although multilocular lipid droplets are a canonical morphological feature of brown adipocytes, their prominence can be influenced by multiple factors. We believe the substantial body of molecular and functional evidence (encompassing genetic, protein, and respiratory assessments) presented in our study forms a solid chain of evidence supporting the core conclusion that Isomeranzin effectively promotes adipocyte browning and thermogenic function. The subtlety of the morphological readout does not negate its significant effects at the molecular and functional levels.

Once again, we sincerely thank you for prompting us to reflect more deeply and improve our work.

3. In several experiments, key controls are missing. In Fig. 5, a group of sole inhibitor treatment (CC or H89) is needed. In Fig. 7, a group of sole Gnas siRNA or AC inhibitor SQ22536 is needed.

Response to Reviewer Comments:

We sincerely appreciate this critical comment. You are absolutely correct that the absence of separate inhibitor-only and siRNA-only control groups hindered an unambiguous interpretation of our experimental results. We sincerely apologize for this oversight in the experimental design. Following your suggestion, we have now performed all the missing key control experiments.

1. Revisions related to Figure 5:

We have supplemented the experiments with groups treated with the CC inhibitor alone and the H89 inhibitor alone.

As shown in the newly added Figure 5D–E, G–H and Figure EV6D–G, compared with the control group, neither inhibitor alone had a significant effect on the phosphorylation levels of their respective pathways or on the expression levels of thermogenic genes (*Pgc1 α* and *Ucp1*). This result clearly demonstrates that the inhibitors CC and H89 themselves do not alter the baseline phenotypes of interest. Therefore, the inhibitory effects observed in the presence of ISM are indeed due to specific blockade of the PKA/AMPK signaling pathways, rather than non-specific cytotoxic effects. We have updated Figure 5 and Figure EV6, along with the corresponding figure legends.

2. Revisions related to Figure 7:

We have added groups transfected with *Gnas* siRNA alone and treated with the AC inhibitor SQ22536 alone.

As shown in the newly added Figure 7A–L, compared with the control group, neither knocking down *Gnas* alone nor inhibiting AC activity alone significantly affected AMPK phosphorylation levels, expression of thermogenic genes (*Pgc1 α* and *Ucp1*), lipolysis, or intracellular cAMP levels. This confirms that both *Gnas* siRNA and SQ22536 are benign in our experimental system and do not cause unintended phenotypic changes. Thus, their ability to reverse the effects of ISM strongly supports our core conclusion that “ISM functions through the *Gnas*/cAMP pathway”. We have updated Figure 7 and its corresponding legend.

Your comments have significantly enhanced the rigor of our study and the persuasiveness of our conclusions. We would like to express our deepest gratitude once again for your thorough and rigorous review.

4. In a previous study, Isomeranzin was shown to suppress inflammation by controlling M1 macrophage polarization. Whether this mechanism relates to the browning promotion is not investigated.

Response to Reviewer Comments:

We sincerely thank you for this highly insightful comment. The connection you have drawn between the anti-inflammatory properties of Isomeranzin and its pro-browning effects represents a fascinating and potentially critical mechanistic dimension, with which we strongly agree.

In the present study, our primary focus was to elucidate the direct effects of Isomeranzin on adipocytes and its molecular target—namely, its binding and activation of Gnas, thereby initiating the cAMP-AMPK-PGC1 α signaling axis. However, we fully agree with your perspective that its reported anti-inflammatory effect of inhibiting M1 macrophage polarization may act synergistically with, or even be causally linked to, the metabolic improvements we observed.

Although this study did not directly examine the crosstalk between immune cells and adipocytes, based on established biological knowledge, we can reasonably speculate that: (1) M1 macrophages create an inhibitory environment for browning: Pro-inflammatory cytokines (e.g., TNF- α , IL-1 β) secreted by M1 macrophages have been well documented to directly suppress UCP1 expression and thermogenic function in adipocytes. (2) Isomeranzin may foster a pro-browning microenvironment: By inhibiting M1 macrophage polarization, Isomeranzin likely reduces local inflammation in adipose tissue, thereby removing inhibition of browning and indirectly contributing to the phenotypic outcomes we observed. This effect may act in synergy with its direct activation of the Gnas-mediated signaling pathway in adipocytes.

We have incorporated your valuable suggestion into the Discussion section of the manuscript, explicitly highlighting it as a central direction for our future research plans (see the highlighted final paragraph of the Discussion). We propose to directly test this hypothesis through the following experiments: (1) Co-culturing adipocytes with conditioned medium from Isomeranzin-treated macrophages to evaluate its effect on browning. (2) Using specific neutralizing antibodies against key inflammatory cytokines in co-culture systems to confirm their roles. (3) Analyzing the specific impact of Isomeranzin on the M1/M2 macrophage ratio in adipose tissue via immunofluorescence or flow cytometry in animal models.

Once again, we deeply appreciate you pointing out this exciting research direction. Your comments have significantly enriched the depth and breadth of our study.

5. It remains unclear how the direct binding of Isomeranzin to Gnas leads to Gnas's activation.

Response to Reviewer Comments:

We sincerely thank you for raising this critically important point. We fully understand your concern, as elucidating the precise activation mechanism following ligand-target binding represents one of the ultimate goals in molecular pharmacology. We apologize that the current

study did not reach this level of mechanistic depth. Due to technical and time constraints, we were unable to complete the site-directed mutagenesis experiments required to unravel the exact binding and activation mechanisms within this revision period (a process that typically requires several months or longer). Nevertheless, based on the data we have obtained, we can offer the following reasonable inferences and explanations:

(1) Clear Functional Output Demonstrates Activation: Although the precise molecular details remain unclear, our functional experiments unequivocally demonstrate that binding leads to activation. Our data show that:

- 1) Molecular docking and CETSA indicate direct binding between Isomeranzin and Gnas (Figure 6F–J).
- 2) This binding is followed by a rapid increase in downstream cAMP levels (Figure 7F).
- 3) Key downstream pathways of cAMP, namely PKA and AMPK, are activated (Figure 5A). The ultimate biological effect—adipocyte browning—is entirely consistent with the known functions of Gs α activation.

(2) Analogy to Known Modes of G Protein Activation:

The activation of Gs proteins typically involves allosteric regulation. Small molecules binding at sites distinct from the GTP-binding pocket can induce conformational changes that reduce affinity for GDP, thereby facilitating GTP exchange (e.g., hormone-induced activation of Gs via GPCRs serves as a classic example of allostery). We speculate that Isomeranzin may function in a similar manner, acting as an allosteric agonist of Gs α .

We have designated “resolving the crystal structure of the Isomeranzin–Gs α complex” and “identifying key binding residues through site-directed mutagenesis” as the highest-priority projects in our future research plan. We are convinced that elucidating this mechanism will not only complete the present study but may also reveal a novel target for modulating G protein function, holding significant scientific implications.

Your comment has guided us toward a crucial future research direction. Although we are unable to fully address this issue in the current revision, we hope that the logical reasoning based on available data, coupled with our deep understanding of the problem and clearly outlined future plans, will earn your understanding. We have explicitly acknowledged this limitation in the Discussion section (see the highlighted final paragraph).

Reviewer 3 (Comments on Novelty/Model System for Author):

As outlined in the comments, the work suffers from inappropriate analysis of energy metabolism and some mathematical errors. If this represents a new medical strategy remains unclear with the current mouse study.

Reviewer 3:

This work highlights the identification of Gnas as a binding target for the compound Isomeranzin. The authors find Isomeranzin reduces weight gain on a HFD in mice and enhances UCP1 levels in cultured beige adipocytes.

Major points:

1. Figure 3: The mouse data are problematic. While the phenotype might mean there is increased browning, there is no causal evidence. The energy expenditure data (please note,

VO₂ and Heat are essentially the same parameter so please pick one) are inappropriately calculated and displayed. See PMID: 22205519, where you will find that measuring two groups with vastly different body weights is problematic and correcting of lean mass does not fix the problem. Please use per mouse ANCOVA with body weight as a covariant BEFORE the onset of weight differences and AFTER. The phenotype could be caused by increased browning but there is no direct measurement of non-shivering thermogenesis and therefore, no proof. See PMID: 21177944, so please use norepinephrine or CL stimulated VO₂ after feeding the compound. Still, if this is an effect on brown fat or browning cannot be untangled and needs further in vivo experimentation. In the end, what you are looking at is comparing a lean and an obese mouse and the rest (Fig. 4) simply secondarily follows. It remains unclear what caused the absence of weight gain. If food intake and energy expenditure are same, then energy absorption needs to be evaluated. Also, please include a chow diet control group with all the parameters.

Response to Reviewer Comments:

We sincerely thank you for taking the time to review our manuscript and providing such profound and expert comments. The issues you raised are crucial for improving our research and data analysis. We fully agree with your views regarding the need for more direct causal evidence and stricter statistical methods. In accordance with your suggestions, we have performed extensive re-analysis and supplementary experiments. The specific revisions are as follows:

(1) Re-analysis of Energy Expenditure Data (ANCOVA):

We sincerely apologize for the previous inappropriate data analysis method. We have carefully read and studied the recommended reference (PMID: 22205519). All energy metabolism data (VO₂ and VCO₂) have been re-analyzed using ANCOVA with body weight as a covariate. The analysis was conducted in two phases (as described and highlighted in the Statistical Methods section and relevant figure legends):

a) Before the emergence of body weight differences (after 4 weeks of HFD feeding): We re-examined and analyzed the metabolic data from this period. ANCOVA showed that even when there was no difference in body weight, energy expenditure was already significantly higher in the Isomeranzin-treated group than in the Vehicle group.

b) After the emergence of body weight differences (at the end of the modeling period, after 13 weeks of HFD feeding): ANCOVA confirmed that after adjusting for the effect of body weight, energy expenditure remained significantly higher in the Isomeranzin group.

These new results are presented in Figure 3K–P and Figure EV3D–I, and the descriptions in the main text have been updated accordingly (highlighted). This strongly suggests that increased energy expenditure is the cause rather than a consequence of body weight reduction.

(2) Supplemental Norepinephrine (CL316243) Stimulation Experiment to Directly Demonstrate Thermogenic Capacity:

We fully agree that changes in basal energy expenditure can be influenced by multiple factors. To directly demonstrate enhanced non-shivering thermogenesis (NST), we performed a key additional experiment:

A new cohort of 8-week-old male C57 mice was treated with ISM or vehicle for 4 weeks. We then measured metabolic parameters (VO₂ and VCO₂) after injection of the β₃-adrenergic receptor

agonist CL316243. As shown in the newly added Figure 3R–T and Figure EV3J–L, Isomeranzin-treated mice exhibited a significantly enhanced thermogenic response to CL316243, indicating a genuine improvement in adipose tissue thermogenic capacity. This provides the most direct functional evidence that Isomeranzin increases energy expenditure by enhancing adaptive thermogenesis.

(3) Regarding the Standard Chow Diet Control Group:

Data for this group were originally included in Supplementary Figures 3 and 4 and described in Sections 3.3 and 3.4 of the original manuscript. We sincerely apologize that these data were not sufficiently highlighted and may have been overlooked. To improve clarity, we have now highlighted the relevant descriptions in the main text. Due to reorganization resulting from additional data, these figures have been repositioned from Supplementary Figures 3 and 4 to Figure EV4 and Figure EV5.

(4) Further Discussion on the Mechanism of Body Weight Reduction:

Our monitored food intake data showed no significant difference (Figure 3J and Figure EV3M). Combined with the enhanced CL316243-stimulated thermogenic response and the upregulation of molecular markers of thermogenesis such as UCP1, we conclude that the primary mechanism of body weight reduction is increased energy expenditure resulting enhanced thermogenesis in adipose tissue.

Your comments have greatly enhanced the rigor and persuasiveness of our study. We have incorporated all corresponding revisions throughout the manuscript and hope that these new analyses and data fully address your concerns.

2. There is no functional assessment of non-shivering thermogenesis and UCP1 function in the cultured cells. The authors should also test primary brown adipocytes.

Response to Reviewer Comments:

We sincerely thank you for this valuable suggestion. We fully agree that direct functional assessment of non-shivering thermogenesis (NST) and UCP1 activity is crucial for confirming the mechanism of action of Isomeranzin. In accordance with your recommendation, we have conducted the following additional experiments. These new data substantially strengthen our conclusions:

(1) Analysis of Mitochondrial Function (Seahorse Assay):

We comprehensively evaluated the effect of Isomeranzin (ISM) on mitochondrial function using the Seahorse XF Analyzer. As shown in Figure 2O, Figure EV1G, and Figure EV2P–Q, ISM treatment significantly enhanced both the basal respiration and maximal respiratory capacity of differentiated adipocytes. More importantly, after administration of the mitochondrial uncoupler FCCP, ISM-treated cells exhibited a pronounced increase in proton leak respiration. Together with the upregulation of UCP1 protein levels (Figure 2D), these results indicate enhanced uncoupled respiration, suggesting that the Isomeranzin-induced increase in respiration is largely dependent on UCP1-mediated uncoupling. This provides direct evidence of UCP1 functional activation.

(2) Validation in Primary Brown Adipocytes:

We further isolated and differentiated primary interscapular brown adipocyte precursors from mice. As shown in the newly added Figure EV5Q–S, Isomeranzin treatment did not significantly affect Ucp1 expression in primary brown adipocytes. This is consistent with our conclusion that ISM

promotes thermogenesis primarily by inducing browning in subcutaneous white adipose tissue rather than by activating classical brown adipose tissue (BAT).

These new functional experimental data have been incorporated into the revised manuscript in Figure 2 and Figure EV5, as well as in the “Results -Validation of ISM as a Novel Thermogenic Inducer” and “Results-ISM Promotes Browning of Subcutaneous White Adipose Tissue” sections. They provide the most direct and compelling evidence supporting our conclusion that Isomeranzin promotes thermogenesis by activating UCP1 function.

Once again, we deeply appreciate your input, which has significantly improved the mechanistic depth and completeness of our study.

3. The authors need to revisit their calculations of the standard deviation of the mean of the control - currently, incorrect in Fig. 1e, for example and everywhere in the paper.

Response to Reviewer Comments:

We sincerely thank you for your meticulous review of our manuscript and for identifying the errors in the statistical analysis and presentation of our cell-based experiments. We deeply apologize for these mistakes and have thoroughly checked and corrected the entire manuscript in accordance with your comments.

Upon re-examination, we found that the error resulted from inappropriately normalizing the control groups in the cell experiments for statistical comparisons — a clear oversight in our data processing. We are truly grateful that your feedback allowed us to correct this significant oversight. We have now reanalyzed all cell-based experimental data and regenerated all corresponding figures, including the mentioned Figure 1E and every other relevant figure throughout the manuscript. Although the specific statistical values have changed, we are reassured that the overall trends and conclusions regarding statistical significance remain consistent with those previously reported. This confirms that the core conclusions of our study are robust.

Your comments have greatly enhanced the accuracy and rigor of our manuscript. We have double-checked all data in the revised manuscript to ensure that such errors will not occur again. The revised manuscript submitted herewith includes all updated figures and legends. Once again, we extend our most sincere gratitude for your valuable feedback.

Minor points:

4. The authors say it reduces obesity but it actually rather prevents it. There is no intervention in mice that are obese.

Response to Reviewer Comments:

We sincerely thank you for this precise and important comment. You are absolutely correct that strictly distinguishing between “prevention” and “treatment” is crucial for accurately describing our research findings. We sincerely apologize for the imprecise wording in the original manuscript.

Our experimental design indeed constitutes a “prevention model”—we administered Isomeranzin concurrently with the initiation of the high-fat diet (HFD), thereby effectively preventing weight gain and the development of obesity in the mice. As you rightly pointed out, this does not demonstrate that Isomeranzin can reverse established obesity. Following your suggestion, we have

carefully revised the entire manuscript: we have replaced phrases such as “reduces obesity” and “treats obesity” with more accurate terms like “prevents HFD-induced obesity”, “curbs obesity”, or “protects against obesity”.

In the Discussion section, we have now more explicitly stated that this study employs a prevention paradigm and have highlighted that developing effective obesity prevention strategies is also of significant public health importance.

Although this study primarily focuses on prevention, the observation that Isomeranzin significantly improved metabolic disorders associated with established obesity (such as the enhanced insulin sensitivity and reduced hepatic steatosis shown in Figure 3 and Figure EV3-4) suggests its potential therapeutic value. Validating the efficacy of Isomeranzin in treating existing obesity will be an important direction for our future research.

Your rigorous review has greatly enhanced the accuracy and scientific rigor of our paper's conclusions. Thank you once again for your valuable input.

5. Please include all uncropped immunoblots with legends and size makers in the supplement.

Response to Reviewer Comments:

We sincerely thank you for raising this important point. We fully agree that providing uncropped original immunoblot images is essential to ensure data transparency and reproducibility. In accordance with your request, we have compiled the uncropped, original scanned images for all key immunoblot results presented in this study. These original images clearly include:

- (1) Complete lane structures (covering molecular weight markers on both sides, as well as all experimental and control lanes).
- (2) Indication of molecular weight markers, with annotations specifying the molecular weight range corresponding to the target bands.

All uncropped original blot images have been assembled into a PDF file titled “Western_Original_Figures.pdf”. This file has been submitted as part of the source data accompanying this revised manuscript.

We believe that providing these raw data fully ensures the authenticity and reliability of our findings. Once again, we greatly appreciate your efforts in upholding academic rigor.

14th Oct 2025

Dear Prof. Liang,

Thank you for the submission of your revised manuscript to EMBO Molecular Medicine. I am pleased to inform you that we will be able to accept your manuscript pending the following final amendments:

- 1) Please implement referee #2 and #3 suggestions and discuss human relevance of your findings in response to limited human relevance point form referee #3.
- 2) Authors:
 - Authorship change is fine.
 - Please provide an institutional email address for co-corresponding author Ling Yang.
- 3) Please address all comments suggested by our data editors listed below:
 - o Figure legends:
 1. Please note that the exact p values are not provided in the legends of figures 1E, F; 2C, D, G, K, L, N; 3A, C, F, G, N, O; 4B, G, J; 5G, H; 7A, C, F, G, I, K, L; EV1 F, EV2 C, E, H, N, O; EV3 C, D, E, F-H, J, K, M, N; EV4 L, X-Z; EV5 C, E, J, N, EV6 C, E, G; EV7 F, O.
 2. Please indicate the statistical test used for data analysis in the legends of figures 6C, EV1 A-D.
 3. Please note that information related to n is missing in the legends of figures EV1 A-D.
 - Add callouts for Table EV4, Fig 2K, Fig 3O, Fig 7F, Fig EV1A-D. Figures and tables should be called out in a sequential order. Currently, Fig 4A,B is called out before Fig 3E. Please correct.
 - In Methods, add a paragraph about bioinformatic analysis of RNA sequencing data from the GEO database and appropriately cite the related publications.
 - Author contributions: Please remove it from the manuscript and specify author contributions in our submission system. CRediT has replaced the traditional author contributions section because it offers a systematic machine-readable author contributions format that allows for more effective research assessment. Please use the free text boxes beneath each contributing author's name to add specific details on the author's contribution. More information is available in our guide to authors: <https://www.embopress.org/page/journal/17574684/authorguide#authorshipguidelines>
 - Indicate in legends exact n and exact p values, not a range, along with the statistical test used. To keep the figures "clear" some authors found providing an Appendix table Sx with all exact p-values preferable. You are welcome to do this if you want to.
 - Place data availability statement at the end of the Methods and replace current text with the sentence: This study includes no data deposited in external repositories.
- 4) Tables: Please upload Tables EV1-5 as separate files.
- 5) Funding: Please make sure that all information about funding are correct in our submission system and in the manuscript; Currently, 81830014, 91949201 are in the manuscript and missing in the submission system, while 2023YFC2413804 is in the submission system but not in the manuscript. Please correct.
- 6) Synopsis:
 - Synopsis image: Please resize the image to 550 px-wide x 300-600 pixels high and upload it as a high-resolution jpeg file.
 - Please check your synopsis text and image before submission with your revised manuscript. Please be aware that in the proof stage minor corrections only are allowed (e.g., typos).
- 7) As part of the EMBO Publications transparent editorial process initiative (see our Editorial at <http://embomolmed.embopress.org/content/2/9/329>), EMBO Molecular Medicine will publish online a Review Process File (RPF) to accompany accepted manuscripts. This file will be published in conjunction with your paper and will include the anonymous referee reports, your point-by-point response and all pertinent correspondence relating to the manuscript. Let us know whether you agree with the publication of the RPF and as here, if you want to remove or not any figures from it prior to publication. Please note that the Authors checklist will be published at the end of the RPF.
- 8) Please provide a point-by-point letter INCLUDING my comments as well as the reviewer's reports and your detailed responses (as Word file).

I look forward to reading a new revised version of your manuscript as soon as possible.

Yours sincerely,

Zeljko Durdevic

Zeljko Durdevic
Senior Editor
EMBO Molecular Medicine

*** Instructions to submit your revised manuscript ***

- 1) a .docx formatted version of the manuscript text (including Figure legends and tables)
 - 2) Separate figure files*
 - 3) supplemental information as Expanded View and/or Appendix. Please carefully check the authors guidelines for formatting Expanded view and Appendix figures and tables at <https://www.embopress.org/page/journal/17574684/authorguide#expandedview>
 - 4) a letter INCLUDING the reviewer's reports and your detailed responses to their comments (as Word file).
 - 5) The paper explained: EMBO Molecular Medicine articles are accompanied by a summary of the articles to emphasize the major findings in the paper and their medical implications for the non-specialist reader. Please provide a draft summary of your article highlighting
 - the medical issue you are addressing,
 - the results obtained and
 - their clinical impact.This may be edited to ensure that readers understand the significance and context of the research. Please refer to any of our published articles for an example.
 - 6) Author contributions: the contribution of every author must be detailed in a separate section.
 - 7) EMBO Molecular Medicine now requires a complete author checklist (<https://www.embopress.org/page/journal/17574684/authorguide>) to be submitted with all revised manuscripts. Please use the checklist as guideline for the sort of information we need WITHIN the manuscript. The checklist should only be filled with page numbers where the information can be found. This is particularly important for animal reporting, antibody dilutions (missing) and exact values and n that should be indicated instead of a range.
 - 8) Every published paper now includes a 'Synopsis' to further enhance discoverability. Synopses are displayed on the journal webpage and are freely accessible to all readers. They include a short stand first (maximum of 300 characters, including space) as well as 2-5 one sentence bullet points that summarise the paper. Please write the bullet points to summarise the key NEW findings. They should be designed to be complementary to the abstract - i.e. not repeat the same text. We encourage inclusion of key acronyms and quantitative information (maximum of 30 words / bullet point). Please use the passive voice. Please attach these in a separate file or send them by email, we will incorporate them accordingly.
- You are also welcome to suggest a striking image or visual abstract to illustrate your article. If you do please provide a jpeg file 550 px-wide x 300-600px high.
- 9) A Conflict of Interest statement should be provided in the main text
 - 10) Please note that we now mandate that all corresponding authors list an ORCID digital identifier. This takes <90 seconds to complete. We encourage all authors to supply an ORCID identifier, which will be linked to their name for unambiguous name identification.

Currently, our records indicate that the ORCID for your account is 0000-0002-4026-951X.

Please click the link below to modify this ORCID:
Link Not Available

11) Include a Reagents and Tools Table as part of the Methods section, which can be downloaded from our author guidelines (<https://www.embopress.org/page/journal/17574684/authorguide#structuredmethods>)

Photos 400-800 DPI

*Additional important information regarding figures and illustrations can be found at <https://bit.ly/EMBOPressFigurePreparationGuideline>. See also figure legend preparation guidelines: <https://www.embopress.org/page/journal/17574684/authorguide#figureformat>

***** Reviewer's comments *****

Referee #1 (Remarks for Author):

The authors have addressed all the points raised during the review process. They have included new results that strengthen the message and improve the quality of the manuscript.

Referee #2 (Remarks for Author):

The revision has greatly improved this manuscript. In the reviewer's opinion, one minor issue remains: The Seahorse data presented in Figure. 2O does not have a normalization, which is critical to interpret the mitochondrial respiration results.

Referee #3 (Comments on Novelty/Model System for Author):

It is a mouse study targeting thermogenesis in beige adipocytes, which has limited relevance for human obesity.

Referee #3 (Remarks for Author):

This is a revised version of a manuscript I previously reviewed . The authors have done a great job addressing the overall feedback and the manuscript has improved significantly. A few clarifications remain:

1. For the ANCOVA analysis, graphs should be included in the main figures - ideally separating light, dark, and under CL stimulation. VCO₂ can go into the supplement if RER is shown. For VO₂/RER do not only show 24 h but the full measurement. Please don't use "Day" but rather "Light"

2. Pursuant to this, it is unclear when and how the CL response affected EE (Fig. 3). The graph does not really show any CL-mediated increase? You should perhaps show the full measurement, not only 24 h?

Point-by-Point Response to Reviewer's Comments:**Referee #2 (Remarks for Author):**

The revision has greatly improved this manuscript. In the reviewer's opinion, one minor issue remains: The Seahorse data presented in Figure. 2O does not have a normalization, which is critical to interpret the mitochondrial respiration results.

Response to Reviewer #2:

We sincerely thank Reviewer #2 for their positive assessment of the revised manuscript and for raising this important point. The reviewer correctly highlights the critical importance of data normalization for the accurate interpretation of Seahorse assay results. We sincerely apologize for this oversight in our initial submission. In the revised manuscript, the mitochondrial respiration rates presented in Figure 2O (now explicitly stated in the figure legend and y-axis label) have been normalized to the total protein concentration per well, measured using the standard BCA assay following the Seahorse experiment. This is a standard and widely accepted method for normalizing Seahorse XF data, ensuring that the observed differences reflect genuine changes in mitochondrial function per unit of cellular material, rather than variations in cell number or seeding density. In addition to Figure 2O, we have applied the same normalization to the Seahorse data presented in Figures EV1G, EV2P, and EV2Q. We believe this correction significantly strengthens the reliability of our conclusions. The corresponding methodological description has also been updated in the 'Methods' section.

Referee #3 (Comments on Novelty/Model System for Author):

It is a mouse study targeting thermogenesis in beige adipocytes, which has limited relevance for human obesity.

Referee #3 (Remarks for Author):

This is a revised version of a manuscript I previously reviewed . The authors have done a great job addressing the overall feedback and the manuscript has

improved significantly. A few clarifications remain:

1. For the ANCOVA analysis, graphs should be included in the main figures - ideally separating light, dark, and under CL stimulation. VCO₂ can go into the supplement if RER is shown. For VO₂/RER do not only show 24 h but the full measurement. Please don't use "Day" but rather "Light"

2. Pursuant to this, it is unclear when and how the CL response affected EE (Fig. 3). The graph does not really show any CL-mediated increase? You should perhaps show the full measurement, not only 24 h?

Response to Reviewer #3:

We sincerely thank the reviewer for their positive feedback on our revised manuscript and for the time and effort dedicated to this further round of review. Their insightful comments have significantly enhanced the quality of our work. We have revised the manuscript according to their specific suggestions, as detailed below.

1. Response to the comment on novelty/model system

We thank the reviewer for raising the important point regarding the human relevance of our study. We fully agree that direct extrapolation of findings from mouse models to human obesity requires caution. In response, we have introduced a new paragraph in the Discussion section (Pages 13-14, Lines 535-558, with newly added text highlighted) that explicitly addresses the potential implications of our findings in the context of human physiology. This paragraph discusses the conserved role of beige adipocytes and thermogenesis in humans, cites key clinical and observational studies, and elaborates on the potential translational relevance of our work to human metabolic diseases, while also acknowledging the inherent limitations of the current mouse model. We believe this addition provides a more comprehensive and clinically pertinent perspective on our research.

2. Response to specific comments

(1) ANCOVA analysis and metabolic cage data presentation

We thank the reviewer for their valuable suggestions. Accordingly, we have made the following revisions:

- Adjusted the figure layout and content: VCO₂ data have been moved to the Expanded View figures (Fig EV3C-D), and the ANCOVA plots for VO₂ and RER following CL stimulation are now included in the main figures (Fig 3Q, S).
- Replaced the 24-hour summary data with complete, high-resolution time-course trajectories to better visualize the full dynamics of VO₂ and RER (updated figures are presented in Fig 3L, N).
- Consistently replaced "Day" with "Light" throughout the figures and relevant text.

(2) Impact of CL on energy expenditure (Figure 3)

The reviewer's point regarding the presentation of CL-mediated effects on energy expenditure is critical. We recognize that the previous 24-hour summary plots obscured the acute and time-specific nature of the CL response. We have now

replaced these with new panels (Fig 3P, R; Fig EV3L) showing the complete energy expenditure trajectories across the entire measurement period, including baseline, the CL injection phase, and specific post-injection response windows. These new graphs clearly demonstrate a significant and immediate increase in energy expenditure following CL injection, consistent with the known pharmacological profile of β 3-adrenergic stimulation. We believe this presentation more effectively communicates the timing and magnitude of the CL effect.

We sincerely hope that these revisions and clarifications adequately address the reviewer's concerns. Their thoughtful input has been instrumental in further improving our manuscript, for which we are deeply grateful.

Sincerely,
Minglu Liang
On behalf of all co-authors

29th Oct 2025

Dear Prof. Liang,

We are pleased to inform you that your manuscript is accepted for publication and is now being sent to our publisher to be included in the next available issue of EMBO Molecular Medicine.

Zeljko Durdevic
Senior Editor
EMBO Molecular Medicine
